# Cross-biobank generalizability and accuracy of electronic health record-based predictors compared to polygenic scores

Kira E. Detrois [1,6], Tuomo Hartonen [1,6], Maris Teder-Laving [2], Bradley Jermy[1], Kristi Läll[2], Zhiyu Yang [1], Estonian Biobank research team*, FinnGen*, Reedik Mägi [2], Samuli Ripatti [1,3,4,5] & Andrea Ganna [1,3,4] ✉

Electronic health record (EHR)-based phenotype risk scores (PheRS) leverage individuals' health trajectories to estimate disease risk, similar to how polygenic scores (PGS) use genetic information. While PGS generalizability has been studied, less is known about PheRS generalizability across healthcare systems and whether PheRS are complementary to PGS. We trained elastic-net-based PheRS to predict the onset of 13 common diseases for 845,929 individuals (age = 32–70 years) from three biobank-based studies in Finland (FinnGen), the UK (UKB) and Estonia (EstB). All PheRS were statistically significantly associated with the diseases of interest and most generalized well without retraining when applied to other studies. PheRS and PGS were only moderately correlated and models including both predictors improved onset prediction compared to PGS alone for 8 of 13 diseases. Our results indicate that EHR-based risk scores can transfer well between EHRs, capture largely independent information from PGS, and provide additive benefits for disease risk prediction.

With the advent of large-scale genetic studies and the widespread availability of electronic health record (EHR) data, it is possible to combine these resources to more efficiently predict the risk of a wide range of diseases[1–3]. Disease risk estimation can guide the efficient allocation of screening, preventative interventions and treatments in the early stages of diseases. Two lines of research have emerged in the past years. Some researchers have focused on machine learning approaches for EHR data[2,4] and showed some promising results in deriving EHR-based predictors for pancreatic cancer[5] and cardiovascular disease[6–8], among others. Many studies have focused on genetic data. Polygenic scores (PGS) use combined information from a person's genome to estimate their genetic risk of developing a specific disease or trait. Numerous studies have examined the predictive ability of PGS across multiple diseases, and there is an extensive discussion about their clinical and public health value[9].

EHR and PGS-based prediction models have different strengths and limitations. EHRs allow access to a vast variety of data, including but not limited to disease diagnosis history, laboratory measurements, free text reports and various socioeconomic information[10]. However, EHRs are also known to be noisy[1,2], and the models are expected to suffer from poor generalizability because of differences in data availability, as well as in clinical and recording practices across healthcare systems[2,3,5,10–13]. So far, most research has been conducted on a single EHR with limited work on validating the models in different EHR systems and countries[2,12,14,15]. Recent studies, however, show promising results when validating EHR-based predictors in different healthcare systems in the US and UK. For example, an EHR-based prediction model trained in a US study (BioMe) outperformed conventional clinical guidelines in predicting coronary artery disease (CAD) susceptibility, and the results could be externally replicated in the UK Biobank (UKB)[7,8].

[1]Institute for Molecular Medicine Finland, FIMM, HiLIFE, University of Helsinki, Helsinki, Finland. [2]Estonian Genome Centre, Institute of Genomics, University of Tartu, Tartu, Estonia. [3]Broad Institute of MIT and Harvard, Cambridge, MA, USA. [4]Analytic and Translational Genetics Unit, Massachusetts General Hospital, Boston, MA, USA. [5]Department of Public Health, University of Helsinki, Helsinki, Finland. [6]These authors contributed equally: Kira E. Detrois, Tuomo Hartonen. *A list of authors and their affiliations appears at the end of the paper. ✉e-mail: andrea.ganna@helsinki.fi

A similar recent study successfully transferred an EHR-based model trained in the BioMe study for the prediction of autoimmune diseases to All of Us, another US-based study. Another systematic effort to train deep learning-based prediction models on the UKB EHR data for 1,568 diseases showed that when transferring these models to the All of Us study, 1,347 (85.9%) of the models improved disease onset prediction over a baseline model with age and sex[16].

The PGS are less likely to suffer from measurement errors compared to EHR-based models; however, they are known to be poorly transferable across ancestries, thus risking increasing health disparities[17,18]. PGS are also not routinely measured in healthcare settings, although some healthcare systems have piloted programs to return PGS to individuals[19,20]. Further, as PGS keep improving through larger and more representative genome-wide association studies (GWAS), there is a growing interest in the integration of other predictors and risk factors to better capture the disease risk of individuals. Some recently published studies have integrated, for example, proteomics[21,22] or metabolomics-based risk scores[23,24] with PGS. Compared to omics, EHR data has the advantage that it is already routinely electronically collected in many countries and does not require invasive and often relatively expensive additional measurements[3]. Notably, there is a gap in our understanding of how PGS complements both established clinical risk factors and EHR-based risk scores. Numerous studies have investigated the additive value of PGS with clinical risk factors for a subset of diseases, including type 2 diabetes (T2D) and CAD[25–27]. For EHR-based risk scores and many other diseases, the added benefit of PGS for disease onset prediction and risk stratification remains understudied.

In this study, we aimed to directly compare, within and across studies, the predictive performance and generalizability of EHR-based scores versus PGS using a longitudinal prospective design. In this context, we define generalizability as the extent to which models trained in one setting (for example, a specific study or population) maintain their predictive accuracy and associations when applied to another, previously unseen context (for example, a different study or population). We conducted this comparison across 13 common diseases and three large biobank-based studies with high-quality EHR data—UK Biobank[28] (UKB; United Kingdom), FinnGen[29] (Finland) and Estonian Biobank (EstB; Estonia)[30]. We created the EHR-based scores using the PheRS framework[31,32] with PheRS derived from longitudinal diagnostic codes translated into consistent disease diagnoses using phecodes[33].

## Results

### Study overview
We included 845,929 individuals (Supplementary Table 1) aged 32–70 years on 1 January 2011 (Fig. 1a). These individuals belong to three biobank-based studies (FinnGen, UKB, EstB) linked with national registers or EHRs. The individuals gathered a total of 293,019 new diagnoses during an 8-year prediction period (1 January 2011 to 31 December 2018) across the following 13 common and high-burden diseases: prostate cancer, breast cancer, colorectal cancer, lung cancer, T2D, atrial fibrillation (AF), major depressive disorder (MDD), coronary heart disease (CHD), hip osteoarthritis (hip OA), knee osteoarthritis

(knee OA), asthma, gout and epilepsy. We observed the highest number of events for knee OA ($n$ = 43,767) and the lowest for lung cancer ($n$ = 4,796; Fig. 1c and Supplementary Table 2).

### Construction of PGS and PheRS
We constructed the PGS and PheRS separately for each disease (Fig. 1b). PGS were previously derived by the INTERVENE consortium[34]. PheRS were based on phecodes[35] recorded during a 10-year observation period (1 January 1999 to 31 December 2009; Fig. 1a), separated from the prediction period (1 January 2011 to 31 December 2018) by a 2-year washout period, during which no phecodes were recorded, meaning all the predictors were collected at minimum two years before the disease occurrence. Overall, we considered 234 phecodes with a prevalence of at least 1% in any study. However, for each disease, we excluded closely related diagnoses as predictors based on the phecodes exclusion ranges (Methods; Supplementary Table 3). For example, we did not use phecodes for secondary diabetes, T1D, or abnormal glucose tolerance as predictors of T2D.

Each PheRS model was trained separately to predict disease occurrence in the prediction period using 50% of the individuals in each study. We used elastic net models, a type of regularized regression method that combines the properties of both Ridge (L2) and Lasso (L1) regression[32]. The effect of age and sex was regressed out from the PheRS, and, when comparing PheRS and PGS, the first ten genetic principal components (PCs) were also regressed out from the scores to make them comparable. A more detailed description of the PheRS construction can be found in the Methods. Disease prevalence during the prediction period (1 January 2011 to 31 December 2018; Fig. 1c) varied substantially across the three studies. For example, we found a higher prevalence of knee OA in the EstB (12.3%, $n$ = 14,180) compared to FinnGen (4.8%, $n$ = 12,874) and the UKB (3.6%, $n$ = 16,713), while T2D diagnoses showed a lower prevalence both in the EstB (3.6%, $n$ = 104,161) and UKB (3.6%, $n$ = 16,850) compared to FinnGen (6.8%, $n$ = 18,099; Supplementary Table 2).

### PheRS were significantly associated with all 13 diseases
We evaluated the association between PheRS and 13 diseases independently of age and sex using Cox proportional hazard models (Cox-PH[36]) on a test set in each study. All PheRS were significantly associated ($P$ < 0.05; Fig. 2a and Supplementary Tables 4 and 5) with higher disease risk, with the largest association for gout (meta-analyzed hazard ratio (HR) per 1 s.d. of PheRS = 1.59, 95% confidence interval (CI) = 1.47–1.71), T2D (HR = 1.49, 95% CI = 1.37–1.61), and lung cancer (HR = 1.46, 95% CI = 1.39–1.54). Further, adding the PheRS to a baseline model with age and sex significantly ($P$ < 0.05; one-tailed $P$ values based on the $z$ scores of the c-index differences) improved the predictive accuracy (c index) in all three studies for 7 of 13 diseases—asthma, MDD, T2D, knee OA, hip OA, gout and AF (Extended Data Fig. 1a and Supplementary Tables 6 and 7; additionally, Supplementary Table 8 shows the area under the precision–recall curve results). However, in the meta-analysis, the differences in baseline (age + sex) c indices in the different studies meant that only the improvements for MDD, gout, epilepsy, and asthma were significant ($P$ < 0.05; Fig. 2b and Supplementary Table 7).

---

**Fig. 1 | Overview of the study design and number of diagnoses across the three biobank-based studies. a**, Outline of the study design. A separate study was conducted for each of the 13 diseases in the three biobank-based studies. Each study consisted of an observation and a prediction period, separated by a washout period that starts two years before the baseline date and during which no predictors were recorded. Each disease's case and control definitions were based on diagnoses acquired in the prediction period (1 January 2011 to 31 December 2018). We removed all individuals diagnosed before our baseline (1 January 2011) and only considered adults aged 32–70 years in 2011 (see Methods for more details). **b**, We compared the PGS with PheRS—trained on phecodes recorded during the observation period (1 January 1999 to

31 December 2008). The PGS were based on recent publicly available GWAS summary statistics using MegaPRS. Ultimately, each individual was assigned 13 different PGS and PheRS scores describing their risk of getting a disease diagnosis during the prediction period. We trained the PheRS on 50% of individuals separately in the three studies (FinnGen, UKB and EstB). In each study, we then used the other half of the population as a test set, where we used the scores as predictors in Cox-PH. **c**, Number of new diagnoses for each disease during the prediction period (1 January 2011 to 31 December 2018) for each of the 13 diseases in the three studies (green, EstB; peach, FinnGen; and brown, UKB). The figure is created with BioRender.com. COPD, chronic obstructive pulmonary disease; OA, osteoarthritis.

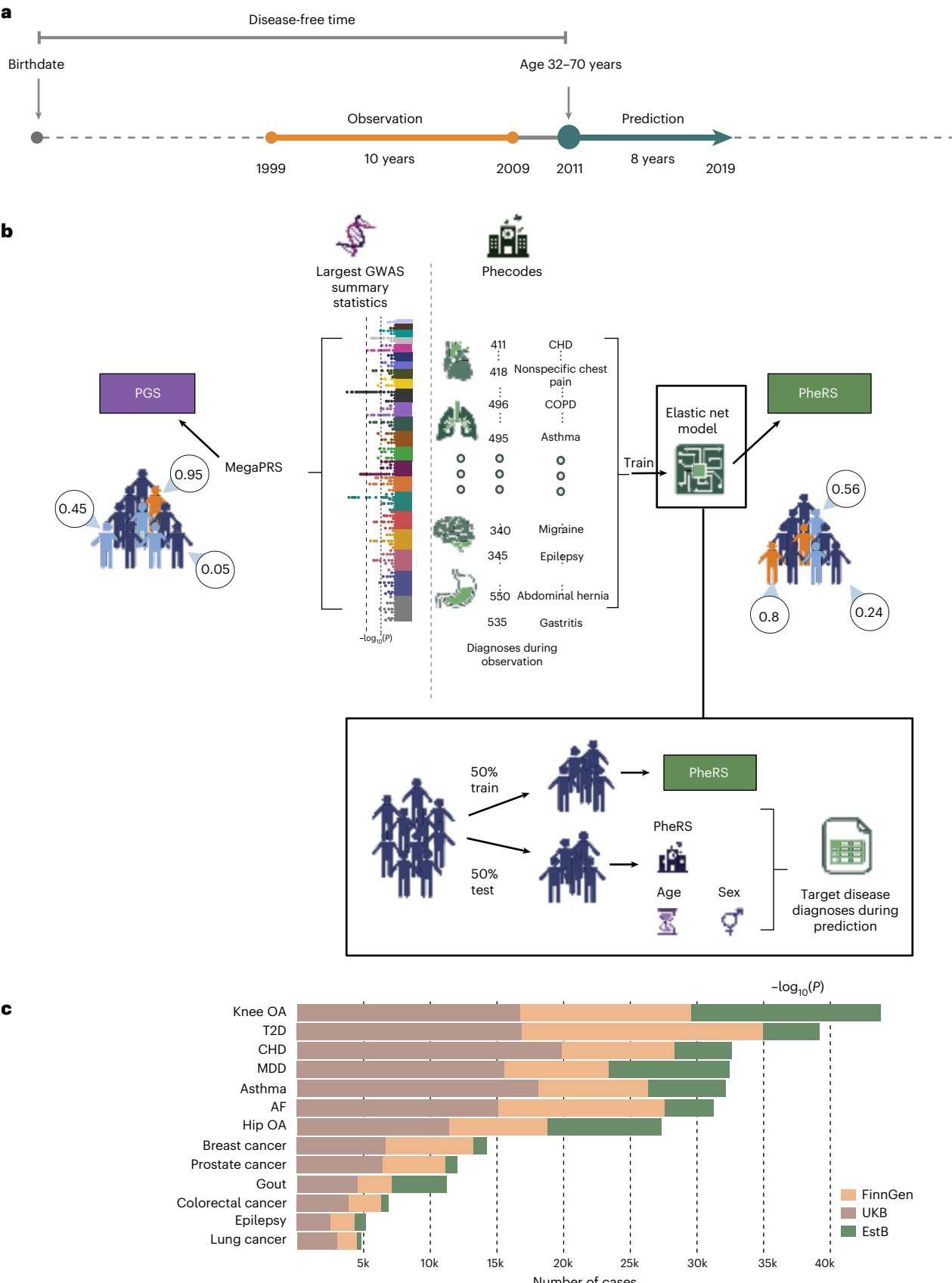

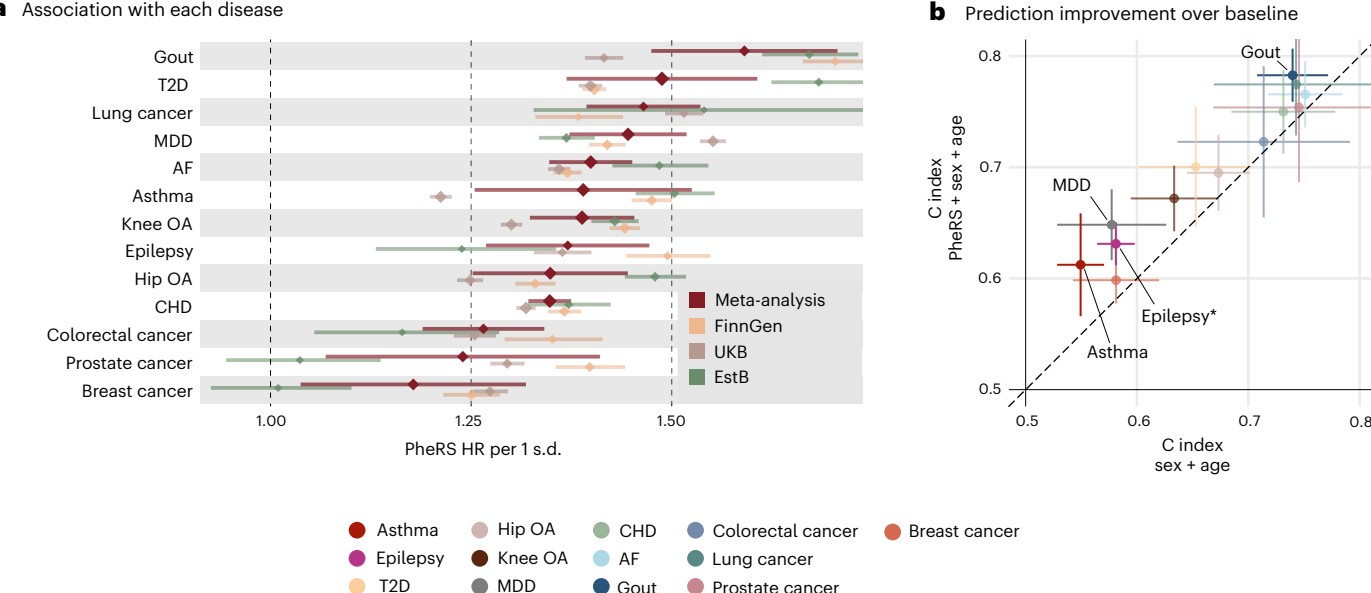

**a** Association with each disease

**b** Prediction improvement over baseline

**Fig. 2 | PheRS performance across studies. a**, Association between PheRS and disease onset during the prediction period, independent of age and sex. The HR point estimates and 95% CIs in each study−FinnGen, peach; UKB, brown; EstB, green; and meta-analyzed results, red. The HRs are shown for an increase of the PheRS by 1 s.d. after regressing out age and sex. **b**, Increase in predictive accuracy when adding the PheRS to a baseline model with age and sex. The meta-analyzed c-index point estimates and 95% CIs of the baseline models (*x* axis) compared to models with added PheRS (*y* axis). Diseases with significant differences are labeled (one-tailed $P < 0.05$ based on the *z* scores of the c index increases), and those passing Bonferroni correction for multiple hypothesis testing ($P < 0.05/13$) are marked with an asterisk. *P* values of the significant c index increase−gout, $1.7 \times 10^{-2}$; MDD, $8.6 \times 10^{-3}$; asthma, $7.6 \times 10^{-3}$; and epilepsy, $6.5 \times 10^{-5}$. All *P* values are listed in Supplementary Table 7, and the number of cases and controls for each disease in Supplementary Table 2.

The improvements persisted for four of these diseases (asthma, MDD, T2D and knee OA) in all of the three studies when compared to a baseline including, additionally, the highest achieved education level and the Charlson comorbidity index[37,38] (CCI; Supplementary Fig. 1a). Overall, integrating education and CCI only led to minor improvements in the model's discriminative ability compared to a baseline with age and sex (Extended Data Fig. 2).

We found that all PheRS were correlated, mostly positively, with the total number of phecodes an individual had recorded (Pearsons' *r* ranging from 0.82 (for asthma in FinnGen) to −0.66 (for breast cancer in FinnGen); Extended Data Fig. 1c). While the two predictors were strongly correlated, we found that the PheRS improved predictions also over an extended baseline model accounting for the number of unique diagnoses for 9 of 13, 11 of 13 and 4 of 13 diseases in FinnGen, UKB and EstB, respectively (Extended Data Fig. 1b) and that the magnitude of the PheRS HRs was only slightly reduced for 1 of 13 diseases in FinnGen, 0 of 13 in the UKB and 5 of 13 in the EstB (Extended Data Fig. 3b). To further test whether this meant that the PheRS were more predictive in older individuals who have had more time to accumulate diagnoses in their EHR, we stratified the FinnGen test set to a younger group aged 32–51 years and an older group aged 52–70 years. However, unexpectedly, we found a substantially stronger association of the PheRS in the younger age group for 4 of 13 diseases, and only for breast cancer was the relative risk in the older group substantially larger than for the younger group, while no differences were observed in the remaining diseases (Extended Data Fig. 4).

## PheRS generalize well between studies

We examined PheRS generalizability by comparing externally- and internally-trained PheRS. All PheRS models for a given disease share the same phecode predictors. When a phecode was not observed in a study but had a nonzero coefficient in a model trained in an external study, the coefficient was multiplied by zero and did not affect the prediction. Externally-trained PheRS were trained on the training

set of the UKB and EstB study and tested on the same test set as the FinnGen and UKB internally-trained PheRS. Externally-trained PheRS were moderately to strongly correlated with internally-trained PheRS (average Pearson's *r* = 0.43, range = −0.05 to 0.76; Fig. 3a and Supplementary Table 9). Not surprisingly, PheRS that were poor predictors of the disease were also poorly correlated between their internally-trained and externally-trained versions (that is, colorectal and breast cancer). Most externally-trained PheRS were substantially associated with disease risk in FinnGen (Fig. 3b, left and middle) and showed substantial improvements in c index over age and sex (Extended Data Fig. 5). In the EstB, the HRs for hip OA, gout and knee OA were not significantly lower when trained in the UKB compared to the PheRS models trained in the EstB (Fig. 3b, right). Furthermore, for the PheRS models for hip OA transferred to FinnGen and the UKB, as well as knee OA, AF, gout and prostate cancer PheRS trained in the EstB and transferred to the UKB, c-index improvements were not significantly different from those achieved by the internally-trained PheRS (Extended Data Fig. 5). Nonetheless, we observed that most PheRS disease associations were significantly lower with the externally-trained PheRS (Fig. 3b).

## Phecode importance varies across studies

Despite good PheRS generalizability, we found marked differences in the prevalence of different phecodes between the studies (Supplementary Table 11). While we found 527 different three-digit phecodes recorded in all three studies for at least five individuals, only 234 had a prevalence of at least 1% in any of the studies and only 48 were common across all studies (Fig. 4a). These differences can be partially explained by different types of diagnostic information from the EHR available in each study. For example, the inclusion of primary care diagnoses in the EstB study leads to a higher number of phecodes, with 33% (*n* = 77) unique to that study (Fig. 4a). The FinnGen and UKB studies, on the other hand, only used diagnoses from secondary care. Notably, we found that training PheRS only using codes common

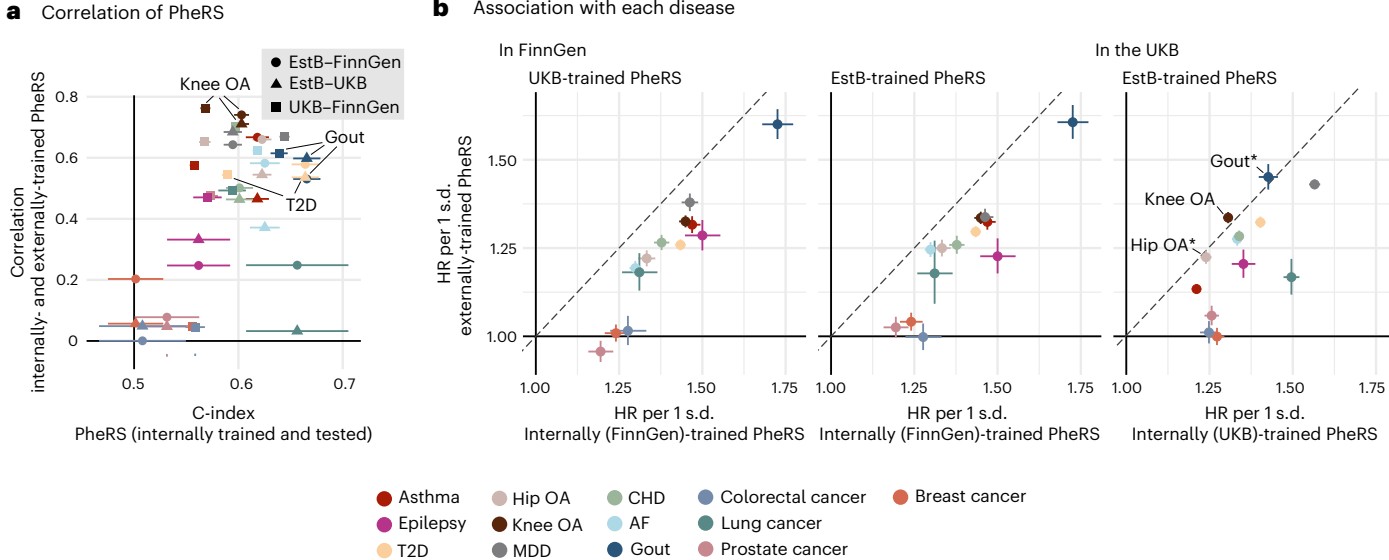

**Fig. 3 | PheRS external validation. a**, Correlations (Persons' *r*) between the internally- and externally-trained PheRS. Partial correlation point estimates and 95% CIs of the PheRS (*y* axis) after regressing out the effect of age and sex, with the c-index point estimates and 95% CIs of the internally-trained and internally-tested PheRS as references (*x* axis). All PheRS were trained on 50% of individuals. The *P* values of the correlations are listed in Supplementary Table 10. **b**, Associations of the internally-trained PheRS with each disease compared to the externally-trained models. HR point estimates and 95% CIs of the internally-trained PheRS (*x* axis) versus the externally-trained PheRS (*y* axis). Left and middle, the models

transferred to FinnGen, with EstB on the middle and UKB on the left. Right, the PheRS trained in the EstB and tested in the UKB. The HRs are shown for a 1 s.d. increase of the PheRS after regressing out age and sex. Diseases with no significant differences (two-tailed *P* ≥ 0.05 of the *z* scores of the HR differences) are marked with an asterisk, and those not passing Bonferroni correction for multiple hypothesis testing (*P* ≥ 0.05/13) are labeled. All *P* values are listed in Supplementary Tables 5 and 7 and the number of cases and controls for each disease in Supplementary Table 2.

(prevalence of at least 1%) to all three studies increased PheRS generalizability from EstB to FinnGen (Extended Data Fig. 6).

The set of phecodes unique to each study included important predictors for many of the diseases. To highlight one example, in each study, we found neuralgia (code 766) to be among the top 20 predictors for hip OA, CHD and MDD in the EstB. In FinnGen, schizophrenia (code 295) was an important predictor in T2D, lung cancer and epilepsy; and in the UKB, tobacco use disorder (code 318) was among the most important predictors for T2D, lung cancer, CHD and MDD. However, other predictors such as hypertension (code 401), overweight (code 278), alcohol abuse (code 317) and peripheral nerve disorders (code 351) were prevalent diagnoses in all three studies and showed a large consistent effect across diseases (Fig. 4b,c and Supplementary Tables 12 and 13).

We took a closer look at the top predictors in the individual PheRS models. Fig. 4d shows the shared and study-specific predictors in the PheRS models for MDD.

We found that the top predictors in each PheRS captured the following three main categories: substance abuse, sleep disorders and pain-related problems. The most consistent phecode related to substance abuse in all three studies was alcohol abuse (code 317; FinnGen rank 3, UKB rank 2 and EstB rank 4), while other diagnoses, such as tobacco use disorder (code 318) were only captured in the UKB study (rank 4). The most important predictors related to pain disorders in FinnGen were intervertebral disc disorders (code 722, rank 4) and migraine (code 340, rank 5), while in the UKB it was back pain (code 760, rank 7) and in the EstB peripheral nerve disorders (code 351, rank 9) and other headache syndromes (code 229, rank 10). Nevertheless, while the list of most important predictors varied, each of the PheRS models also captured other pain-related diagnoses with lower ranks (Supplementary Table 12). An additional leave-one-out analysis (LOO) performed in FinnGen (Extended Data Fig. 7) showed that removing most top predictors only minimally reduced the models' performances, underscoring the shared contribution of correlated features to the

overall performance. Extended Data Fig. 8 shows, for six additional diseases, how common and study-specific phecodes contributed to PheRS prediction.

### PGS and PheRS are orthogonal predictors

Finally, we compared the PheRS and corresponding PGS associations. Both were significantly associated with all diseases in the meta-analysis (*P* < 0.05). However, the magnitude of the associations varied across diseases. For 4 of 13 diseases (epilepsy, MDD, knee OA and lung cancer), the PheRS showed a stronger association with the diseases than the PGS, and for 4 of 13 there were no substantial differences (Extended Data Figs. 9a-1 and 10a). However, when looking at the top 10% of most at-risk individuals compared to the 20% at average risk (40–60% percentile), the PheRS captured the risk better for 8 of 13 diseases (T2D, gout, lung cancer, asthma, MDD, epilepsy, hip OA and knee OA; Fig. 5a). Moreover, PheRS provided additional information on top of PGS. Adding PheRS to a model with PGS, age, sex, and the first ten PCs, significantly increased the c index for 9 of 13 diseases in FinnGen, 2 of 4 diseases in the UKB, and 6 of 13 diseases in the EstB (Fig. 5c). Similarly, adding the PGS to a model with PheRS, age, and sex and PCs led to significant improvements for 10 of 13 in FinnGen, 3 of 4 in the UKB and 6 of 13 in the EstB (Extended Data Fig. 9c). The number of diseases with significant improvements due to adding the PheRS was similar to that achieved when adding the PheRS to age and sex (Extended Data Fig. 1a; see Methods and Supplementary Methods and Supplementary Results for more details)

Overall, we found that the EHR data and genetic information capture largely orthogonal information as shown by the low correlation between the two scores (average Pearsons' *r* = 0.02, range = 0.00–0.08; Supplementary Table 9), which was not just driven by the low predictiveness of the PGS (Fig. 5b and Extended Data Fig. 9b shows the correlations in relation to Nagelkerke's pseudo-*R*²). Furthermore, when adding the PGS to the models, the HRs of the PheRS did not change

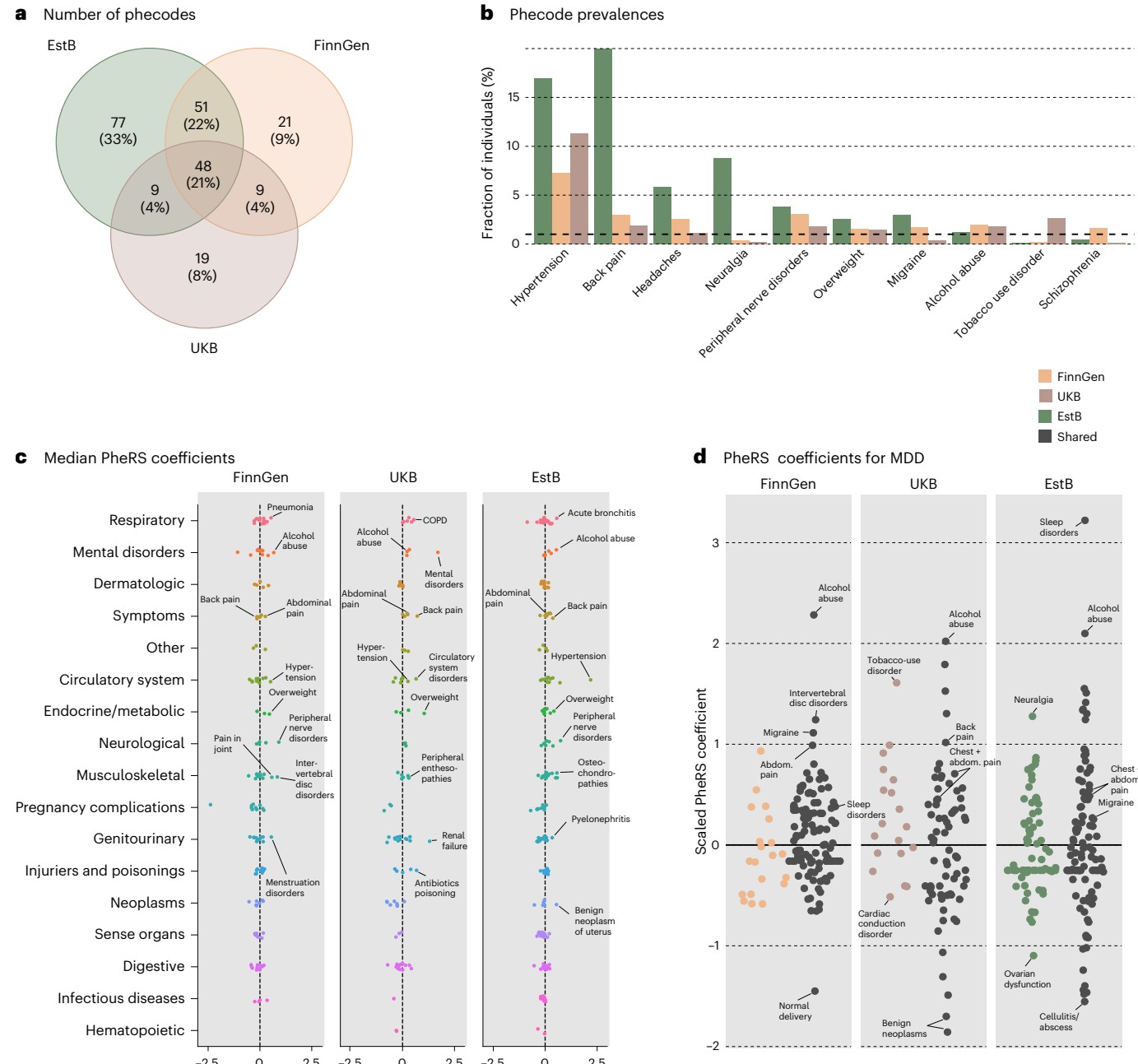

**Fig. 4 | Phecode prevalence and coefficients in each study. a**, A Venn diagram showing the number of phecodes present in each of the three studies and shared between all combinations of the three studies. We only considered phecodes with a prevalence of ≥1% in each study. Peach color indicates FinnGen-specific codes, brown indicates UKB-specific codes and green indicates EstB-specific codes. The same color coding applies to panels **b** and **d**. **b**, Phecode prevalences for selected example codes in the three studies. The black dashed line indicates a prevalence of 1%. **c**, Median of PheRS coefficients over the 13 diseases in each study. Only coefficients used by at least 7 of 13 models in the studies are shown (see Methods for phecode exclusion rules in the PheRS models). Different colors

and the *y*-axis labels indicate different phecode categories. Black dashed lines correspond to a coefficient value of 0. **d**, A detailed look at all the PheRS coefficients for MDD in the three studies. Black color marks common phecodes in the MDD PheRS models across the studies, while other colors indicate biobank-specific codes (peach, FinnGen; brown, UKB; and green, EstB). PheRS coefficients were standardized to 0 mean and 1 s.d. for each model separately for easier comparison of coefficient importance across the studies. Please note that this normalization will force approximately half of the coefficients to have a negative scaled value with respect to the mean at 0. Abdom., abdominal.

substantially and vice versa for the PGS HRs, when adding the PheRS (Fig. 5a, right, and Extended Data Fig. 9a-2).

## Discussion

In this study, we investigated the accuracy and generalizability of EHR-based models (PheRS) in predicting the 8-year risk for 13 common

diseases in three large biobank-based studies (FinnGen, EstB and UKB) compared to PGS. Our results highlight the complementarity of PheRS and PGS for a range of diseases, suggesting that combining EHR and genetic data can be an advantageous strategy for the prediction of many common diseases. Both PheRS and PGS were derived to be independent from age and sex effects, thus providing orthogonal information to

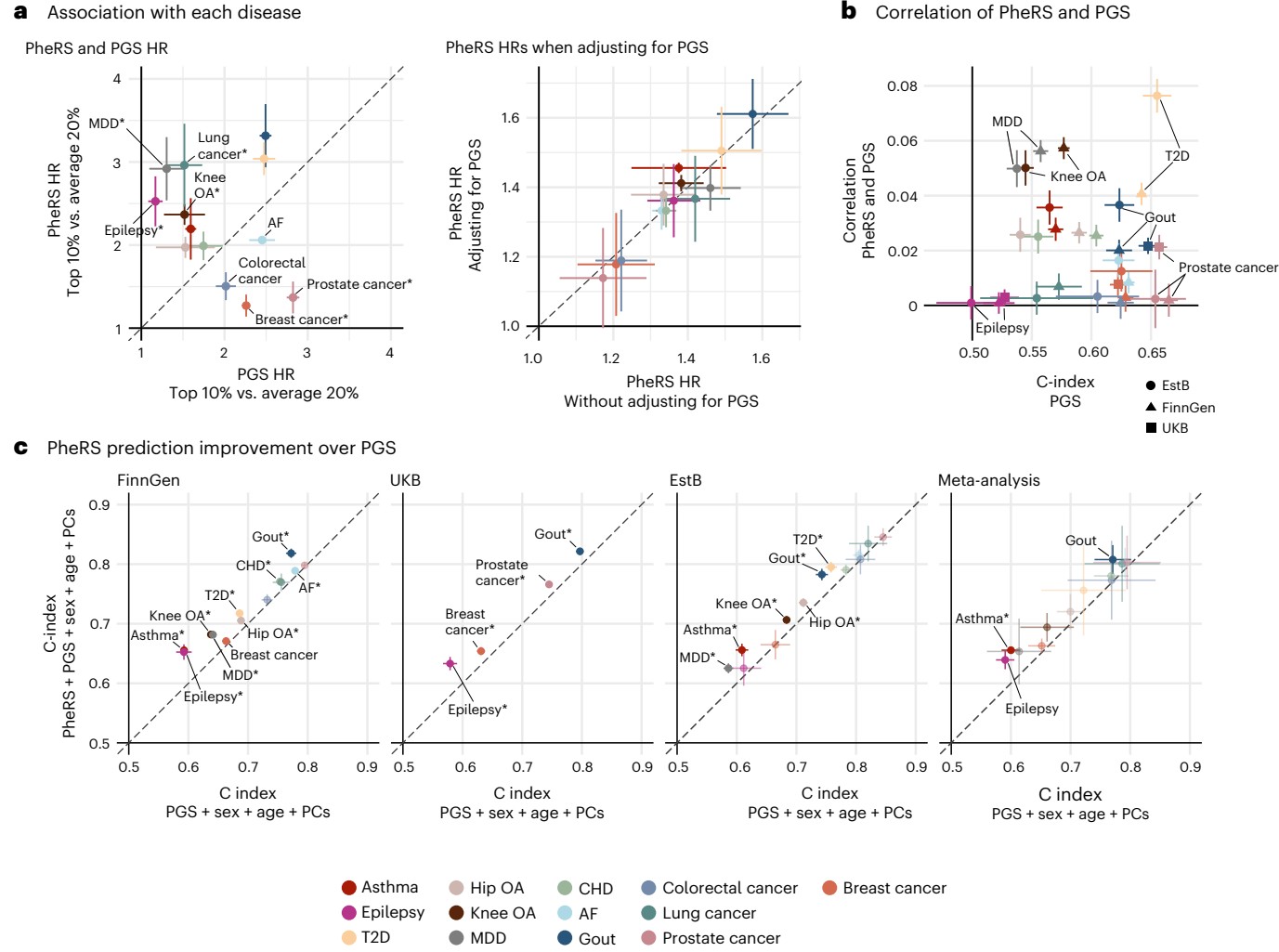

**Fig. 5 | Comparison of PGS and PheRS. a**, Left, associations of PGS (x axis) and PheRS (y axis) scores with each disease. vs., versus. The meta-analyzed HR point estimates (95% CI) for the top 10% at risk compared to those in the average 20% risk group (40–60% percentile) based on the scores after regressing out age, sex and the first ten PCs. Right, PheRS HR point estimates per 1 s.d. increase in models without PGS (x axis) and ones where we adjust for the effects of PGS (y axis). **b**, Correlation point estimates (Persons' r, y axis) and 95% CIs between the PheRS and PGS scores separately in each study (FinnGen, triangle; UKB, square; and EstB, circle), with the c-index point estimates of the PGS models as a reference after regressing out the effect of age, sex and the first ten PCs (x axis). The P values of the correlations are listed in Supplementary Table 9.

**c**, C-index improvements when adding PheRS to models with PGS. C-index point estimates and 95% CIs of the models with only PGS, sex, age and PCs (x axis) compared to those with added PheRS (y axis). Due to sample overlap with the GWASs, PGS could only be calculated for four diseases in UKB (see Methods for details). Diseases with significant differences are labeled (P < 0.05; two-tailed P values based on the z scores of the HR differences and one-tailed P values for the c-index increases) and those passing Bonferroni correction of multiple hypothesis testing (P < 0.05/13) are marked with an asterisk. All P values are listed in Supplementary Tables 5 and 7 and the number of cases and controls for each disease in Supplementary Table 2.

these two key risk factors. Furthermore, we were able to successfully externally validate the models trained in EstB and UKB, suggesting that the PheRS models capture relevant risk factors that are not only study or healthcare system specific.

While the performance of the PheRS models varied between diseases, the PheRS for asthma, MDD, T2D, knee OA and gout, in particular, showed consistent improvement over baseline across all three studies. For asthma, knee OA and gout, the PheRS were less predictive in the UKB compared to the other cohorts. This could be reflecting the richer health information available in FinnGen with longer register coverage, and in EstB with inclusion of primary care data, both allowing more accurate phenotyping and detection of diagnoses before the baseline date. Colorectal cancer, prostate cancer, and breast cancer PheRS models performed poorly across the studies, likely due to the low case counts for these diseases in our data. PheRS outperformed a baseline of age and sex for all 13 diseases across the studies (except for breast

and prostate cancers in the EstB), and substantially improved over counting the number of previous diagnoses for 9, 11 and 4 diseases in FinnGen, UKB and EstB, respectively, indicating that different existing diagnoses contribute to disease risk differently depending on the target disease predicted. We expected to see low generalizability of the PheRS between studies due to differences in clinical and disease coding practices in different countries and healthcare systems. Nonetheless, we found that the PheRS replicated well for many of the diseases; although, as expected, most PheRS trained within-study performed better. The good generalizability of PheRS was also surprising given the large variability in prevalence of phecodes we found across studies, with only 20% of them observed in all three studies. However, our results are in line with a few previous studies that show that it is possible to create predictors that are generalizable across healthcare systems[6–8,16].

Upon closer examination of the phecodes prevalent in each study and their importance in the PheRS models, we find that the PheRS

models that generalize well from the UKB or EstB to FinnGen use both study-specific phecodes and phecodes shared between the studies. In some cases, such as gout, a large part of the generalizability of the models could already be explained by a few major risk factors, such as hypertension, high BMI and diabetes that are consistent in all three studies[39]. In other cases, the relevance of each predictor was more intricate. For example, in UKB, one of the most important phecodes for MDD was tobacco use disorder, but this code had very low prevalence in FinnGen. Instead, we found that the phecode for alcohol abuse (code 317) was a prevalent and important predictor in all three studies. Both alcohol abuse and sleep disorders, another important and prevalent predictor in all the studies, are known complex comorbidities of MDD[40]. We hypothesize that many of the different phecodes captured a single underlying risk factor. For example, several different pain-related diagnoses were among the top predictors for MDD, each likely capturing underlying pain problems[41], with the top predictors differing between models. The elastic net penalty allows a nonzero coefficient for many correlating phecodes, which alleviates the issue of the same underlying medical issue being coded differently in different EHRs. This suggests that leveraging similarity between diagnostic codes is an essential aspect in creating generalizable EHR-based predictors.

We kept the PheRS approach simple to demonstrate its feasibility. More complex models could further exploit the longitudinal nature of EHR information and use other data modalities available in the EHR systems[5,15,42]. Further, by using a 2-year washout period and excluding very closely related conditions from the predictors, we remained conservative in removing comorbidities directly related to the disease. Without this buffer, the performance of the models will likely increase and be more relevant in a clinical context[6]. To improve the generalizability of the models, we collapsed phecodes into the first three digits to reduce the effect of different diagnostic codes being used in different countries to describe the same underlying phenomenon[43]. For example, in the EstB study, phecodes hypertensive heart disease (code 401.21) and essential hypertension (code 401.1) were equally prevalent diagnoses, capturing the risk factor hypertension (code 401), while in FinnGen hypertensive heart disease (code 401.21) had a prevalence of <1%. Other approaches could include mapping diagnostic codes to OMOP-concepts, which have been shown to facilitate EHR-based models that transfer between different countries[16]. Regardless, differences in the types of data available in different cohorts remain a challenge. For example, in our study, EstB was the only study with primary care information. Such fundamental differences would be best addressed by compiling comprehensive datasets. Even if all the studies included the same sources of health data, medical codes are used differently in different countries to describe the same underlying condition. This would require approaches that can map the meaning of the codes used, rather than the actual codes.

Notably, while we did not exclude individuals based on their genetic ancestry, the UKB still consists mainly, and FinnGen and the EstB almost exclusively, of individuals of European ancestry. Thus, our study does not properly assess the important issue that individuals of different ethnicities face inequalities in healthcare access[44,45]. Important open questions for future work, in addition to the generalizability of EHR-based scores for non-European genetic ancestries, include how to optimally model diagnostic codes for best generalizability as well as leveraging data from different and diverse cohorts with, for example, federated learning approaches[46].

To our knowledge, the correlation between PGS and EHR-based scores has not been comprehensively studied. For CAD, details in ref. 7 showed that the inclusion of PGS did not improve prediction compared to an EHR-based score, while details in ref. 6 showed that the inclusion of genetic information substantially improved models with both EHR-based predictors and the gold standard model for CAD risk prediction (American College of Cardiology/American Heart Association Pooled Cohort Risk Equations). For 8/13 diseases studied

here, we observe a substantial improvement in onset prediction when integrating PheRS on top of PGS. While for many of the cancers (colorectal cancer, prostate cancer, and breast cancer), the PGS was more informative, for diseases such as MDD, epilepsy and knee OA, the PheRS better captured the risk. Interestingly, PheRS was specifically better than PGS in capturing high-risk individuals. Individuals in the top 10% of PheRS had higher HRs than those in the top 10% of PGS for 8 of 13 diseases, probably reflecting those individuals with key comorbidities. However, while we used the same methodology for the percentile assignment for the PheRS and PGS, prior work on the PGS, as detailed in ref. 47, has shown that there can be large uncertainties in the assignment of an individual to a risk group. Similar comprehensive robustness analyses of PheRS percentile assignment are an interesting direction for future work. Overall, we observed very low correlation between PGS and PheRS, indicating that these two data sources contain largely independent information. A few prior studies on the interaction between PGS and selected risk factors found no evidence for interaction[40,48].

A patient's diagnostic history has always been a key piece of information for medical professionals when considering future treatment. As we move towards translating PGS for clinical use, it is worth considering a comprehensive integration of information about an individual's diagnosis history, which is often collected in a centralized electronic manner in many countries. This would not be a large shift from current practice, as selected comorbidities are used in many clinical risk stratification algorithms, for example, QRisk[49] for evaluating the risk of heart attack or stroke in the next 10 years, or QDiabetes for evaluating the 10-year risk of T2D[50]. A recent study showed that EHR-based models explicitly trained to predict the risk of five different cardiovascular events performed similarly or better than conventional risk scores (QRISK3, ASCVD and SCORE2)[16]. Similarly, details in ref. 6 show that machine learning models trained on longitudinal EHR data outperformed the gold standard risk model (American College of Cardiology/American Heart Association Pooled Cohort Risk Equations) for the prediction of cardiovascular disease. These comparisons are interesting for diseases with established risk scores. However, for many of the diseases studied here, there are no established risk algorithms, making an EHR-based risk stratification approach even more relevant.

In this study, we showed that across many diseases and multiple studies with different underlying healthcare systems and EHRs, relatively simple elastic net-based risk scores that consider an individual's previous diagnosis history can improve disease risk prediction when combined with PGS. Information already available from the EHR provides orthogonal information to PGS and could be a cost-effective approach for risk estimation.

## Online content

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

**Estonian Biobank research team**

**Reedik Mägi**[2]

**FinnGen**

**Samuli Ripatti**[1,3,4,5] **& Andrea Ganna**[1,3,4]

## Methods

### Ethics declarations

Patients and control participants in FinnGen provided informed consent for biobank research, based on the Finnish Biobank Act. Alternatively, separate research cohorts, collected before the Finnish Biobank Act came into effect (in September 2013) and the start of FinnGen (August 2017), were collected based on study-specific consents and later transferred to the Finnish Biobanks after approval by Fimea (Finnish Medicines Agency), the National Supervisory Authority for Welfare and Health. Recruitment protocols followed the biobank protocols approved by Fimea. The Coordinating Ethics Committee of the Hospital District of Helsinki and Uusimaa (HUS) approved the FinnGen study under ethics statement HUS/990/2017. The study is also approved by the Finnish Institute for Health and Welfare (permits THL/2031/6.02.00/2017, THL/1101/5.05.00/2017, THL/341/6.02.00/2018, THL/2222/6.02.00/2018, THL/283/6.02.00/2019, THL/1721/5.05.00/2019 and THL/1524/5.05.00/2020); the Digital and Population Data Services Agency (permits VRK43431/2017-3, VRK/6909/2018-3 and VRK/4415/2019-3); the Social Insurance Institution of Finland (permits KELA 58/522/2017, KELA 131/522/2018, KELA 70/522/2019, KELA 98/522/2019, KELA 134/522/2019, KELA 138/522/2019, KELA 2/522/2020 and KELA 16/522/2020); Findata (permits THL/2364/14.02/2020, THL/4055/14.06.00/2020, THL/3433/14.06.00/2020, THL/4432/14.06/2020, THL/5189/14.06/2020, THL/5894/14.06.00/2020, THL/6619/14.06.00/2020, THL/209/14.06.00/2021, THL/688/14.06.00/2021, THL/1284/14.06.00/2021, THL/1965/14.06.00/2021, THL/5546/14.02.00/2020, THL/2658/14.06.00/2021 and THL/4235/14.06.00/2021); Statistics Finland (permits TK-53-1041-17, TK/143/07.03.00/2020 (earlier TK-53-90-20), TK/1735/07.03.00/2021 and TK/3112/07.03.00/2021) and the Finnish Registry for Kidney Diseases (permission based on the meeting minutes dated 4 July 2019). The biobank access decisions for FinnGen samples and data used in FinnGen Data Freeze 10 include approvals from the following biobanks: THL Biobank (BB2017_55, BB2017_111, BB2018_19, BB_2018_34, BB_2018_67, BB2018_71, BB2019_7, BB2019_8, BB2019_26, BB2020_1 and BB2021_65); Finnish Red Cross Blood Service Biobank (7 December 2017); Helsinki Biobank (HUS/359/2017, HUS/248/2020, HUS/150/2022 §§12–18 and §23); Auria Biobank (AB17-5154 and amendment 1 (17 August 2020), amendments BB_2021-0140, BB_2021-0156 (26 August 2021, 2 February 2022), BB_2021-0169, BB_2021-0179, BB_2021-0161, AB20-5926 and amendment 1 (23 April 2020) with its modification (22 September 2021)); Biobank Borealis of Northern Finland (2017_1013, 2021_5010, 2021_5018, 2021_5015, 2021_5023, 2021_5017 and 2022_6001); Biobank of Eastern Finland (1186/2018 and amendments §§22/2020, 53/2021, 13/2022, 14/2022 and 15/2022); Finnish Clinical Biobank Tampere (MH0004 and amendments (21 February 2020 and 6 October 2020), §§8/2021, 9/2022, 10/2022, 12/2022, 20/2022, 21/2022, 22/2022 and 23/2022); Central Finland Biobank (1-2017); Terveystalo Biobank (STB 2018001 and amendment dated 25 August 2020); Finnish Hematological Registry and Clinical Biobank (decision dated 18 June 2021) and Arctic Biobank (P0844: ARC_2021_1001).

Ethics approval for the UK Biobank study was obtained from the North West Centre for Research Ethics Committee (11/NW/0382). The UK Biobank data used in this study were obtained under approved application 78537.

The activities of the EstBB are regulated by the Human Genes Research Act, which was adopted in 2000 specifically for the operations of the EstBB. Individual-level data analysis in the EstBB was carried out under ethical approval 1.1-12/624 from the Estonian Committee on Bioethics and Human Research (Estonian Ministry of Social Affairs), using data according to release application S22, document 6-7/GI/16259 from the EstBB.

### Study setup

As outlined in Fig. 1b, each study consisted of a 10-year observation (6 years for EstBB due to shorter follow-up) and an 8-year prediction period, separated by a 2-year washout period. Each disease's case and control definitions were based on diagnoses acquired in the 8-year prediction period (from 1 January 2011 to after 1 January 2019). The International Classification of Diseases (ICD) codes used to define the cases for each disease were based on previous harmonization between FinnGen and the EstBB phenotypes by the INTERVENE consortium[34] (Supplementary Table 14). We consider all individuals as controls who were not cases. We only considered adults aged 32–70 in 1 January 2011 and removed all individuals diagnosed with the disease before this time. The lower limit for age of inclusion was chosen due to the inclusion of education level in some of the models and was determined based on the median age of obtaining a doctoral degree in the FinnGen dataset. Using this lower limit, most individuals included have finished their highest level of education. Furthermore, we removed all individuals with a diagnosis outside the prediction period (from 1 January 2011 to after 1 January 2019) and those lost to follow-up before the start of the prediction period. The ICD-codes used to define the cases for each disease and the number of cases and controls in each study are listed in Supplementary Tables 2 and 14.

We included 845,929 individuals (Supplementary Table 1) from three biobank-based studies—FinnGen[29], UKB[28] and EstB[30] linked with national registers or EHRs. In FinnGen, we used Data Freeze 10, which includes 412,090 individuals, of whom 266,179 were aged 32–70 years in 1 January 2011. The longitudinal ICD-code diagnoses used to define the phecodes and the case and control status for each disease were based on in- and outpatient hospital register information. The UKB study included 464,076 individuals aged 40–70 years, with the ICD-code diagnoses based on inpatient information. The EstB study included 199,868 individuals, of whom 115,674 were aged 32–70 years. Here we also had primary care data as well as self-reported diagnoses available. More details on the phenotype harmonization can be found in ref. 34 and the Supplementary Methods.

### Predictors

**PGS.** The PGS were previously computed by the INTERVENE consortium[34] and based on the recent publicly available GWAS summary statistics, with minimal overlap with our study cohorts (Supplementary Table 15) using MegaPRS[51] with the BLD-LDAK heritability model. For the Cox-PH models, we removed individuals from the studies that were part of the GWAS on which the PGS were based. Due to the large overlap with the UKB individuals, we only had PGS for gout, epilepsy, breast and prostate cancer available in the UKB.

**PheRS.** For the EHR-based models, we trained elastic net models[32] on ICD-9 and ICD-10 diagnoses mapped to phecodes. The phecode mapping was based on v1.2b1 of the phecode map[33,35] from https://phewascatalog.org/, with some manual additions. Since we only considered diagnoses during the observation period starting in 1999, all diagnoses were ICD-10-based in our data. To obtain the most comprehensive mapping, we removed all special characters from the ICD codes. If a match could not be found in the phecode map, we shortened the code by one digit until it could be mapped or was removed. The complete mapping used can be found in Supplementary Table 16. We gathered all phecodes in their three-digit parent node in the phecode ontology; for example, type 1 diabetes (250.1), T2D (250.2), and T2D with ketoacidosis (250.21) were all mapped to the same phecode diabetes mellitus (250). For each disease, we separately excluded predictors that were part of the exclusion range of the phecodes (Supplementary Table 3), for example, for T2D, we did not use secondary diabetes (phecode 249), diabetes mellitus (250) and conditions complicating pregnancy (649) as predictors. The phecode conditions complicating pregnancy was excluded

because it was the parent node of the phecode Diabetes or abnormal glucose tolerance complicating pregnancy (649.1), which is in the exclusion range of the phecode for T2D (250.2). We only considered phecodes with a prevalence of at least 1% of the study population (Supplementary Table 11).

We implemented the PheRS using the LogisticRegression function from scikit-learn (version 1.3.2)[52]. We included age (at the start of the prediction period 1 January 2011) and sex as predictors in the PheRS models because they are important predictors, and otherwise the models would reconstruct predictors for age and sex using combinations of the phecode diagnoses, which would make interpretation of the phecode coefficient values challenging. Nonetheless, the effect of age and sex was then regressed out when evaluating the performances of the PheRS (see below). Models were penalized with the elastic net penalty. Predictors were coded as 1/0, where 1 = 'predictor observed during the observation window' and 0 = 'predictor not observed during the observation window', for each disease separately. For training, 50% of the data was used, and this was further divided into training (85%) and hold-out test (15%) sets. Sizes of the training datasets are shown for each disease and study in Supplementary Table 22. L1 to L2 ratio hyperparameter of the elastic net models was optimized using grid search and fivefold cross-validation over the range 0.05–0.95 (step size = 0.05), simultaneously with inverse of the regularization strength (C) over the following possible values: $1 \times 10^{-5}$, $5 \times 10^{-5}$, $1 \times 10^{-4}$, $5 \times 10^{-4}$, $1 \times 10^{-3}$, $5 \times 10^{-3}$, $1 \times 10^{-2}$, $5 \times 10^{-2}$, $1 \times 10^{-1}$, $5 \times 10^{-1}$, 1. Balanced class weights were used, based on class frequencies in the training data. The LOO analysis assessing the impact of the removal of individual phecodes to PheRS performance was performed using a ridge penalty instead of elastic net. This was done to cut running time substantially, as using ridge removes the L1 to L2 ratio hyperparameter and its optimization. Otherwise, the ridge models were fitted similarly to the elastic net models. Before running the LOO analysis, we tested that switching to ridge did not generally reduce the PheRS performance in FinnGen (Extended Data Fig. 7a).

Model fitting was done using stochastic average gradient descent. The best L1 to L2 ratio was selected based on the average precision score using 5-fold cross-validation on the training split. Missing values of predictors were imputed to the mean of the corresponding predictor in the study-specific training data, and all predictors were standardized to zero mean and unit variance on the study-specific training data before model fitting.

The PheRS models trained within the UKB or the EstB data on 50% of individuals were used to make predictions in FinnGen and UKB test sets, as is without any retraining within the studies. Standardization and imputation were performed based on the biobank-specific training data, meaning that, for example, when assessing the performance of the UKB-trained model in FinnGen, the FinnGen test set data were imputed and standardized based on the feature-specific means and s.d. from the UKB.

### Cox-PH models
Ultimately, each individual was assigned 13 different PGS and PheRS scores describing their risk of getting a disease diagnosis in the prediction period based on genetic or EHR-based information. To make the PheRS and PGS comparable, we regressed out the effect of age, sex and the first ten genetic PCs from all continuous scores using the residuals from a logistic regression with the score as outcome. When only considering PheRS performance, we regressed out only age and sex. Subsequently, we scaled all predictors to have a mean of zero and a s.d. of 1. We then used these scores in separate Cox-PH analyses, with survival time defined as the period from 2011 until diagnosis, censoring (end of follow-up), or the end of the prediction period.

Additionally, we considered the CCI (Supplementary Methods)[37,38]—developed to account for the individual's overall comorbidity burden—and the individual's highest achieved education level

in 2011 as an indicator of their socioeconomic status. For the CCI, we compared the top 10% of individuals with the highest CCI to the rest. The high-risk group included individuals with a CCI ≥ 2 and a few younger ones with a CCI of 1. For the highest education level, we mapped each study's education coding to the 2011 International Standard Classification of Education (ISCED-11; Supplementary Table 17) codes. We compared the risk of individuals with basic education (ISCED-11: 1–4) to those who achieved higher education levels (ISCED-11: 5–7).

### Statistics
We used the survival[53] package (version 3.2-7) in R for creating the Cox-PH models and the Hmisc[54] package (version 5.1.0) to calculate the c indices and 95% CIs. For a Cox-PH model with binary outcomes, the predicted survival times can be shown to be equal to the survival probability, so the c index is equivalent to the area under the receiver operating characteristic curve (AUC)[55,56]. The meta-analysis of the HRs and c indices was performed using the metafor[57,58] package (version 4.6-0) in R with a random effects model. We used two-tailed P values, calculated using the pnorm function in the stats package (version 3.6.2) in R, based on the z scores of the β differences to compare the differences in HR magnitudes and one-tailed P values for the statistical testing of increases in the c indices. Additionally, we used Bonferroni correction to account for multiple hypothesis testing in each study (n = 13). Correlations were calculated, using the cor.test function from the stats package in R. For regressing out the covariates—age, sex and PCs—from the PheRS and PGS, we used scaled residuals from glm models with the stats package in R.

### Comparison of phecode coefficients between different PheRS models
The elastic net hyperparameters were separately optimized for each PheRS model. This means that the absolute magnitudes of the coefficients for phecodes are not comparable between different PheRS. However, the relative importances of phecodes can still be compared, that is, whether, for example, the same phecodes are among the most important predictors in two different PheRS. To make visualization of the phecode importances in different PheRS clearer, we standardized the coefficients of each PheRS separately to a mean of 0 and a s.d. of 1 for the display items. Further, in each study, we ranked the phecodes in descending order by the PheRS coefficient values and assigned them ascending ranks. Thus, a lower rank indicates a higher PheRS coefficient in the model. Both the unscaled PheRS coefficients and ranks are Supplementary Table 12.

### Reporting summary
Further information on research design is available in the Nature Portfolio Reporting Summary linked to this article.

### Data availability
The individual-level data in these studies are protected for data privacy, and access is regulated through the biobanks. The Finnish biobank data can be accessed through the Fingenious services (https://site.fingenious.fi/en/) managed by FINBB. Researchers interested in EstBB can request access at https://genomics.ut.ee/en/content/estonian-biobank#dataaccess and the UKB data are available through a procedure described at https://www.ukbiobank.ac.uk/use-our-data/apply-for-access/. The GWAS data used in this study are available in the GWAS catalog database (https://www.ebi.ac.uk/gwas/) under accession codes listed in Supplementary Table 15. The PGS scores generated in this study are available in the PGS Catalog (https://www.pgscatalog.org/) under publication ID PGP000618 and score IDs PGS004869–PGS004886. The mapping between ICD and phecodes used in this study is available at https://phewascatalog.org. Source data are provided with this paper.

## Code availability

The code for PheRS model training is available at https://github.com/intervene-EU-H2020/INTERVENE_PheRS and https://zenodo.org/records/15691460 (ref. 59) and for the Cox-PH models as well as the final analysis of results at https://github.com/intervene-EU-H2020/onset_prediction and https://zenodo.org/records/15681882 (ref. 60).

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

## Acknowledgements

We would like to acknowledge the participants and investigators of the FinnGen, UK Biobank and the Estonian Biobank studies.

This study has received funding from the European Union's Horizon 2020 research and innovation program (grant agreement 101016775). This Estonian Biobank study was funded by the European Union through the European Regional Development Fund (project 2014-2020.4.01.15-0012 GENTRANSMED). A.G. has received funding from the European Union's Horizon 2020 research and innovation program (grant 101016775), the European Research Council under the same program (grant 945733) and the Academy of Finland (fellowship grant 323116).

We would like to acknowledge the participants and investigators of the FinnGen study listed in Supplementary Table 19. The FinnGen project is funded by two grants from Business Finland (HUS 4685/31/2016 and UH 4386/31/2016) and the following industry partners: AbbVie, AstraZeneca UK, Biogen MA, Bristol Myers Squibb (and Celgene Corporation & Celgene International II Sàrl), Genentech, Merck Sharp & Dohme, Pfizer, GlaxoSmithKline Intellectual Property Development, Sanofi US Services, Maze Therapeutics, Janssen Biotech, Novartis AG and Boehringer Ingelheim International GmbH. Following biobanks are acknowledged for delivering biobank samples to FinnGen: Auria Biobank (www.auria.fi/biopankki), THL Biobank (www.thl.fi/biobank), Helsinki Biobank (www.helsinginbiopankki.fi), Biobank Borealis of Northern Finland (https://www.ppshp.fi/Tutkimus-ja-opetus/Biopankki/Pages/Biobank-Borealis-briefly-in-English.aspx), Finnish Clinical Biobank Tampere (www.tays.fi/en-US/Research_and_development/Finnish_Clinical_Biobank_Tampere), Biobank of Eastern Finland (www.ita-suomenbiopankki.fi/en), Central Finland Biobank (www.ksshp.fi/fi-FI/Potilaalle/Biopankki), Finnish Red Cross Blood Service Biobank (www.veripalvelu.fi/verenluovutus/biopankkitoiminta), Terveystalo Biobank (www.terveystalo.com/fi/Yritystietoa/Terveystalo-Biopankki/Biopankki/) and Arctic Biobank (https://www.oulu.fi/en/university/faculties-and-units/faculty-medicine/northern-finland-birth-cohorts-and-arctic-biobank). All Finnish Biobanks are members of BBMRI.fi infrastructure (https://www.bbmri-eric.eu/national-nodes/finland/). Finnish Biobank Cooperative—FINBB (https://finbb.fi/) is the coordinator of BBMRI-ERIC operations in Finland.

The authors would like to thank participants and scientists involved in making the UK Biobank resource available at https://www.ukbiobank.ac.uk/. This research has been conducted using the UKB resource under approved application 78537.

The EstBB research team received funding from the Estonian Research Council (grant TT17 'Estonian Centre for Genomics'). Data analysis was carried out in part in the High-Performance Computing Center of the University of Tartu. K.L. and R.M. received funding from the Estonian Research Council grants PUT (PRG1911) and TK (TK214).

We would like to acknowledge CSC—IT Center for Science, Finland, for computational resources.

The funders had no role in study design, data collection and analysis, decision to publish or preparation of the manuscript.

## Author contributions

K.E.D., T.H. and A.G. wrote the manuscript with substantial input from all the other authors. T.H. and K.E.D. developed the PheRS code, and K.E.D. wrote the Cox model code. B.J. contributed the PGS. K.E.D. and T.H. conducted the analyses in FinnGen; T.H. and K.E.D. in the UK Biobank; and M.T.-L. and K.L. in the Estonian Biobank. K.E.D. combined the results and created the figures together with T.H. K.E.D., T.H. and Z.Y. preprocessed the FinnGen and UK Biobank data; M.T.-L. and K.L. preprocessed the Estonian Biobank data. T.H., B.J. and A.G. developed the original draft of the manuscript. A.G., S.R. and R.M. supervised the study.

## Funding

## Competing interests

A.G. is the founder of Real World Genetics Oy. B.J. became an employee of BioMarin after his part of this work was completed. K.L. has participated as an analyst in a collaborative research project at the Institute of Genomics, University of Tartu, which was funded by Geneto OÜ. The other authors declare no competing interests.

## Additional information

**Extended data** is available for this paper at https://doi.org/10.1038/s41588-025-02298-9.

**Correspondence and requests for materials** should be addressed to Andrea Ganna.

## a - prediction improvements with PheRS over baseline in each study

### 1 - age and sex

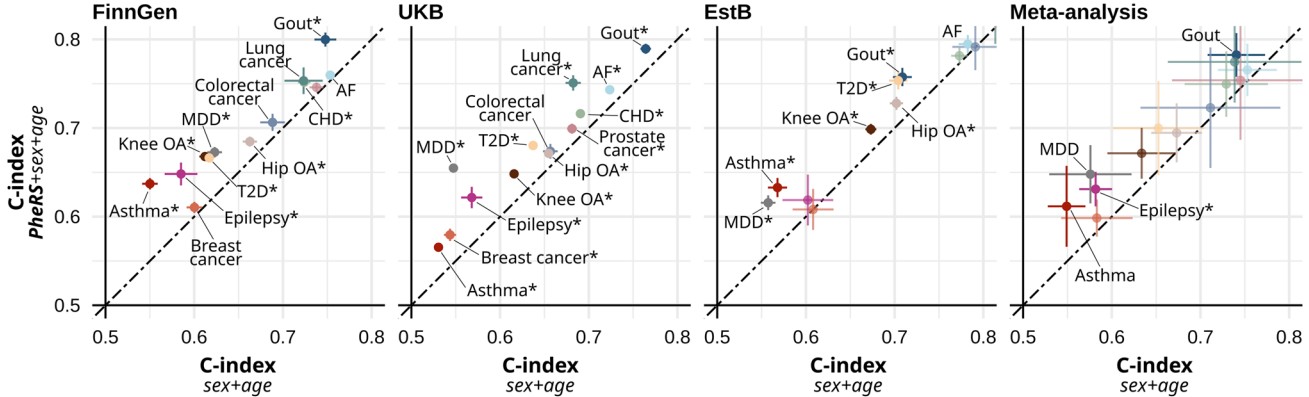

### 2 - age, sex, unique number of diagnoses

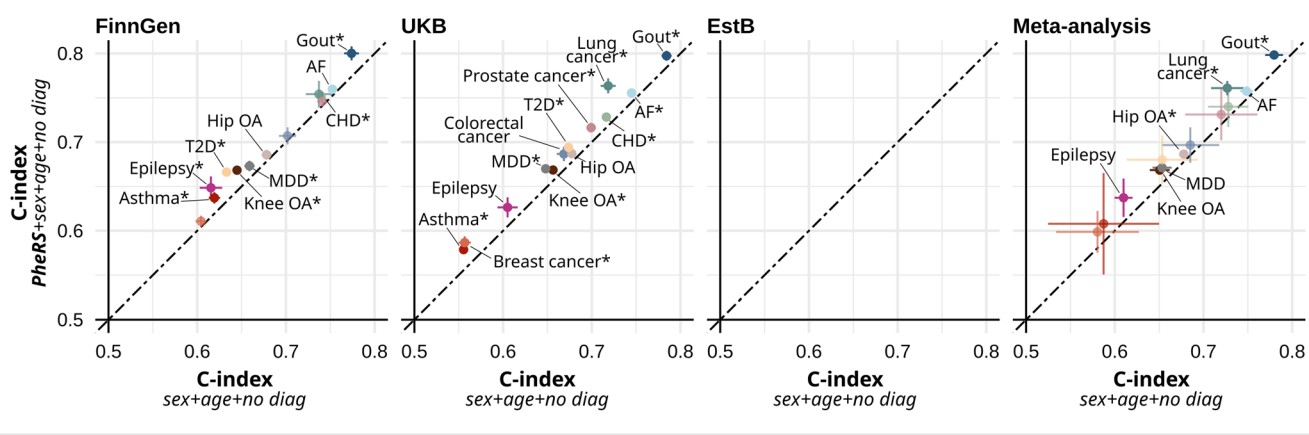

- ● Asthma
- ● Epilepsy
- ● Type 2 diabetes (T2D)
- ● Hip osteoarthritis (hip OA)
- ● Knee osteoarthritis (knee OA)
- ● Major depression (MDD)
- ● Coronary heart disease (CHD)
- ● Atrial fibrillation (AF)
- ● Gout
- ● Colorectal cancer
- ● Lung cancer
- ● Prostate cancer
- ● Breast cancer

## b - prediction accuracy of the PheRS

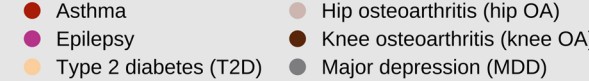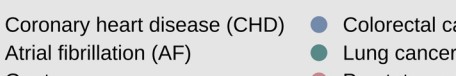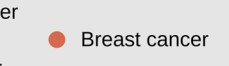

## c - correlation of PheRS with the number diagnoses

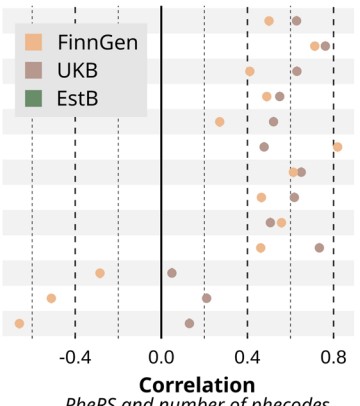

**Extended Data Fig. 1 | See next page for caption.**

**Extended Data Fig. 1 | Prediction accuracies of the PheRS. a**, (1) Increase in prediction accuracy when adding the PheRS to a baseline model with age and sex in all three studies. The c-index point estimates and 95% CIs of the baseline model (x-axis), compared to a model with added PheRS (y-axis). (2) Increase in prediction accuracy when adding the PheRS to a baseline model with the additional predictor of number of unique phecodes/diagnoses. Diseases with a significant difference are labeled (one-tailed p-values < 0.05 based on the z-scores of the c-index increases) and those passing Bonferroni correction for multiple hypothesis testing (p-value < 0.05/13) are marked with *. All p-values are listed in Supplementary Table 7 and the number of cases and controls for each disease in Supplementary Table 2. **b**, The c-index point estimates of the PheRS model and 95% CIs separately in each study (FinnGen: peach, UKB: brown, EstB: green and meta-analyzed results: red). The effect of age and sex was removed by regressing them out from the PheRS. **c**, Correlation between the PheRS and the total number of unique phecodes of each individual (FinnGen: peach, UKB: brown, EstB: green). Pearson's r point estimates and 95% CI (x-axis) for each disease (y-axis). The p-values of the correlations are listed in Supplementary Table 10 and the number of cases and controls for each disease in Supplementary Table 2.

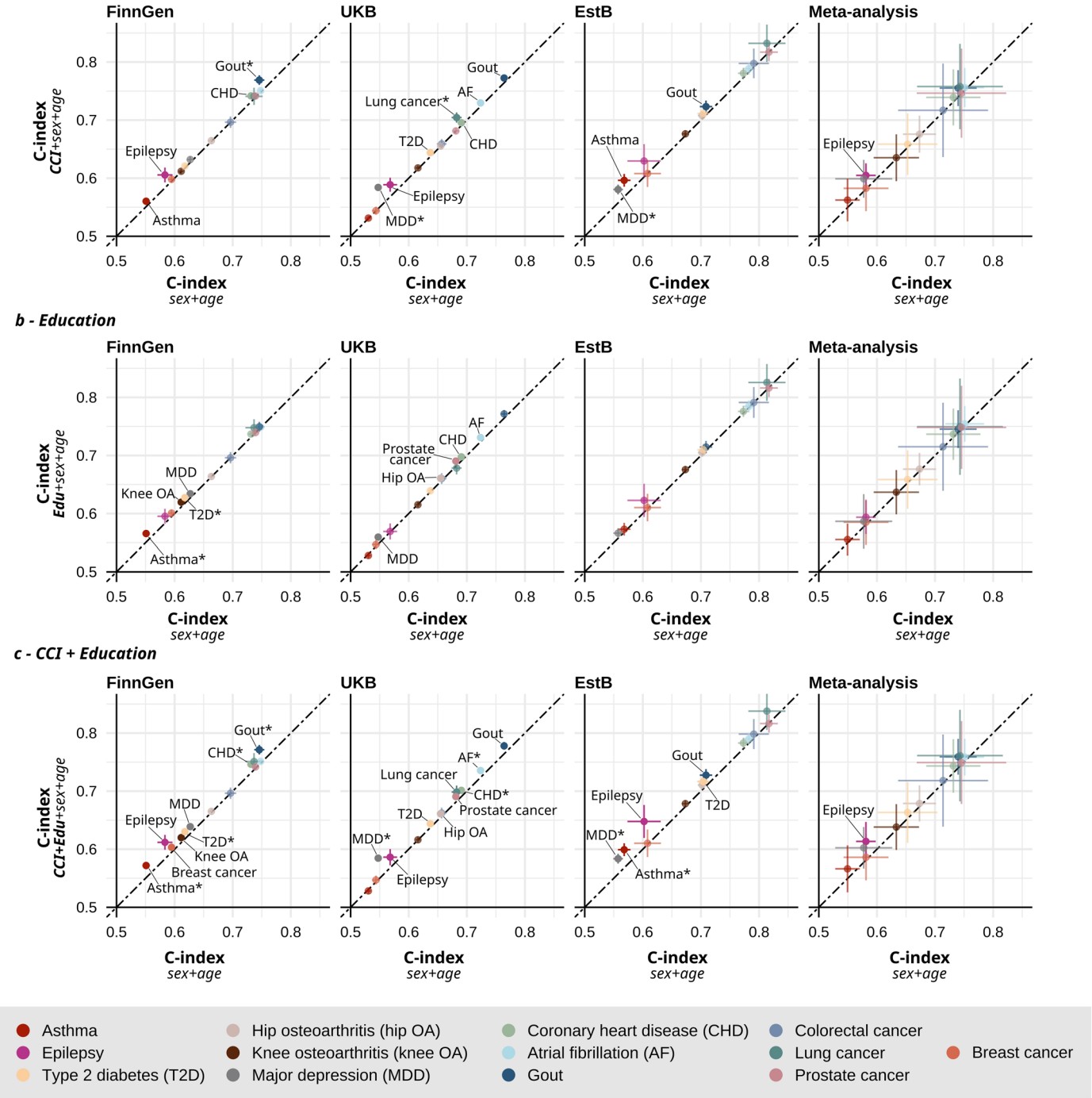

**Extended Data Fig. 2 | Prediction improvements of CCI and education.** Increase in prediction accuracy when adding: CCI (**a**), education (**b**), and both CCI and education (**c**), to a baseline model with age and sex in all three studies. All panels show the c-index point estimates and 95% CIs of the baseline model (x-axis), compared to a model with added predictors (y-axis). Diseases with significant increases in c index are labeled (p-values < 0.05; one-tailed p-values based on the z-scores of the c-index increases) and those passing Bonferroni correction for multiple hypothesis testing (p-values < 0.05/13) are marked with *. All p-values are listed in Supplementary Table 7 and the number of cases and controls for each disease in Supplementary Table 2.

## a - association of PheRS vs. number of phecodes with each disease

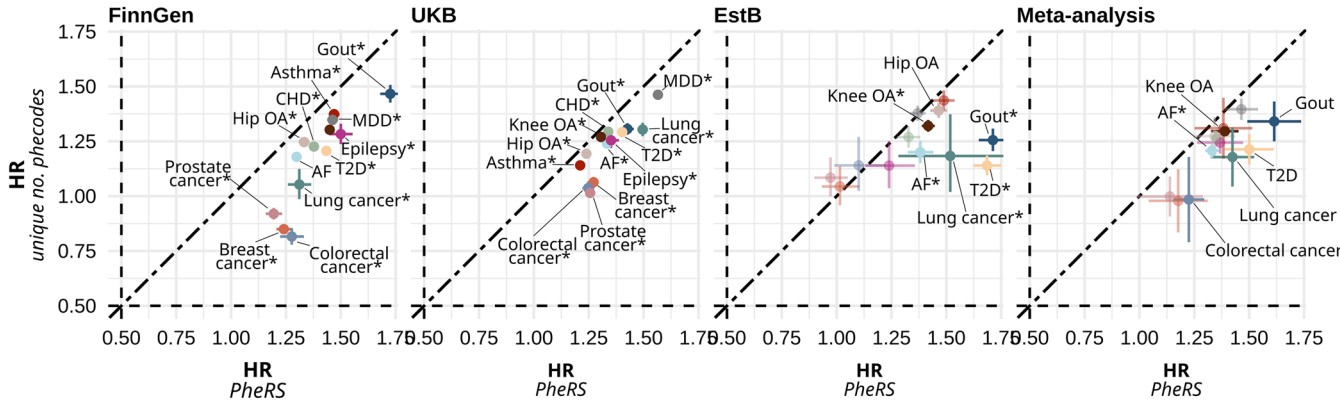

## b - PheRS HRs when adjusting for the number of phecodes

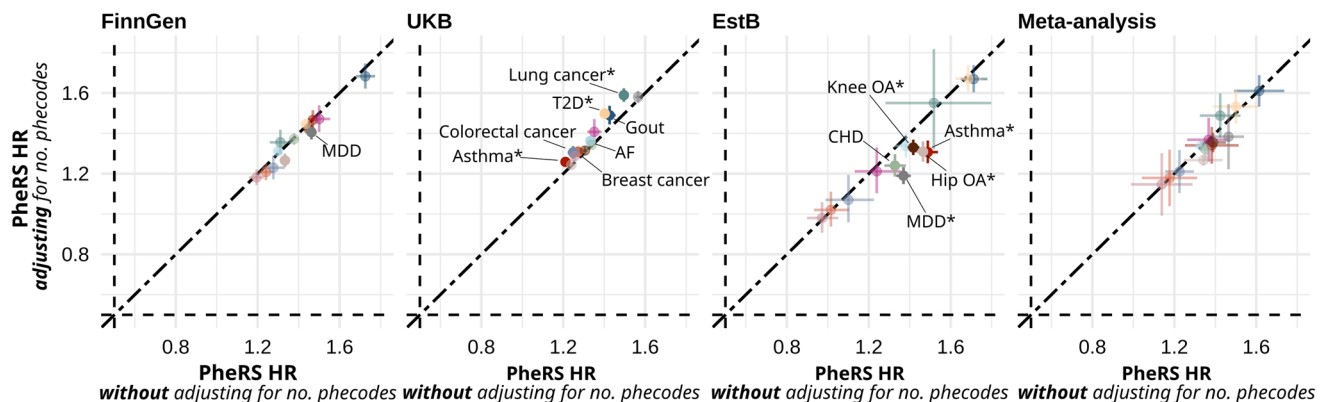

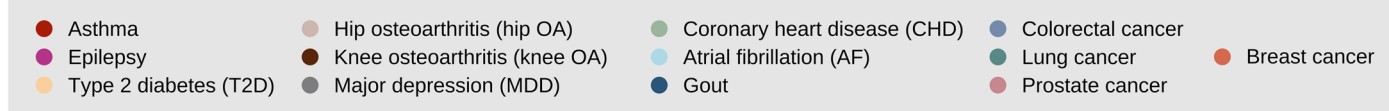

**Extended Data Fig. 3 | PheRS vs. the unique number of phecodes. a**, Association of the unique number of phecodes (y-axis) vs. PheRS (x-axis) with each disease. **b**, Association of the PheRS with each disease in a model without (x-axis) and with the unique number of phecodes (y-axis). Both panels show HR point estimates and 95% CIs. Diseases with significant differences are labeled (p-values < 0.05; two-tailed p-values based on the z-scores of the HR differences and one-tailed p-values for the c-index increases) and those passing Bonferroni correction for multiple hypothesis testing (p-values < 0.05/13) are marked with *. All p-values are listed in Supplementary Table 7 and the number of cases and controls for each disease in Supplementary Table 2.

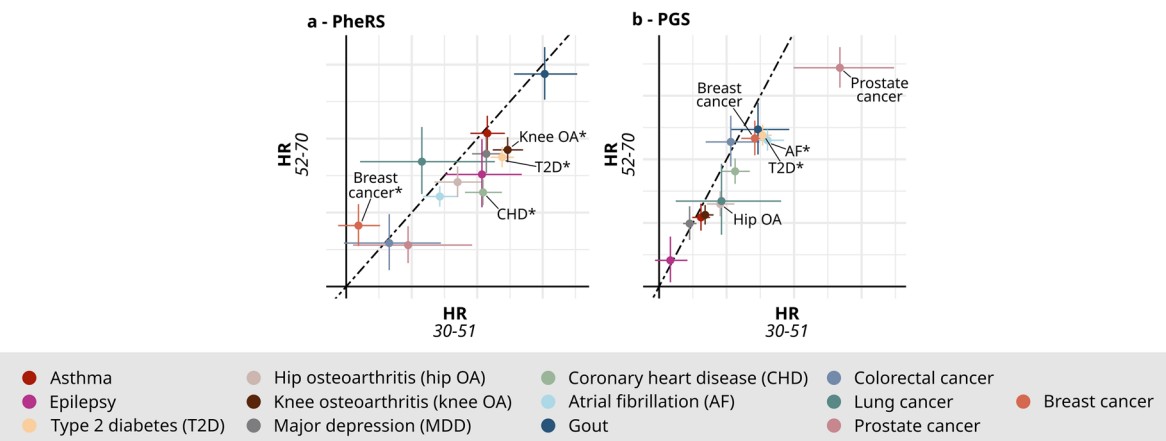

**Extended Data Fig. 4 | Age-stratified results in FinnGen.** HR point estimates and 95% CIs of the PheRS (**a**) and PGS (**b**) in younger (32-51) compared to older individuals (52–70 years). The PheRS were trained on all individuals aged 32–70 years. Diseases with significant differences are labeled (p-values < 0.05; two-tailed p-values based on the z-scores of the HR differences) and those passing Bonferroni correction for multiple hypothesis testing (p-values < 0.05/13) are marked with *. All p-values are listed in Supplementary Table 7 and the number of cases and controls for each disease in Supplementary Table 2.

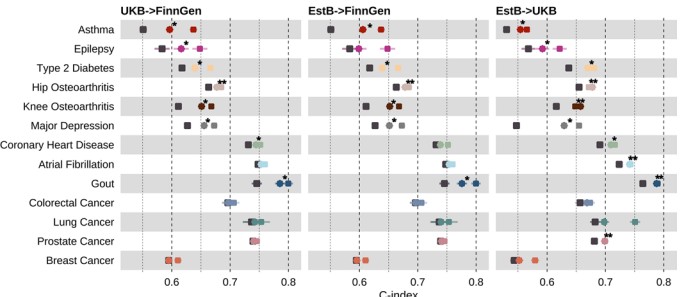

**Extended Data Fig. 5 | Internally-trained vs. externally-trained PheRS models.** The c-index point estimates and 95% CIs of the baseline models with age and sex and the models with added PheRS. The c-indices of the baseline models are shown by the dark gray boxes (the left-most marker for each disease). The results of the PheRS models trained internally in FinnGen (left and middle plot) and the UKB (right plot) are shown as boxes with the color of each disease and the externally-trained PheRS as stars. The left panel shows the results with the UKB-trained and the right panel the EstB-trained models. A single star (*) denotes diseases where the c-index increase of the externally-trained PheRS is significantly higher than the baseline and two stars (**) where this increase is not significantly different from the internally-trained PheRS. All p-values are listed in Supplementary Table 7 and the number of cases and controls for each disease in Supplementary Table 2.

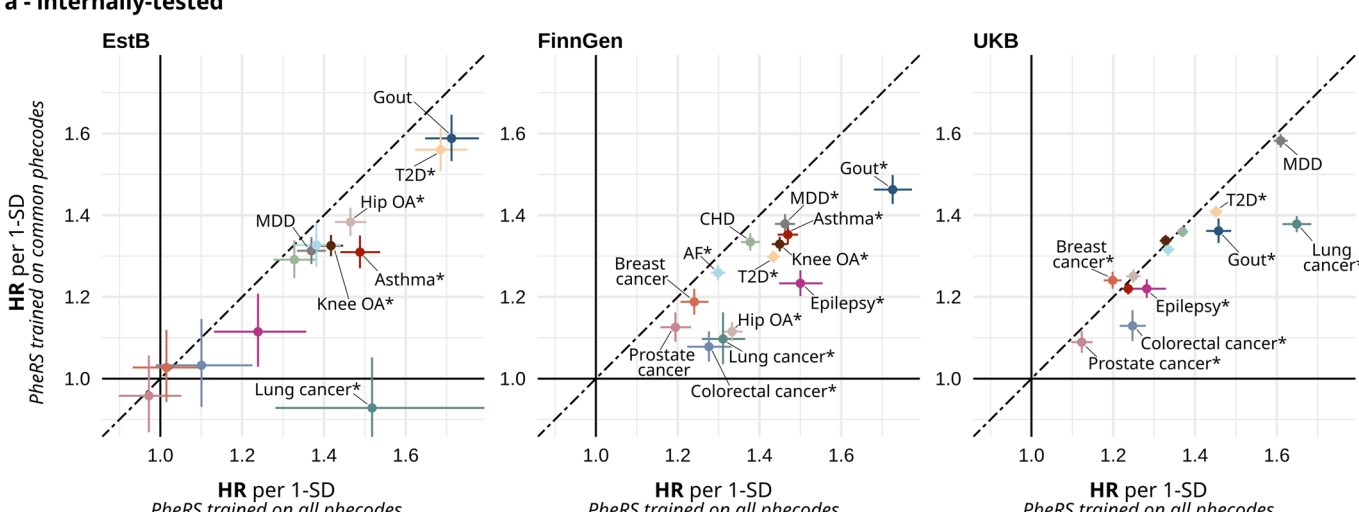

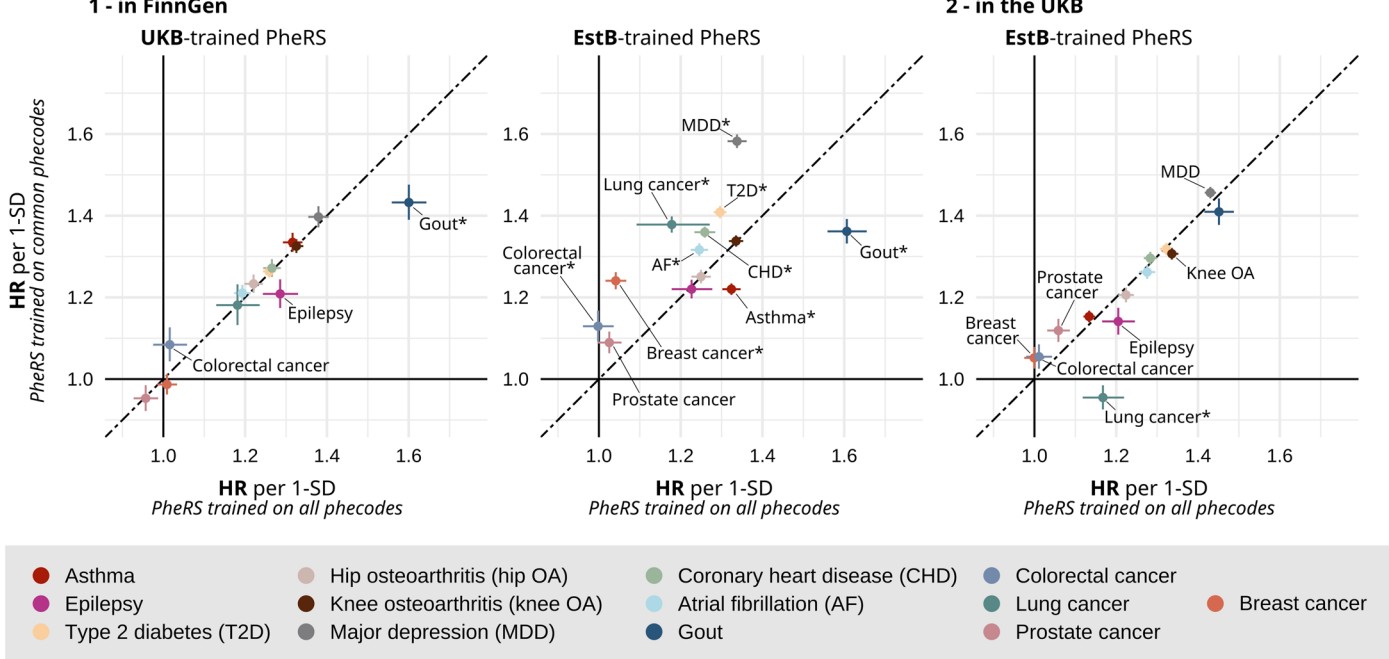

**Extended Data Fig. 6 | PheRS trained on all phecodes vs. only common phecodes.** Association of PheRS trained on all phecodes in a study (x-axis) vs. PheRS trained only on phecodes common in all three studies (y-axis). Phecodes are considered common if they are recorded for at least 1% of all three study populations during the observation period. Both panels show HR point estimates and 95% CIs. Diseases with significant differences are labeled (p-values < 0.05; two-tailed p-values based on the z-scores of the HR differences and one-tailed p-values for the c-index increases) and those passing Bonferroni correction for multiple hypothesis testing (p-values < 0.05/13) are marked with *. All p-values are listed in Supplementary Table 7 and the number of cases and controls for each disease in Supplementary Table 2. **a**, PheRS trained and tested in the same study. **b**, PheRS trained and tested in different studies. The first and second plots show the PheRS tested in FinnGen and trained in the UKB and EstB. The third plot shows the PheRS tested in the UKB and trained in the EstB.

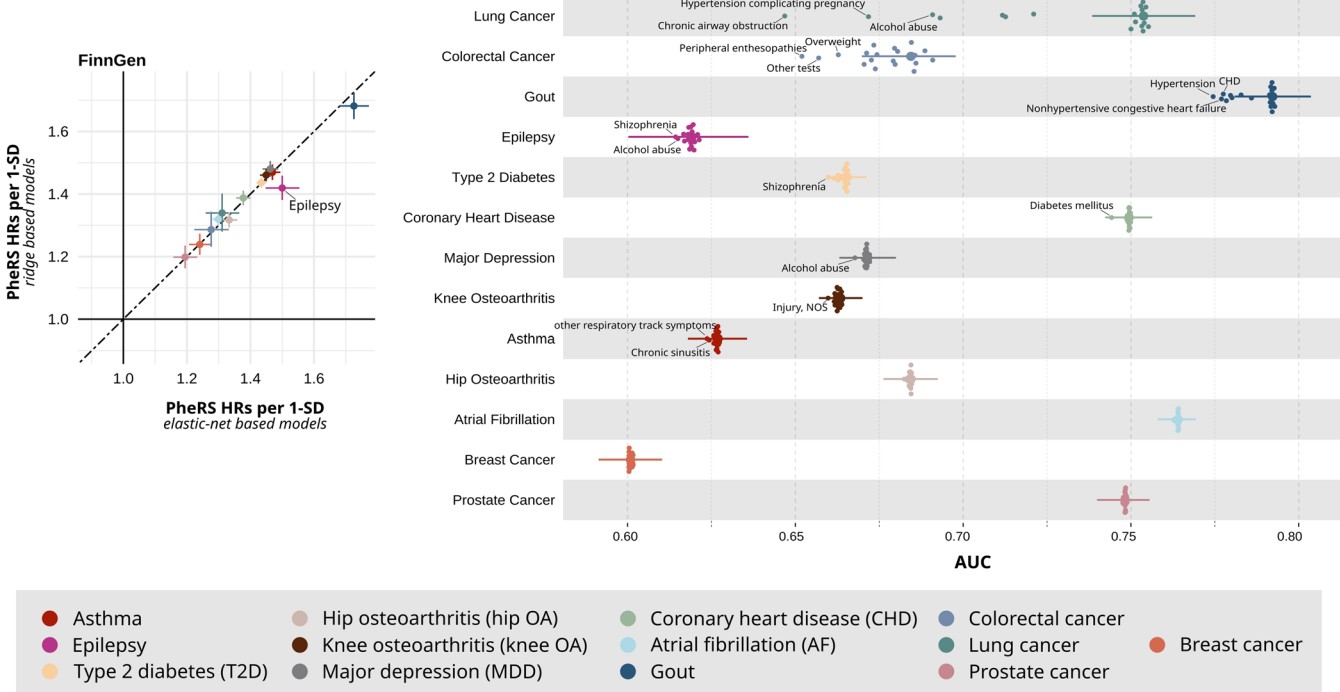

**a - ridge vs. elastic-net PheRS**

**b - leave-one-out analysis of the top predictors**

Asthma ●   Hip osteoarthritis (hip OA) ●   Coronary heart disease (CHD) ●   Colorectal cancer ●
Epilepsy ●   Knee osteoarthritis (knee OA) ●   Atrial fibrillation (AF) ●   Lung cancer ●   Breast cancer ●
Type 2 diabetes (T2D) ●   Major depression (MDD) ●   Gout ●   Prostate cancer ●

**Extended Data Fig. 7 | Leave-one-out analyses in FinnGen. a**, HR point estimates and 95% CIs for PheRS trained using ridge regression (y-axis) and elastic net (x-axis) in FinnGen and tested in FinnGen. All models also included age and sex as predictors. **b**, Leave-one-out analysis of the top 20 predictors of each PheRS model for the different diseases using ridge regression. Area under the receiver-operating curve (AUC) of each PheRS model, with the bars indicating the 95% CIs of the full model with all predictors using bootstrapping (N = 1,000).

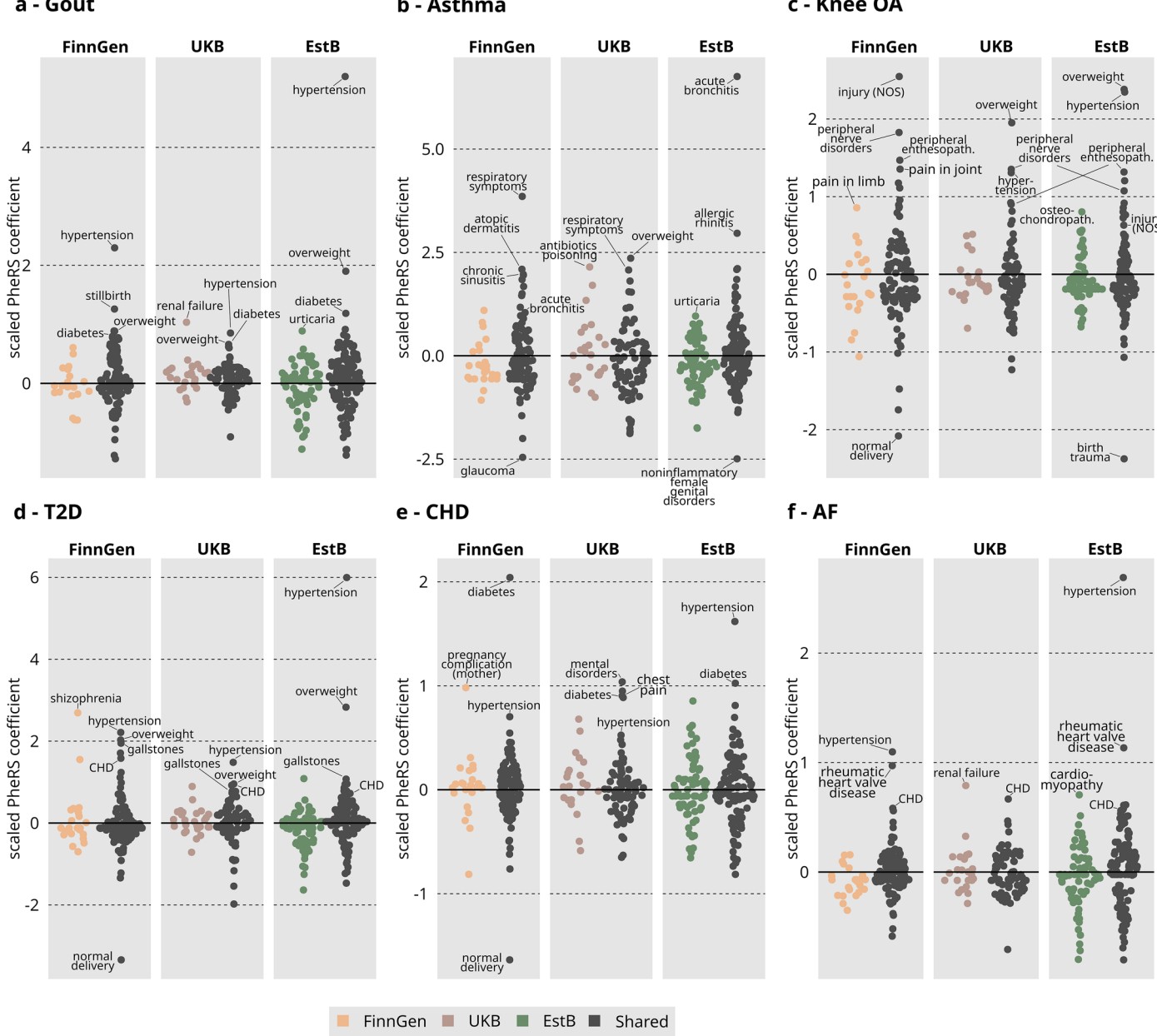

**Extended Data Fig. 8 | PheRS coefficients.** A detailed look at all the PheRS coefficients in the three studies for (**a**) gout, (**b**) asthma, (**c**) knee osteoarthritis, (**d**) type 2 diabetes, (**e**) coronary heart disease and (**f**) atrial fibrillation. Black color marks common phecodes in the PheRS models across the studies, while other colors indicate biobank-specific codes (peach = FinnGen, brown = UKB, green = EstB). PheRS coefficients are standardized to 0 mean and 1 standard deviation for each model separately for easier comparison of coefficient importances across the studies.

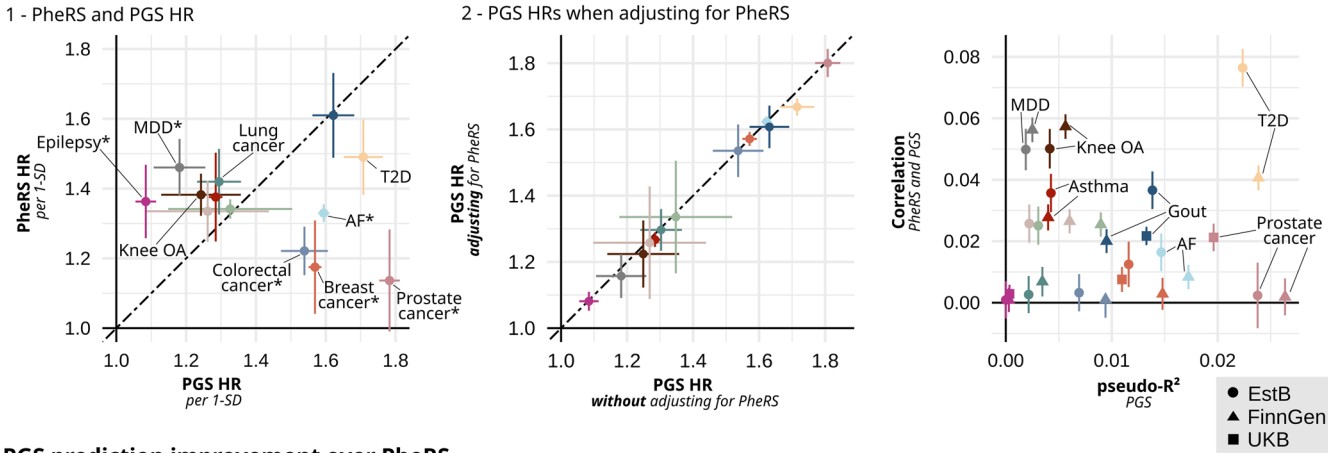

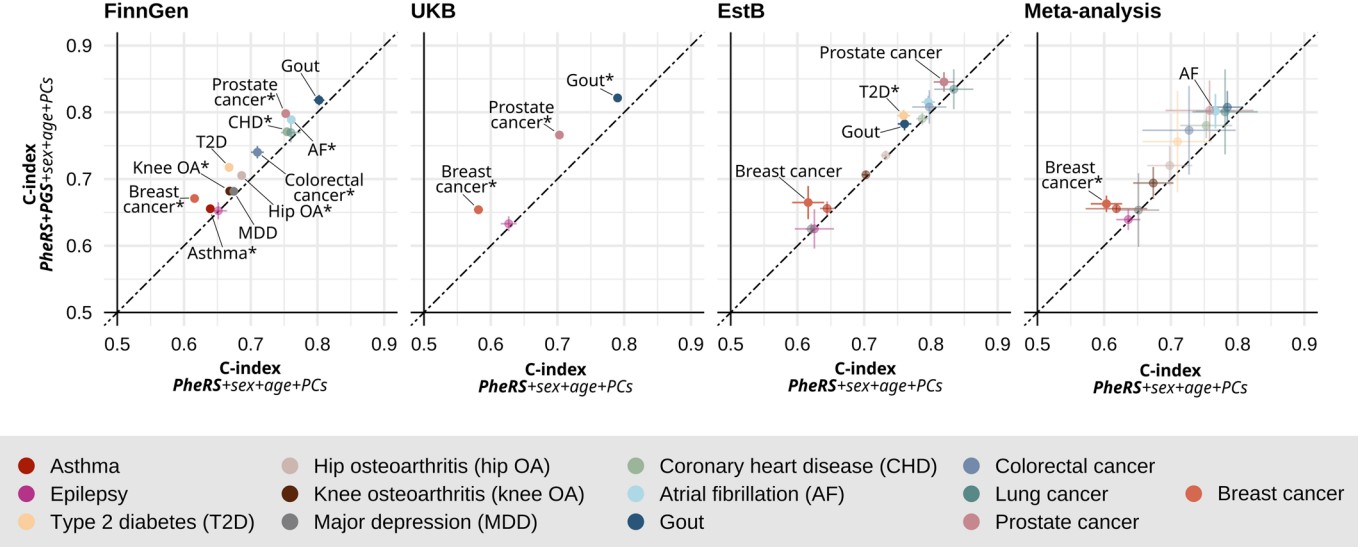

**Extended Data Fig. 9 | PheRS and PGS comparison. a**, (**1**) Association of the PGS (x-axis) and PheRS (y-axis) scores with each disease. The meta-analyzed HR point estimates and 95% CIs for 1 s.d. increase on the scores after regressing out age, sex, and the first 10 PCs. (**2**) PGS HR point estimates and 95% CIs for 1 s.d. increase in a model without PheRS (x-axis) and one where we adjust for the effect of PheRS (y-axis). **b**, Correlation (Pearsons' *r, y-axis*) point estimates and 95% CIs between the PheRS and PGS scores separately in each study (FinnGen: triangles, UKB: squares, EstB: circles), with the Nagelkerke's pseudo-R² point estimates of the PGS (x-axis). **c**, C-index improvements when adding PGS to models with PheRS.

C-index point estimates and 95% CIs of the models with only PheRS, sex, age, and PCs (x-axis) compared to those with added PGS (y-axis). Due to sample overlap with the GWASs, PGS could only be calculated for 4 diseases in UKB (see Methods for details). Diseases with significant differences are labeled (p-values < 0.05; two-tailed p-values based on the z-scores of the HR differences and one-tailed p-values for the c-index increases) and those passing Bonferroni correction for multiple hypothesis testing (p-values < 0.05/13) are marked with *. All p-values are listed in Supplementary Tables 5 and 7 and the number of cases and controls for each disease in Supplementary Table 2.

## a - association of PGS vs. PheRS with each disease

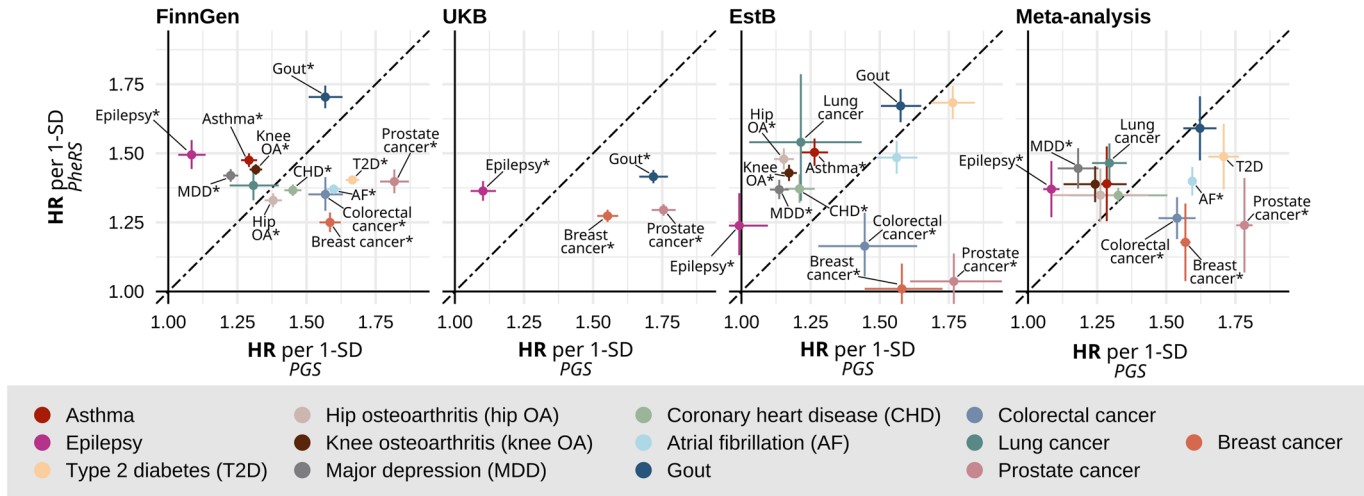

## b - PGS association with each diseases

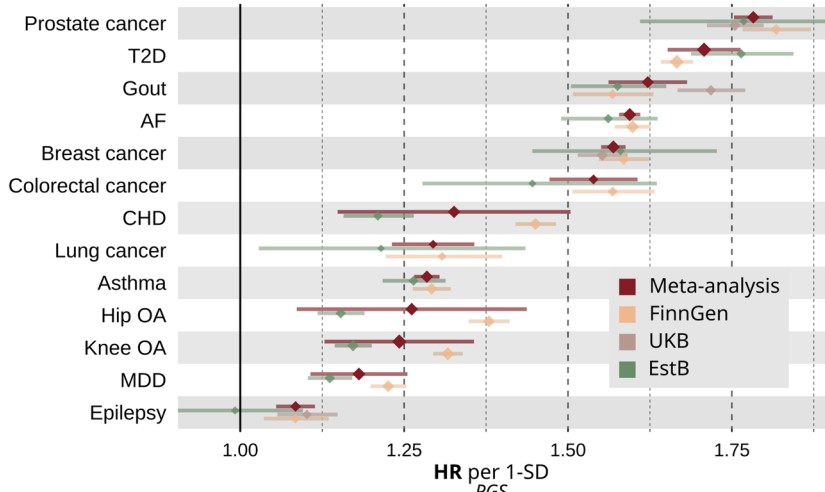

**Extended Data Fig. 10 | PGS associations. a**, Association of PGS (x-axis) vs. PheRS (y-axis) with each disease in each study. HR point estimates and 95% CIs.The HRs are shown for 1 s.d. increase of the PheRS after regressing-out age, sex, and the first 10 PCs. Diseases with significant differences are labeled (p-values < 0.05; two-tailed p-values based on the z-scores of the HR differences and one-tailed p-values for the c-index increases) and those passing Bonferroni correction for multiple hypothesis testing (p-values < 0.05/13) are marked with *. All p-values

are listed in Supplementary Tables 5 and 7 and the number of cases and controls for each disease in Supplementary Table 2. **b**, Association between PGS and disease onset during the prediction period. The HR point estimates and 95% CIs in each study−FinnGen: peach, UKB: brown, EstB: green, and meta-analyzed result: red. The HRs are shown for an increase of the PGS by 1 s.d. after regressing-out age, sex, and the first ten PCs. Due to sample overlap with the GWASs, PGS could only be calculated for 4 diseases in UKB (see Methods for details).

# Reporting Summary

## Statistics

For all statistical analyses, confirm that the following items are present in the figure legend, table legend, main text, or Methods section.

| n/a | Confirmed | |
|---|---|---|
| ☐ | ☒ | The exact sample size (*n*) for each experimental group/condition, given as a discrete number and unit of measurement |
| ☒ | ☐ | A statement on whether measurements were taken from distinct samples or whether the same sample was measured repeatedly |
| ☐ | ☒ | The statistical test(s) used AND whether they are one- or two-sided<br>*Only common tests should be described solely by name; describe more complex techniques in the Methods section.* |
| ☐ | ☒ | A description of all covariates tested |
| ☐ | ☒ | A description of any assumptions or corrections, such as tests of normality and adjustment for multiple comparisons |
| ☐ | ☒ | A full description of the statistical parameters including central tendency (e.g. means) or other basic estimates (e.g. regression coefficient) AND variation (e.g. standard deviation) or associated estimates of uncertainty (e.g. confidence intervals) |
| ☐ | ☒ | For null hypothesis testing, the test statistic (e.g. *F*, *t*, *r*) with confidence intervals, effect sizes, degrees of freedom and *P* value noted<br>*Give P values as exact values whenever suitable.* |
| ☒ | ☐ | For Bayesian analysis, information on the choice of priors and Markov chain Monte Carlo settings |
| ☒ | ☐ | For hierarchical and complex designs, identification of the appropriate level for tests and full reporting of outcomes |
| ☐ | ☒ | Estimates of effect sizes (e.g. Cohen's *d*, Pearson's *r*), indicating how they were calculated |

*Our web collection on statistics for biologists contains articles on many of the points above.*

## Software and code

Policy information about availability of computer code

| Data collection | No software was used in this study for data collection. |
|---|---|
| Data analysis | All data analysis steps are described in Methods. Briefly, The PGS were previously computed by the INTERVENE consortium using MegaPRS with the BLD-LDAK heritability model. We implemented the PheRS using the LogisticRegression function from scikit-learn (version 1.3.2). The code for training the PheRS models is available at: https://github.com/intervene-EU-H2020/INTERVENE_PheRS. All downstream data analysis was performed using R 4.4. The code and additional documentation is available here: https://github.com/intervene-EU-H2020/onset_prediction/tree/main. To calculate the CCI we used the comorbidity package (version 1.0.7) in R (see also https://github.com/dsgelab/ICCI). For regressing out the covariates age, sex, and PCs from the PheRS and PGS we use the scaled residuals from glm models with the stats package (version 3.6.2) in R. All scaling of predictors was done using the base scale function in R with mean of zero and standard deviation of 1. We used the survival package (version 3.2-7) in R for creating the Cox-PH models (function Surv and coxph) and the rcorr.cens function from the Hmisc package (version 5.1.0) to calculate the c-indices and 95% CIs. The meta-analysis of the HRs and c-indices was performed using the metafor package (version 4.6-0) in R with a random effects model. We derived two-tailed p-values based on the z-scores using the pnorm function in the stats package in R to compare the difference in HR magnitude and significant increases in the c-index. All pearsons' correlations are two sided tests using the cor.test function from the stats package in R. The rank of each phecode was simply created by ordering the coefficients in descending order and giving them ascending ranks. Figure 1 was created with the help of Biorender.com. All Figures were created in R using ggplot2 (version 3.5.1) and cowplot (version 1.1.3) together with inkscape (version 1.4). |

For manuscripts utilizing custom algorithms or software that are central to the research but not yet described in published literature, software must be made available to editors and reviewers. We strongly encourage code deposition in a community repository (e.g. GitHub). See the Nature Portfolio guidelines for submitting code & software for further information.

## Data

Policy information about availability of data

All manuscripts must include a data availability statement. This statement should provide the following information, where applicable:
- Accession codes, unique identifiers, or web links for publicly available datasets
- A description of any restrictions on data availability
- For clinical datasets or third party data, please ensure that the statement adheres to our policy

The individual-level data in these studies is protected for data privacy, access is regulated through the biobanks. The Finnish biobank data can be accessed through the Fingenious® services (https://site.fingenious.fi/en/) managed by FINBB. Researchers interested in EstBB can request access at https://genomics.ut.ee/en/content/estonian-biobank#dataaccess and the UKB data are available through a procedure described at https://www.ukbiobank.ac.uk/use-our-data/apply-for-access/. The GWAS data used in this study are available in the GWAS catalog database (https://www.ebi.ac.uk/gwas/) under accession codes listed in Supplement Table 10. The PGS scores generated in this study are available in the PGS Catalog (https://www.pgscatalog.org/) under publication ID: PGP000618 and score IDs: PGS004869-PGS004886. The mapping between ICD-, and phecodes used in this study is available at https://phewascatalog.org.

## Research involving human participants, their data, or biological material

Policy information about studies with human participants or human data. See also policy information about sex, gender (identity/presentation), and sexual orientation and race, ethnicity and racism.

| | |
|---|---|
| Reporting on sex and gender | Sex is based on register information in all three studies and was used as a covariate in all analyses. Of the 845,929 included individuals 361,462 were reported male and 484,467 female. |
| Reporting on race, ethnicity, or other socially relevant groupings | While we did not exclude individuals based on their genetic ancestry, the UKB still consists mainly, and FinnGen and the EstB almost exclusively, of individuals of European ancestry. |
| Population characteristics | The study population was restricted to individuals aged 32-70 on 01/01/2011. For each disease, we excluded individuals that already had the corresponding diagnosis before 01/01/2011. In FinnGen, we used Data Freeze 10, which includes 412,090 individuals, of which 266,179 were aged 32-70 in 01/01/2011. The longitudinal ICD-code diagnoses used to define the phecodes and the case and control status for each disease were based on in- and outpatient hospital register information. The UKB study included 464,076 individuals aged 40-70, with the ICD-code diagnoses based on inpatient information. The EstB study included 199,868 individuals of which 115,674 were aged 32-70. Here we also had primary care data as well as self-reported diagnoses available. Figure 1c shows the number of cases for each of the 13 diseases considered here in each of the three studies. For detailed information on the FinnGen study please refer to Kurki et al., Nature 2023 (Supplement Section 2). For detailed information on the UKB please refer to Bycroft et al., Nature 2018. For detailed information on the EstBB study please refer to Leitsalu et al., International Journal of Epidemiology 2015. |
| Recruitment | For detailed information on the FinnGen study please refer to Kurki et al., Nature 2023 (Supplement Section 2). For detailed information on the UKB please refer to Bycroft et al., Nature 2018. For detailed information on the EstBB study please refer to Leitsalu et al., International Journal of Epidemiology 2015. |
| Ethics oversight | Patients and control subjects in FinnGen provided informed consent for biobank research, based on the Finnish Biobank Act. Alternatively, separate research cohorts, collected prior to the Finnish Biobank Act came into effect (in September 2013) and the start of FinnGen (August 2017), were collected based on study-specific consents and later transferred to the Finnish Biobanks after approval by Fimea (Finnish Medicines Agency), the National Supervisory Authority for Welfare and Health. Recruitment protocols followed the biobank protocols approved by Fimea. The Coordinating Ethics Committee of the Hospital District of Helsinki and Uusimaa (HUS) statement number for the FinnGen study is Nr HUS/990/2017. The FinnGen study is approved by Finnish Institute for Health and Welfare (permit numbers: THL/2031/6.02.00/2017, THL/1101/5.05.00/2017, THL/341/6.02.00/2018, THL/2222/6.02.00/2018, THL/283/6.02.00/2019, THL/1721/5.05.00/2019 and THL/1524/5.05.00/2020), digital and population data service agency (permit numbers: VRK43431/2017-3, VRK/6909/2018-3, VRK/4415/2019-3), the Social Insurance Institution (permit numbers: KELA 58/522/2017, KELA 131/522/2018, KELA 70/522/2019, KELA 98/522/2019, KELA 134/522/2019, KELA 138/522/2019, KELA 2/522/2020, KELA 16/522/2020), Findata permit numbers THL/2364/14.02/2020, THL/4055/14.06.00/2020, THL/3433/14.06.00/2020, THL/4432/14.06/2020, THL/5189/14.06/2020, THL/5894/14.06.00/2020, THL/6619/14.06.00/2020, THL/209/14.06.00/2021, THL/688/14.06.00/2021, THL/1284/14.06.00/2021, THL/1965/14.06.00/2021, THL/5546/14.02.00/2020, THL/2658/14.06.00/2021, THL/4235/14.06.00/2021, Statistics Finland (permit numbers: TK-53-1041-17 and TK/143/07.03.00/2020 (earlier TK-53-90-20) TK/1735/07.03.00/2021, TK/3112/07.03.00/2021) and Finnish Registry for Kidney Diseases permission/extract from the meeting minutes on 4th July 2019. The Biobank Access Decisions for FinnGen samples and data utilized in FinnGen Data Freeze 10 include: THL Biobank BB2017_55, BB2017_111, BB2018_19, BB_2018_34, BB_2018_67, BB2018_71, BB2019_7, BB2019_8, BB2019_26, BB2020_1, BB2021_65, Finnish Red Cross Blood Service Biobank 7.12.2017, Helsinki Biobank HUS/359/2017, HUS/248/2020, HUS/150/2022 § 12, §13, §14, §15, §16, §17, §18, and §23, Auria Biobank AB17-5154 and amendment #1 (August 17 2020) and amendments BB_2021-0140, BB_2021-0156 (August 26 2021, Feb 2 2022), BB_2021-0169, BB_2021-0179, BB_2021-0161, AB20-5926 and amendment #1 (April 23 2020)and it´s modification (Sep 22 2021), Biobank Borealis of Northern Finland_2017_1013, 2021_5010, 2021_5018, 2021_5015, 2021_5023, 2021_5017, 2022_6001, Biobank of Eastern Finland 1186/2018 and amendment 22 § /2020, 53§/2021, 13§/2022, 14§/2022, 15§/2022, Finnish Clinical Biobank Tampere MH0004 and amendments (21.02.2020 and 06.10.2020), §8/2021, §9/2022, §10/2022, §12/2022, §20/2022, §21/2022, §22/2022, §23/2022, Central Finland Biobank 1-2017, and Terveystalo Biobank STB 2018001 |

and amendment 25th Aug 2020, Finnish Hematological Registry and Clinical Biobank decision 18th June 2021, Arctic Biobank P0844: ARC_2021_1001.

Ethics approval for the UK Biobank study was obtained from the North West Centre for Research Ethics Committee (11/NW/0382). UK Biobank data used in this study were obtained under approved application 78537.

The activities of the EstBB are regulated by the Human Genes Research Act, which was adopted in 2000 specifically for the operations of the EstBB. Individual level data analysis in the EstBB was carried out under ethical approval 1.1-12/624 from the Estonian Committee on Bioethics and Human Research (Estonian Ministry of Social Affairs), using data according to release application S22, document number 6-7/GI/16259 from the EstBB.

Note that full information on the approval of the study protocol must also be provided in the manuscript.

# Field-specific reporting

Please select the one below that is the best fit for your research. If you are not sure, read the appropriate sections before making your selection.

☒ Life sciences ☐ Behavioural & social sciences ☐ Ecological, evolutionary & environmental sciences

For a reference copy of the document with all sections, see nature.com/documents/nr-reporting-summary-flat.pdf

# Life sciences study design

All studies must disclose on these points even when the disclosure is negative.

| | |
|---|---|
| Sample size | In total we included 845,929 individuals aged 32 to 70 on 01/01/2011. No sample size calculation was performed - all eligible data after the exclusion rules (see below) were included. In FinnGen we used Data Freeze 10, which includes 412,090 individuals, of which 266,179 were aged 32-70 in 01/01/2011. The UKB study included 464,076 individuals aged 40-70. The EstB study included 199,868 individuals of which 115,674 were aged 32-70. The individuals gathered a total of 293,019 new diagnoses during an 8-year prediction period (01/01/2011 - 31/12/2018) across 13 common and high-burden diseases. All sample sizes in each study and for each disease are listed in Supplement Table 2. The available sample sizes were sufficient to observe statistically significant signals for all diseases considered. |
| Data exclusions | We only considered phecodes with at least 1% prevalence in each study. As our target phenotypes were defined based on ICD-codes we exclude predictors part of the exclusion range of the phecodes separately for each phenotype. We only considered individuals aged between 32 and 70 on 01/01/2011 and separately removed all individuals diagnosed before 01/01/2011. For the Cox-PH models we removed individuals from the studies that were part of the GWAS on which the PGS were based. In the UKB we only considered the diseases for gout, epilepsy, breast and prostate cancer for the PGS due to the large overlap with the UKB individuals. |
| Replication | We performed the analyses separately in three biobank-based studies (FinnGen, EstBB, and UKB). Additionally, we transferred the PheRS models trained in FinnGen to EstBB and UKB. Not all of the results replicated in each three studies, detailed description of each replication analysis can be found from the manuscript. |
| Randomization | Not relevant, cases and controls were based on recorded disease diagnoses in the biobank studies. All cases and controls passing the exclusion criteria were used. |
| Blinding | Not relevant as the analyses were based on large data sets of registry and genetic data using computational methods. |

# Behavioural & social sciences study design

All studies must disclose on these points even when the disclosure is negative.

| | |
|---|---|
| Study description | *Briefly describe the study type including whether data are quantitative, qualitative, or mixed-methods (e.g. qualitative cross-sectional, quantitative experimental, mixed-methods case study).* |
| Research sample | *State the research sample (e.g. Harvard university undergraduates, villagers in rural India) and provide relevant demographic information (e.g. age, sex) and indicate whether the sample is representative. Provide a rationale for the study sample chosen. For studies involving existing datasets, please describe the dataset and source.* |
| Sampling strategy | *Describe the sampling procedure (e.g. random, snowball, stratified, convenience). Describe the statistical methods that were used to predetermine sample size OR if no sample-size calculation was performed, describe how sample sizes were chosen and provide a rationale for why these sample sizes are sufficient. For qualitative data, please indicate whether data saturation was considered, and what criteria were used to decide that no further sampling was needed.* |
| Data collection | *Provide details about the data collection procedure, including the instruments or devices used to record the data (e.g. pen and paper, computer, eye tracker, video or audio equipment) whether anyone was present besides the participant(s) and the researcher, and whether the researcher was blind to experimental condition and/or the study hypothesis during data collection.* |
| Timing | *Indicate the start and stop dates of data collection. If there is a gap between collection periods, state the dates for each sample cohort.* |
| Data exclusions | *If no data were excluded from the analyses, state so OR if data were excluded, provide the exact number of exclusions and the rationale behind them, indicating whether exclusion criteria were pre-established.* |

| Non-participation | *State how many participants dropped out/declined participation and the reason(s) given OR provide response rate OR state that no participants dropped out/declined participation.* |
|---|---|
| Randomization | *If participants were not allocated into experimental groups, state so OR describe how participants were allocated to groups, and if allocation was not random, describe how covariates were controlled.* |

# Ecological, evolutionary & environmental sciences study design

All studies must disclose on these points even when the disclosure is negative.

| Study description | *Briefly describe the study. For quantitative data include treatment factors and interactions, design structure (e.g. factorial, nested, hierarchical), nature and number of experimental units and replicates.* |
|---|---|
| Research sample | *Describe the research sample (e.g. a group of tagged Passer domesticus, all Stenocereus thurberi within Organ Pipe Cactus National Monument), and provide a rationale for the sample choice. When relevant, describe the organism taxa, source, sex, age range and any manipulations. State what population the sample is meant to represent when applicable. For studies involving existing datasets, describe the data and its source.* |
| Sampling strategy | *Note the sampling procedure. Describe the statistical methods that were used to predetermine sample size OR if no sample-size calculation was performed, describe how sample sizes were chosen and provide a rationale for why these sample sizes are sufficient.* |
| Data collection | *Describe the data collection procedure, including who recorded the data and how.* |
| Timing and spatial scale | *Indicate the start and stop dates of data collection, noting the frequency and periodicity of sampling and providing a rationale for these choices. If there is a gap between collection periods, state the dates for each sample cohort. Specify the spatial scale from which the data are taken* |
| Data exclusions | *If no data were excluded from the analyses, state so OR if data were excluded, describe the exclusions and the rationale behind them, indicating whether exclusion criteria were pre-established.* |
| Reproducibility | *Describe the measures taken to verify the reproducibility of experimental findings. For each experiment, note whether any attempts to repeat the experiment failed OR state that all attempts to repeat the experiment were successful.* |
| Randomization | *Describe how samples/organisms/participants were allocated into groups. If allocation was not random, describe how covariates were controlled. If this is not relevant to your study, explain why.* |
| Blinding | *Describe the extent of blinding used during data acquisition and analysis. If blinding was not possible, describe why OR explain why blinding was not relevant to your study.* |

Did the study involve field work? ☐ Yes ☐ No

## Field work, collection and transport

| Field conditions | *Describe the study conditions for field work, providing relevant parameters (e.g. temperature, rainfall).* |
|---|---|
| Location | *State the location of the sampling or experiment, providing relevant parameters (e.g. latitude and longitude, elevation, water depth).* |
| Access & import/export | *Describe the efforts you have made to access habitats and to collect and import/export your samples in a responsible manner and in compliance with local, national and international laws, noting any permits that were obtained (give the name of the issuing authority, the date of issue, and any identifying information).* |
| Disturbance | *Describe any disturbance caused by the study and how it was minimized.* |

# Reporting for specific materials, systems and methods

We require information from authors about some types of materials, experimental systems and methods used in many studies. Here, indicate whether each material, system or method listed is relevant to your study. If you are not sure if a list item applies to your research, read the appropriate section before selecting a response.

## Materials & experimental systems

| n/a | Involved in the study |
|-----|----------------------|
| ☒ | Antibodies |
| ☒ | Eukaryotic cell lines |
| ☒ | Palaeontology and archaeology |
| ☒ | Animals and other organisms |
| ☒ | Clinical data |
| ☒ | Dual use research of concern |
| ☒ | Plants |

## Methods

| n/a | Involved in the study |
|-----|----------------------|
| ☒ | ChIP-seq |
| ☒ | Flow cytometry |
| ☒ | MRI-based neuroimaging |

## Antibodies

| | |
|---|---|
| Antibodies used | *Describe all antibodies used in the study; as applicable, provide supplier name, catalog number, clone name, and lot number.* |
| Validation | *Describe the validation of each primary antibody for the species and application, noting any validation statements on the manufacturer's website, relevant citations, antibody profiles in online databases, or data provided in the manuscript.* |

## Eukaryotic cell lines

Policy information about cell lines and Sex and Gender in Research

| | |
|---|---|
| Cell line source(s) | *State the source of each cell line used and the sex of all primary cell lines and cells derived from human participants or vertebrate models.* |
| Authentication | *Describe the authentication procedures for each cell line used OR declare that none of the cell lines used were authenticated.* |
| Mycoplasma contamination | *Confirm that all cell lines tested negative for mycoplasma contamination OR describe the results of the testing for mycoplasma contamination OR declare that the cell lines were not tested for mycoplasma contamination.* |
| Commonly misidentified lines (See ICLAC register) | *Name any commonly misidentified cell lines used in the study and provide a rationale for their use.* |

## Palaeontology and Archaeology

| | |
|---|---|
| Specimen provenance | *Provide provenance information for specimens and describe permits that were obtained for the work (including the name of the issuing authority, the date of issue, and any identifying information). Permits should encompass collection and, where applicable, export.* |
| Specimen deposition | *Indicate where the specimens have been deposited to permit free access by other researchers.* |
| Dating methods | *If new dates are provided, describe how they were obtained (e.g. collection, storage, sample pretreatment and measurement), where they were obtained (i.e. lab name), the calibration program and the protocol for quality assurance OR state that no new dates are provided.* |

☐ Tick this box to confirm that the raw and calibrated dates are available in the paper or in Supplementary Information.

| | |
|---|---|
| Ethics oversight | *Identify the organization(s) that approved or provided guidance on the study protocol, OR state that no ethical approval or guidance was required and explain why not.* |

Note that full information on the approval of the study protocol must also be provided in the manuscript.

## Animals and other research organisms

Policy information about studies involving animals; ARRIVE guidelines recommended for reporting animal research, and Sex and Gender in Research

| | |
|---|---|
| Laboratory animals | *For laboratory animals, report species, strain and age OR state that the study did not involve laboratory animals.* |
| Wild animals | *Provide details on animals observed in or captured in the field; report species and age where possible. Describe how animals were caught and transported and what happened to captive animals after the study (if killed, explain why and describe method; if released, say where and when) OR state that the study did not involve wild animals.* |
| Reporting on sex | *Indicate if findings apply to only one sex; describe whether sex was considered in study design, methods used for assigning sex. Provide data disaggregated for sex where this information has been collected in the source data as appropriate; provide overall* |

*numbers in this Reporting Summary. Please state if this information has not been collected.  Report sex-based analyses where performed, justify reasons for lack of sex-based analysis.*

Field-collected samples | *For laboratory work with field-collected samples, describe all relevant parameters such as housing, maintenance, temperature, photoperiod and end-of-experiment protocol OR state that the study did not involve samples collected from the field.*

Ethics oversight | *Identify the organization(s) that approved or provided guidance on the study protocol, OR state that no ethical approval or guidance was required and explain why not.*

Note that full information on the approval of the study protocol must also be provided in the manuscript.

# Clinical data

Policy information about clinical studies

All manuscripts should comply with the ICMJE guidelines for publication of clinical research and a completed CONSORT checklist must be included with all submissions.

Clinical trial registration | *Provide the trial registration number from ClinicalTrials.gov or an equivalent agency.*

Study protocol | *Note where the full trial protocol can be accessed OR if not available, explain why.*

Data collection | *Describe the settings and locales of data collection, noting the time periods of recruitment and data collection.*

Outcomes | *Describe how you pre-defined primary and secondary outcome measures and how you assessed these measures.*

# Dual use research of concern

Policy information about dual use research of concern

## Hazards

Could the accidental, deliberate or reckless misuse of agents or technologies generated in the work, or the application of information presented in the manuscript, pose a threat to:

No | Yes
- ☐ | ☐ Public health
- ☐ | ☐ National security
- ☐ | ☐ Crops and/or livestock
- ☐ | ☐ Ecosystems
- ☐ | ☐ Any other significant area

## Experiments of concern

Does the work involve any of these experiments of concern:

No | Yes
- ☐ | ☐ Demonstrate how to render a vaccine ineffective
- ☐ | ☐ Confer resistance to therapeutically useful antibiotics or antiviral agents
- ☐ | ☐ Enhance the virulence of a pathogen or render a nonpathogen virulent
- ☐ | ☐ Increase transmissibility of a pathogen
- ☐ | ☐ Alter the host range of a pathogen
- ☐ | ☐ Enable evasion of diagnostic/detection modalities
- ☐ | ☐ Enable the weaponization of a biological agent or toxin
- ☐ | ☐ Any other potentially harmful combination of experiments and agents

# Plants

| | |
|---|---|
| Seed stocks | *Report on the source of all seed stocks or other plant material used. If applicable, state the seed stock centre and catalogue number. If plant specimens were collected from the field, describe the collection location, date and sampling procedures.* |
| Novel plant genotypes | *Describe the methods by which all novel plant genotypes were produced. This includes those generated by transgenic approaches, gene editing, chemical/radiation-based mutagenesis and hybridization. For transgenic lines, describe the transformation method, the number of independent lines analyzed and the generation upon which experiments were performed. For gene-edited lines, describe the editor used, the endogenous sequence targeted for editing, the targeting guide RNA sequence (if applicable) and how the editor was applied.* |
| Authentication | *Describe any authentication procedures for each seed stock used or novel genotype generated. Describe any experiments used to assess the effect of a mutation and, where applicable, how potential secondary effects (e.g. second site T-DNA insertions, mosiacism, off-target gene editing) were examined.* |

# ChIP-seq

## Data deposition

☐ Confirm that both raw and final processed data have been deposited in a public database such as GEO.

☐ Confirm that you have deposited or provided access to graph files (e.g. BED files) for the called peaks.

| | |
|---|---|
| Data access links
*May remain private before publication.* | *For "Initial submission" or "Revised version" documents, provide reviewer access links. For your "Final submission" document, provide a link to the deposited data.* |
| Files in database submission | *Provide a list of all files available in the database submission.* |
| Genome browser session
(e.g. UCSC) | *Provide a link to an anonymized genome browser session for "Initial submission" and "Revised version" documents only, to enable peer review. Write "no longer applicable" for "Final submission" documents.* |

## Methodology

| | |
|---|---|
| Replicates | *Describe the experimental replicates, specifying number, type and replicate agreement.* |
| Sequencing depth | *Describe the sequencing depth for each experiment, providing the total number of reads, uniquely mapped reads, length of reads and whether they were paired- or single-end.* |
| Antibodies | *Describe the antibodies used for the ChIP-seq experiments; as applicable, provide supplier name, catalog number, clone name, and lot number.* |
| Peak calling parameters | *Specify the command line program and parameters used for read mapping and peak calling, including the ChIP, control and index files used.* |
| Data quality | *Describe the methods used to ensure data quality in full detail, including how many peaks are at FDR 5% and above 5-fold enrichment.* |
| Software | *Describe the software used to collect and analyze the ChIP-seq data. For custom code that has been deposited into a community repository, provide accession details.* |

# Flow Cytometry

## Plots

Confirm that:

☐ The axis labels state the marker and fluorochrome used (e.g. CD4-FITC).

☐ The axis scales are clearly visible. Include numbers along axes only for bottom left plot of group (a 'group' is an analysis of identical markers).

☐ All plots are contour plots with outliers or pseudocolor plots.

☐ A numerical value for number of cells or percentage (with statistics) is provided.

## Methodology

| | |
|---|---|
| Sample preparation | *Describe the sample preparation, detailing the biological source of the cells and any tissue processing steps used.* |
| Instrument | *Identify the instrument used for data collection, specifying make and model number.* |
| Software | *Describe the software used to collect and analyze the flow cytometry data. For custom code that has been deposited into a community repository, provide accession details.* |

| Cell population abundance | *Describe the abundance of the relevant cell populations within post-sort fractions, providing details on the purity of the samples and how it was determined.* |
| Gating strategy | *Describe the gating strategy used for all relevant experiments, specifying the preliminary FSC/SSC gates of the starting cell population, indicating where boundaries between "positive" and "negative" staining cell populations are defined.* |

☐ Tick this box to confirm that a figure exemplifying the gating strategy is provided in the Supplementary Information.

# Magnetic resonance imaging

## Experimental design

| Design type | *Indicate task or resting state; event-related or block design.* |
| Design specifications | *Specify the number of blocks, trials or experimental units per session and/or subject, and specify the length of each trial or block (if trials are blocked) and interval between trials.* |
| Behavioral performance measures | *State number and/or type of variables recorded (e.g. correct button press, response time) and what statistics were used to establish that the subjects were performing the task as expected (e.g. mean, range, and/or standard deviation across subjects).* |

## Acquisition

| Imaging type(s) | *Specify: functional, structural, diffusion, perfusion.* |
| Field strength | *Specify in Tesla* |
| Sequence & imaging parameters | *Specify the pulse sequence type (gradient echo, spin echo, etc.), imaging type (EPI, spiral, etc.), field of view, matrix size, slice thickness, orientation and TE/TR/flip angle.* |
| Area of acquisition | *State whether a whole brain scan was used OR define the area of acquisition, describing how the region was determined.* |

Diffusion MRI          ☐ Used          ☐ Not used

## Preprocessing

| Preprocessing software | *Provide detail on software version and revision number and on specific parameters (model/functions, brain extraction, segmentation, smoothing kernel size, etc.).* |
| Normalization | *If data were normalized/standardized, describe the approach(es): specify linear or non-linear and define image types used for transformation OR indicate that data were not normalized and explain rationale for lack of normalization.* |
| Normalization template | *Describe the template used for normalization/transformation, specifying subject space or group standardized space (e.g. original Talairach, MNI305, ICBM152) OR indicate that the data were not normalized.* |
| Noise and artifact removal | *Describe your procedure(s) for artifact and structured noise removal, specifying motion parameters, tissue signals and physiological signals (heart rate, respiration).* |
| Volume censoring | *Define your software and/or method and criteria for volume censoring, and state the extent of such censoring.* |

## Statistical modeling & inference

| Model type and settings | *Specify type (mass univariate, multivariate, RSA, predictive, etc.) and describe essential details of the model at the first and second levels (e.g. fixed, random or mixed effects; drift or auto-correlation).* |
| Effect(s) tested | *Define precise effect in terms of the task or stimulus conditions instead of psychological concepts and indicate whether ANOVA or factorial designs were used.* |

Specify type of analysis:          ☐ Whole brain          ☐ ROI-based          ☐ Both

Statistic type for inference          *Specify voxel-wise or cluster-wise and report all relevant parameters for cluster-wise methods.*

(See Eklund et al. 2016)

| Correction | *Describe the type of correction and how it is obtained for multiple comparisons (e.g. FWE, FDR, permutation or Monte Carlo).* |

## Models & analysis

| n/a | Involved in the study |
|-----|----------------------|
| ☐ | ☐ Functional and/or effective connectivity |
| ☐ | ☐ Graph analysis |
| ☐ | ☐ Multivariate modeling or predictive analysis |

**Functional and/or effective connectivity**
*Report the measures of dependence used and the model details (e.g. Pearson correlation, partial correlation, mutual information).*

**Graph analysis**
*Report the dependent variable and connectivity measure, specifying weighted graph or binarized graph, subject- or group-level, and the global and/or node summaries used (e.g. clustering coefficient, efficiency, etc.).*

**Multivariate modeling and predictive analysis**
*Specify independent variables, features extraction and dimension reduction, model, training and evaluation metrics.*

