## [Peer Review File · Nature Genetics]

Cross-biobank generalizability and accuracy of electronic health record-based predictors compared to polygenic scores

Corresponding Author: Professor Andrea Ganna

Version 0:

Decision Letter:

16th Oct 2024

Dear Professor Ganna,

Your Article, "Transferability and accuracy of electronic health record-based predictors compared to polygenic scores" has now been seen by 2 referees. You will see from their comments below that while they find your work of interest, some important points are raised. We are interested in the possibility of publishing your study in Nature Genetics, but would like to consider your response to these concerns in the form of a revised manuscript before we make a final decision on publication.

To guide the scope of the revisions, the editors discuss the referee reports in detail within the team with a view to identifying key priorities that should be addressed in revision. In this case, we think both referees have provided constructive reviews aimed at strengthening the analyses and improving the presentation. We particularly ask that you enhance the comparison of PheRS and PGS, and address all referee comments as thoroughly as possible with appropriate revisions. We hope that you will find the prioritized set of referee points to be useful when revising your study.

We therefore invite you to revise your manuscript taking into account all reviewer and editor comments. Please highlight all changes in the manuscript text file. At this stage we will need you to upload a copy of the manuscript in MS Word .docx or similar editable format.

*2) If you have not done so already please begin to revise your manuscript so that it conforms to our Article format instructions, available

[here](http://www.nature.com/ng/authors/article_types/index.html).

*3) Include a revised version of any required Reporting Summary: <https://www.nature.com/documents/nr-reporting-summary.pdf>

Link Redacted

We hope to receive your revised manuscript within 3 to 6 months. If you cannot send it within this time, please let us know.

Nature Genetics is committed to improving transparency in authorship. As part of our efforts in this direction, we are now requesting that all authors identified as 'corresponding author' on published papers create and link their Open Researcher and Contributor Identifier (ORCID) with their account on the Manuscript Tracking System (MTS), prior to acceptance. ORCID helps the scientific community achieve unambiguous attribution of all scholarly contributions. You can create and link your ORCID from the home page of the MTS by clicking on 'Modify my Springer Nature account'. For more information please visit please visit www.springernature.com/orcid.

Sincerely,
Wei

Wei Li, PhD
Senior Editor
Nature Genetics
www.nature.com/ng

Reviewers' Comments:

Reviewer #1 (Remarks to the Author):

This is an interesting and timely study on the transferability of EHR Phecode-based predictors relative to polygenic scores across three European studies. The key conclusions are that transferability is quite good, and that PheRS are mostly orthogonal to PGS and hence integrated scores are likely better predictors. I do not find any major issues with the analysis or presentation, but do feel that there are a couple of questions that could be answered that would be within scope of the paper. One is that the study does not directly address the reason for the overall lower performance of PheRS in the UKB relative to EstB and FinnGen. The other is that the repeatability and robustness of PheRS is not quantified.

Regarding the first concern, it is notable in Figure 2A that UKB HRs are consistently the lowest, particularly for the non-cancers. In places the authors hint that this might be due to the balance of primary and secondary records in the UK versus Finland and Estonia. It is an important point, particularly in light of emerging evidence that heritability in AllofUS is consistently lower, almost half, than it is in matched populations in the UKB (for example: <https://www.biorxiv.org/content/10.1101/2024.08.06.606846v1>) whereas the authors cite a study [ref 31] showing that PheRS transferability is good from UKB to AllofUS in the sense of improving risk prediction – though no relative performance was mentioned. In any case, it is also not clear from the description how external training using internal model parameters deals with the fact that many of the Phecodes are missing in external datasets – do these just drop from the model, and is that the reason for drop in performance? Is it possible to perform down-sampling of the included Phecodes to either include the same number in each study, or preferably just the same Phecodes, such that relative performance is then mostly a function of prevalence and comorbidity? Relatedly, are the results reported in Figure 3B replicated when applied to predicting disease in UKB?

Regarding repeatability, the authors make the interesting observation in Figure 5 that HR comparing the top 10% to average 20% (also please define this term – is it the middle 40%-60% decile or a permuted average) is notably greater for PheRS than PGS. Given Ding et al's findings (<https://pubmed.ncbi.nlm.nih.gov/34931067/>) regarding large uncertainty in PGS decile assignment, this claim for PheRS superiority should be buttressed with comparable analysis of the sensitivity of PheRS assessment to drop-out of codes. How confident should a person be that they are truly in the top 10% decile, and what are the consequences for estimating the benefit to stratification? Relatedly, in addition to reporting improved c-statistics, some quantification of precision-recall for PheRS would add to the study.

Minor points:

Readers new to PheRS would probably benefit from more discussion of how filtering is performed to ensure that the Phecodes are not just independent codes for the disease. For example, is impaired glucose tolerance a predictor of diabetes, or hypertension a predictor of CAD? What does the Charlson Comorbidity Index capture that Phecodes do not? I realize that space is limited, but perhaps a supplementary methods explaining this would be useful.

The first paragraph of the Introduction concludes with references to 15 studies that debate the utility of PRS. This is arbitrary – I could easily list another 15 as relevant – and hence exclusionary as well, just one or two key reviews would suffice.

Number of codes was correlated with PheRS values in FinnGen. Was this not the case in the other two studies, or was the same analysis not performed?

In Figure 4D, please provide an intuition as to why the majority of the scaled PheRS coefficients are negative?

Since this is a Nature Genetics submission and there is actually very little genetic analysis reported, I would recommend bringing the results in Suppl Fig 7 into the main paper (perhaps swap with Figure 3 if needed)

Reviewer #2 (Remarks to the Author):

Detroit and Hartonen et al provide a multi-cohort study on the ability of clinical phenotype codes (phecodes) to predict disease. This is a clever use of electronic health record data and allow comparisons of generalizability between cohorts as well as comparisons to risk scores solely based on genetics, e.g. standard polygenic risk scores. Overall, the authors find that risk scores based on phecodes (pheRS) are more predictive of disease than baseline (a model with age and sex) and that these scores explain orthogonal information from genetic PRS.

I have a few questions and requests for clarifying analyses that I believe would strength the paper and derived insights.

Major comments:

- 1) The terms generalizability and transferability were used to assess whether pheRS trained in one European cohort (such as FinnGen) are predictive of disease in a different European cohort (such as the Estonian biobank). In PRS literature, transferability typically means across different ancestries, rather than healthcare systems, which are both important. However, for the purpose of this paper, I feel that generalizability and transferability are simply required for a "good" model that is not overfit. It's a different thing to say a European model transfer well to another population, because you wouldn't necessarily expect that due to evolutionary differences in linkage and allele frequency etc. So similarly, in this manuscript, you would expect transferability of a risk score built in one European healthcare system to another. Therefore, I think the use of the terms generalizability and transferability need to be toned down to simply say that the authors tested for out-of-sample prediction to assess if the original model was overfit or not (which is a standard requirement for model assessment).
- 2) In the latter half of the manuscript, the authors compare the prediction of a pheRS with a genetic PRS and conclude that the two scores provide orthogonal information. This analysis leaves the reader (especially if they are a population geneticist) with several questions. First, how predictive were the genetic PRS to start? Standard PRS from pruning and thresholding have notoriously low accuracy, and although the authors use a newer method (<https://www.nature.com/articles/s41467-021-24485-y>), it would be very helpful to report how predictive the PRS were in pseudo-R2 values, such that the authors can further compare this to the SNP-based heritability of the phenotype from those recent GWAS. It's impossible to interpret that from the hazard ratios in Figure 5A. With pseudo-R2 values representing predictive accuracy and the heritability (theoretical max of the predictive accuracy), it would be more straightforward to interpret the relationship with pheRS. How much variance of the PRS can be explained by the pheRS? While this is shown in Figure 5B, there is no reference point for how predictive the PRS were in the first place, in terms of variance (an R2 or pseudo-R2, which would be on a linear scale with the correlation presented in Figure 5B).
- 3) 242 phecodes (as mentioned on page 5) seems like a small number. How many did the authors begin with and was the only criterion for selection that the phecode be in at least 1% of any study? It's especially small when considering that only 20% of these were present in all of the compared cohorts.
- 4) If sex, age, and genotyping PCs were regressed out of the pheRS and PRS, then I think Figure 2B should have "C-index: sex + age + genotyping PCs" on the x-axis. Similarly need to add geno PCs to the y-axis.
- 5) In Figure 2B, there is a very marginal improvement of the pheRS compared to the baseline model. While the authors report that 7/13 of the diseases were significantly better at $p < 0.05$ compared to baseline, did they perform multiple hypothesis correction? It looks like MDD and Gout may be the only ones to survive.
- 6) The authors state that pheRS were strongly correlated with the number of phecodes an individual had listed. This is concerning and makes me wonder if this is biological (and truly predictive) or an artifact of the dataset (such as some doctors are more descriptive and prescribe more phecodes even if redundant). Is the number of listed phecodes a covariate that the authors should have in the baseline model as well? Especially while different cohorts have different phecodes available? How would Figure 2B change if the x-axis baseline model was age + sex + geno PCs + number of phecodes per individual? Moreover, was there a systematic difference between number of phecodes for one cohort vs another?
- 7) In Figure 2B, another reason why the pheRS might perform better is simply the addition of an extra feature. Therefore, what would the figure look like if the authors reported the adjusted R2 between the baseline model and the pheRS model? Does the c-index account for multiple predictors like an adjusted R2 does?

8) Figure 3A would visually benefit from adding error bars on the correlations.

9) I was hoping that the authors did more to deal with the fact that there are different phecodes available (or found with at least 1% prevalence) in different cohorts. What if the authors imputed the "missing" phecodes for other cohorts (since they are highly correlated anyway) such that they have data for all 242 phecodes for each individual of each cohort. This might help the interpretation of the finding that the number of recorded phecodes was highly predictive of the pheRS and would allow some standardization. This might also help out-of-cohort prediction.

10) Regarding out of cohort prediction, when sparsity-enforcing regression like lasso/enet picks 1 of several highly correlated phecodes as a predictive feature for cohort A and that specific phecode isn't recorded or isn't prevalent in cohort B, how do the authors handle this? This is obviously going to make the out of cohort prediction take a hit and immediately restricts transferability across healthcare systems in a needlessly random way. Related to the above, imputation might help here.

10) Figure 4D could be misleading because phecodes are highly correlated with one another, and the authors used feature selection models that randomly chose between correlated features (e.g. lasso/enet). Therefore, I'm wondering, if you remove each of the top predictors, one at a time in a leave-one-out analysis, how does it affect model performance? Does it simply identify other phecodes that are correlated and predictive? Specifically, I'm interested in whether "sleep disorders" is really the most predictive feature, or if it's just a proxy for something else being captured by other phecodes?

11) On page 10, the authors say that pheRS and PRS associations were both significantly associated with disease in the meta-analysis. Was multiple hypothesis testing correction applied? If no, why not?

Minor comments:

There are some typos throughout the paper that if fixed would improve the presentation of the writing and the findings.

- 1) Page 1 last line: "the PGS for 8/13 disease", needs an s.
- 2) Page 2: "All of US", the s should be lowercase.
- 3) Page 2: values such as 1.347 and 1.568 should be presented with commas.
- 4) Figure 1: $-\log_{10}$ p-value on the y-axis of manhattan plot is upside down, and would look better right side up.
- 5) Please define "washout period" used on page 5
- 6) Page 6: "4/13 disease", needs an s.
- 7) Page 7: "moderate to strongly", should be moderately.

Reviewer #2 (Remarks on code availability):

The code provides a resource for recreating the INTERVENE PheRS, including a tutorial, data / dummy data, and documentation on the inputs and different analyses (including downsampling). The instructions provided are sufficient to install and run the application.

Version 1:

Decision Letter:

Our ref: NG-A66655R

29th Apr 2025

Dear Dr. Ganna,

Thank you for submitting your revised manuscript "Trans-biobank generalizability and accuracy of electronic health record-based predictors compared to polygenic scores" (NG-A66655R). It has now been seen by the original referees and their comments are below. The reviewers find that the paper has improved in revision, and therefore we'll be happy in principle to publish it in Nature Genetics, pending minor revisions to comply with our editorial and formatting guidelines.

Sincerely,
Wei

Wei Li, PhD
Senior Editor
Nature Genetics

Reviewer #1 (Remarks to the Author):

Thank you for your comprehensive response to the reviews.

Reviewer #2 (Remarks to the Author):

The authors provide a substantially revised manuscript with improved statistics, comparisons (and baselines), and new and/or updated figures in the main text and supplement. I find that the authors have been maximally responsive to both reviewers, and I support the few times they pointed out that reviewer concerns were beyond the scope of the current study, yet would be interesting as a future research direction. At this time, I have no remaining concerns about the work. Thank you for addressing all of my comments and concerns.

Editorial comments

Comment: To guide the scope of the revisions, the editors discuss the referee reports in detail within the team with a view to identifying key priorities that should be addressed in revision. In this case, we think both referees have provided constructive reviews aimed at strengthening the analyses and improving the presentation. We particularly ask that you enhance the comparison of PheRS and PGS, and address all referee comments as thoroughly as possible with appropriate revisions. We hope that you will find the prioritized set of referee points to be useful when revising your study.

Response: We thank the editorial team and the reviewers for their thoughtful comments and suggestions. We have thoroughly addressed all reviewer comments in the revised version of the manuscript, and detail our responses and the corresponding changes made to the manuscript below in our point-by-point responses. Specifically, as requested by the editor and following the reviewer comments, we have enhanced the comparison of PheRS and PGS by:

- Quantifying the increase in c-index when integrating PheRS to a model already including age, sex and PGS (new Figure 5C). This analysis shows that the inclusion of PheRS on top of a model with PGS increases disease prediction, highlighting the independence between the two scores.
- Comparing the correlation of PGS and PheRS as a function of how predictive the PGS are (new Figure 5B). The hypothesis is that if the low correlation between PheRS and PGS were merely due to both scores being poor predictors, we would expect higher correlation for diseases where the PGS performs well. However, this is not observed, indicating that the correlation between PheRS and PGS is independent of the predictive performance of the PGS.
- Showing that the hazard ratios for the association between PheRS and diseases do not change when adjusting for the PGS, further supporting the independent effect of the two scores (new Figure 5A-2).

Figure 5: Comparison of PGS and PheRS.

Panel A: 1 - Association of PGS (x-axis) and PheRS (y-axis) scores with each disease. The meta-analyzed HRs (95% CI) for the top 10% at risk compared to those in the average 20% risk group (40%-60% percentile) based on the scores after regressing out age, sex, and the first 10 PCs. **2** - PheRS HRs per 1-SD increase in a model without PGS (x-axis) and one where we adjust for the effect of PGS (y-axis). **Panel B:** Correlation (Persons' r , y-axis) between the PheRS and PGS scores separately in each study (FinnGen: pyramid, UKB: square, EstB: circle), with the c-index of the PGS model as reference with the effect of age, sex, and the first 10 PCs regressed out from the PGS (x-axis). **Panel C** - C-index improvements when adding PheRS to models with PGS. C-index of the models with only PGS, sex, age, and PCs (x-axis) compared to those with added PheRS (y-axis). Due to sample overlap with the GWASs, PGS could only be calculated for 4 diseases in UKB (see Methods for details). Supplement Figure 9B shows the correlations in relation to Nagelkerke's pseudo- R^2 . Diseases with significant differences are labelled (p-values < 0.05; two-tailed p-values based on the z-scores of the HR differences and one-tailed p-values for the c-index increases) and those passing multiple hypothesis testing (p-values < 0.05/13) are marked with *.

Moreover, we have performed new analyses to answer reviewers' comments:

- We compared the PheRS to a new predictor based on the number of unique phecodes recorded for each individual.
- We trained new consensus PheRS only on phecodes that were common in all studies (prevalence of at least 1%) to see how this affects the generalizability of the models.
- We added new results where we transferred the EstB-trained PheRS also to the UKB.
- We performed a leave-one-out analysis in FinnGen where we separately removed each of the 20 most important predictors from the models for each disease.

Changes are detailed in our following point-by-point response to the reviewer comments:

Reviewer #1

Remarks to the Author

Comment: This is an interesting and timely study on the transferability of EHR Phecode-based predictors relative to polygenic scores across three European studies. The key conclusions are that transferability is quite good, and that PheRS are mostly orthogonal to PGS and hence integrated scores are likely better predictors. I do not find any major issues with the analysis or presentation, but do feel that there are a couple of questions that could be answered that would be within scope of the paper. One is that the study does not directly address the reason for the overall lower performance of PheRS in the UKB relative to EstB and FinnGen. The other is that the repeatability and robustness of PheRS is not quantified.

Response: We thank the reviewer for their interest in our work and for their constructive comments. We detail below our responses to the specific comments and have answered to the specific concern mentioned in this comment in Response to comment 1, below.

Comment #1

Comment: Regarding the first concern, it is notable in Figure 2A that UKB HRs are consistently the lowest, particularly for the non-cancers. In places the authors hint that this might be due to the balance of primary and secondary records in the UK versus Finland and Estonia. It is an important point, particularly in light of emerging evidence that heritability in AllofUS is consistently lower, almost half, than it is in matched populations in the UKB (for example: <https://www.biorxiv.org/content/10.1101/2024.08.06.606846v1>) whereas the authors cite a study [ref 31] showing that PheRS transferability is good from UKB to AllofUS in the sense of improving risk prediction – though no relative performance was mentioned.

Response: The reviewer raises an important point in noting that the UKB PheRS HRs are clearly lower than in FinnGen and the EstB for some diseases (exactly 3/9 of the non-cancers: gout, asthma and knee osteoarthritis). However, we think it is difficult to generalise that the UKB PheRS performance is overall worse than in the other two cohorts, since there are also examples where the performance is similar to other studies, or even clearly better in the case of major depression. Nevertheless, we think the lower performance of the UKB PheRS for some diseases is related to how well we can capture the diseases in the different cohorts. First, in FinnGen the registers cover a longer period in the past than in the UKB meaning that, at baseline, we can exclude individuals that have already had a certain diagnosis with more certainty. We would expect this to have a bigger effect on diseases that are usually diagnosed at a relatively young age like asthma, or chronic conditions affecting individuals for a long period of time, like knee osteoarthritis. The Finnish inpatient registries are also generally known to be of very high quality (Sund, 2012, PMID: 22899561). Second, the fact that the EstB cohort utilizes primary care data is advantageous compared to UKB both in detecting previous diagnoses before the baseline date, but also in detecting correctly the first occurrence of diseases typically diagnosed in primary care, such as asthma. The relative completeness of phenotyping in the three studies is also reflected in the

phecode prevalences in our study population: in UKB 85 phecodes have prevalence of >1%, while the number is 129 in FinnGen and 185 in EstB, respectively (Figure 4A). We have clarified this in the Discussion section of the revised manuscript:

For asthma, knee OA, and gout the PheRS were less predictive in the UKB compared to the other cohorts. This could be reflecting the richer health information available in FinnGen with longer register coverage, and in EstB with inclusion of primary care data, both allowing more accurate phenotyping and detection of diagnoses before the baseline date.

It is worth noting that the reason the primary care data was not used for FinnGen and UKB was its incompleteness: In FinnGen, primary care records are available only starting 2011, while in the UKB the primary care data coverage is currently partial both in terms of population and temporally (see: <https://www.ukbiobank.ac.uk/enable-your-research/about-our-data/health-related-outcomes-data>).

Comment #2

Comment: In any case, it is also not clear from the description how external training using internal model parameters deals with the fact that many of the Phecodes are missing in external datasets – do these just drop from the model, and is that the reason for drop in performance?

Response: We thank the reviewer for pointing out that the description of how externally-trained PheRS were computed in FinnGen was not clear enough. PheRS for a given disease uses the same set of features in all studies, and when a specific phecode is not observed in the FinnGen test set, the corresponding coefficient of the externally-trained PheRS is multiplied by zero. This can be part of the reason for drop in performance. We have now clarified this by adding the following to the Results of the revised manuscript:

All PheRS models for a given disease share the same phecode predictors. When a phecode is not observed in a study, but has a non-zero coefficient in a model trained in an external study, the coefficient is multiplied by zero and does not affect the prediction.

Comment #3

Comment: Is it possible to perform down-sampling of the included Phecodes to either include the same number in each study, or preferably just the same Phecodes, such that relative performance is then mostly a function of prevalence and comorbidity?

Response: We thank the reviewer for this interesting suggestion. We have included a sensitivity analysis in the revised version of the manuscript, where we train additional consensus PheRS models across the studies so that only the codes common in all three cohorts are used as features. This analysis is presented in the new Supplement Figure 7, with the following description in the Supplement Results:

Supplement Figure 7. In this sensitivity analysis, we train additional consensus PheRS models across the studies with only phecodes common (prevalence of at least 1%) to all three studies. When considering internal testing, we see a significant decrease in performances for 7/13, 13/13 and 8/13 models in FinnGen, UKB and EstB, respectively, with only one minor improvement for breast cancer in the UKB. However, interestingly, when considering external testing, we find a stronger association for 8/13 diseases in FinnGen with the EstB-trained models and only for 2/13 diseases a weaker association. This is likely explained by the similarity of the EstB and FinnGen data, with the main difference being the wider availability of

primary care data in EstB compared to FinnGen. For the other models, the changes are comparatively minor. Performance in the external test set decreases significantly for epilepsy and gout when testing the UKB-trained models in FinnGen, for asthma and gout when testing the EstB-trained models in FinnGen and for knee OA, epilepsy and lung cancer when testing the EstB-trained models in UKB.

Association of PheRS trained all phecodes vs. only common phecodes with each disease

A - internally-tested

B - externally-tested

Supplement Figure 7

Association of PheRS trained on all phecodes in a study (x-axis) vs. PheRS trained only on phecodes common in all three studies (y-axis). Phecodes are considered common if they are recorded for at least 1% of the population during the observation period. **A** - PheRS trained and tested in the same study. **B** - PheRS trained and tested in different studies. The first and second plot show the PheRS tested in FinnGen and trained in the UKB and EstB. The third plot shows the PheRS tested in the UKB and trained in the EstB. Diseases with a significant difference are labelled (two-tailed p-values < 0.05 based on the z-scores of the HR differences) and those passing multiple hypothesis (p-value < 0.05/13) testing are marked with *.

Comparing consensus PheRS with the original pheRS, we observed a significant decrease in the within-study PheRS performances for 7/13, 13/13 and 8/13 models in FinnGen, UKB and EstB, respectively, with only one minor improvement (breast cancer in the UKB). However, we see improvement or comparable performances when considering external validation. Here we found a stronger association of the consensus PheRS for 8/13 diseases in FinnGen with the EstB-trained models and only for 2/13 diseases a weaker association. For the remaining models, the differences were minor. This observation is likely explained by the similarity of the EstB and FinnGen data, with

the main difference being the broader availability of primary care data in EstB compared to FinnGen. We have also added the following description of this analysis to the Results section:

Notably, we found that training PheRS only using codes common (prevalence of at least 1%) to all three studies increases PheRS generalizability from EstB to FinnGen (Supplement Figure 7).

Comment #4

Comment: Relatedly, are the results reported in Figure 3B replicated when applied to predicting disease in UKB?

Response: We appreciate the reviewer's interest in the external validation results across studies. We were able to extend Figure 3B by including the transfer of models from EstB to UKB. This added comparison highlights that the models for gout, hip osteoarthritis (OA), and knee OA trained in UKB and EstB appear very similar. Both demonstrate strong associations in FinnGen and no significant differences in HR in UKB when between the UKB- and EstB-trained models. Unfortunately, due to FinnGen's strict regulations we could not include the transfer of models from FinnGen to UKB.

When looking at the correlation between internally- and externally-trained PheRS (Figure 3A), for some diseases (Knee OA, Gout, and T2D) the correlation is very similar in all three external validation examples. Whereas for others, such as lung cancer, asthma, and AF the correlation of the EstB-trained models transferred to the UKB show markedly lower (>0.2 difference) correlations compared to the FinnGen validation.

We have updated Figure 3 and the Figure caption, as well as the original Supplement Figure 4 (now 6) which shows the c-index changes comparing the internal- and externally-trained PheRS to a baseline. Additional changes to the Figures are due to only regressing out age and sex from the PheRS at this step whereas previously we also removed the PCs from all PheRS due to comment #4 by reviewer #2. This led to slightly lower HRs for the transferred PheRS models for MDD, AF, and hip OA.

We updated the respective section in the results with

In the EstB, the HRs for hip OA, Gout, and knee OA were not significantly worse when trained in the UKB compared to the PheRS models trained in the EstB (Figure 3B-2). Further, for the PheRS models for hip OA transferred to FinnGen and the UKB, as well as, knee OA, AF, Gout, and prostate cancer PheRS trained in the EstB and transferred to the UKB, c-index improvements were not significantly different from those achieved by the internally-trained PheRS (Supplement Figure 6).

Figure 3: PheRS external validation.

Panel A: Correlations (Persons' r) between the internally- and externally-trained PheRS. Partial correlations and 95% CI (x-axis) of the PheRS (y-axis) after regressing out the effect of age and sex, with the c-index of the internally-trained and -tested PheRS as reference (x-axis). All PheRS were trained on 50% of individuals. **Panel B:** Association of internally-trained PheRS with each disease compared to the externally-trained models. HRs and 95% CIs of the FinnGen-trained PheRS (x-axis) vs. the externally-trained PheRS (y-axis). **1** - shows the models transferred to FinnGen with EstB on the left and UKB on the right and **2** - the PheRS trained in the EstB and tested in the UKB. The HRs are shown for a 1-SD increase of the PheRS after regressing out age and sex. Diseases with no significant differences are labelled (two-tailed p-values < 0.05 based on the z-scores of the HR differences) and those passing multiple hypothesis testing (p-value < 0.05/13) are marked with *.

Supplement Figure 6 shows the c-index improvements of the externally-trained PheRS over age and sex in FinnGen and the UKB, compared to the internally-trained PheRS. With the UKB-trained models we see significant improvements for 8/13 diseases (nominal p-value < 0.05; asthma, epilepsy, T2D, hip OA, knee OA, MDD, CHD, and Gout) over baseline and for hip OA the improvements were not significantly different to those achieved by the FinnGen-trained PheRS. With the EstB-trained models we see significant improvements for 5/13 diseases (p-value < 0.05; asthma, knee OA, T2D, MDD, and gout) over baseline in FinnGen, with also only the hip OA improvements not significantly different to the FinnGen-trained PheRS. Notably, the EstB-trained models tested in the UKB showed improvements that were not statistically different to the UKB-trained PheRS for 5/13 (p-value < 0.05; T2D, hip OA, knee OA, AF, gout, and prostate cancer). For 10/13 the models significantly increased the c-index over the baseline model (nominal p-value < 0.05).

Supplement Figure 6

Improvement in predictive accuracy in FinnGen and UKB with internally-trained and externally-trained PheRS models. The c-index and 95% CI of the baseline model with age and sex and the model with added PheRS. The c-index of the baseline models is shown by the dark grey boxes (the leftmost marker for each disease). The results of the PheRS models trained internally in FinnGen (left and middle plot) and the UKB (right most plot) are shown as boxes with the color of each disease and the externally-trained PheRS as stars. The left panel shows the results with the UKB-trained and the right panel the EstB-trained models. A single star denotes diseases where the c-index increase of the externally-trained PheRS is significantly higher than the baseline and two stars where this increase is not significantly different from the internally-trained PheRS.

Comment #5

Comment: Regarding repeatability, the authors make the interesting observation in Figure 5 that HR comparing the top 10% to average 20% (also please define this term – is it the middle 40%-60% decile or a permuted average) is notably greater for PheRS than PGS. Given Ding et al's findings (<https://pubmed.ncbi.nlm.nih.gov/34931067/>) regarding large uncertainty in PGS decile assignment, this claim for PheRS superiority should be buttressed with comparable analysis of the sensitivity of PheRS assessment to drop-out of codes. How confident should a person be that they are truly in the top 10% decile, and what are the consequences for estimating the benefit to stratification?

Response: The average 20% does indeed refer to the middle 40%-60% decile, and we have now clarified this in the Results section of the revised manuscript. In addition to the results presented in Figure 5A, we have presented the PheRS HRs also for 1-SD in Supplement Figure 9A-1, which we also referenced in the Results. In this context, it was interesting to see that the disease association of the PheRS for the most at risk individuals was more pronounced than with the averaged 1-SD increase.

We thank the reviewer for pointing out the work of Ding et al and think it would be an interesting approach to further develop for PheRS. However, since the approach would have to be separately adapted to the methodology of our PheRS, we think that exploring this approach is beyond the scope of this paper. Nonetheless, we have highlighted the challenges in the interpretation of PheRS, similar to PGS, and highlighted the work of Ding et al. as a possible solution for these challenges.

However, while we used the same methodology for the percentile assignment for the PheRS and PGS, prior work on the PGS by Ding et al.⁴⁶ has shown that there can be large uncertainties in the assignment of an individual to a risk group. Similar comprehensive robustness analysis of PheRS percentile assignment is an interesting direction for future work.

Comment #6

Comment: Relatedly, in addition to reporting improved c-statistics, some quantification of precision-recall for PheRS would add to the study.

Response: We thank the reviewer for their suggestion, and have now added quantification of Area Under Precision-Recall Curve (AUPRC) for selected models as a new Supplement Table 5. We note that the AUPRCs are computed for the full 8 year survival. We find that the AUPRCs for the models with PheRS+age+sex are larger than those of the baseline model with only age and sex for all diseases in all studies, except for breast cancer in the EstB. Compared to the PGS, the AUPRCs of the PheRS models is larger for asthma, epilepsy, gout, knee OA, lung cancer, and MDD in FinnGen (6/13),

for epilepsy in the UKB (1/4), and for asthma, epilepsy, gout, hip OA, knee OA, lung cancer, and MDD in the EstB (7/13). Results are generally consistent with what we observed for c-indices and HRs.

Minor comment #1

Comment: Readers new to PheRS would probably benefit from more discussion of how filtering is performed to ensure that the Phecodes are not just independent codes for the disease. For example, is impaired glucose tolerance a predictor of diabetes, or hypertension a predictor of CAD?

Response: We thank the reviewer for pointing out that the description of how phecodes were filtered would benefit from expanding. We have now added the following text to the Results section, respectively, with an example for T2D of what codes we exclude in prediction:

However, for each disease we excluded closely related diagnoses as predictors based on the phecodes exclusion ranges (see Methods, Supplement Table 14). For example, we did not use phecodes for secondary diabetes, T1D, or abnormal glucose tolerance as predictors of T2D.

And expanded the corresponding description in the Methods section:

We gathered all phecodes in their 3-digit parent node in the phecode ontology, for example, *type 1 diabetes* (250.1), *type 2 diabetes* (250.2), and *type 2 diabetes with ketoacidosis* (250.21) were all mapped to the same phecode *diabetes mellitus* (250). For each disease, we separately excluded predictors that were part of the exclusion range of the phecodes (Supplement Table 14), for example, for T2D we did not use *secondary diabetes* (phecode 249), *diabetes mellitus* (250), and *conditions complicating pregnancy* (649) as predictors. The phecode *conditions complicating pregnancy* was excluded because it was the parent node of the phecode *Diabetes or abnormal glucose tolerance complicating pregnancy* (649.1) which is in the exclusion range of the phecode for T2D (250.2). We only considered phecodes with a prevalence at least 1% of the study population (Supplement Table 9).

We base the exclusion of predictors on the phecode exclusion ranges that have been crafted in collaboration with clinical experts by the authors of the phecode system. Thus, for example our models for CAD, do use hypertension as a predictor. Instead we exclude various forms of conductive disorders and other circulatory diseases, the list of which is a bit too long to mention in the main text but can be found in Supplement Table 14.

Minor comment #2

Comment: What does the Charlson Comorbidity Index capture that Phecodes do not? I realize that space is limited, but perhaps a Supplement methods explaining this would be useful.

Response: For most diseases and across the three cohorts, the Charlson Comorbidity Index (CCI) does not add any predictive information when added on top of age and sex, as shown in Supplement Figure 3A of the revised manuscript (Supplement Figure 2A in the original submitted manuscript). On the other hand, PheRS improves over CCI for most of the diseases in each of the cohorts (Supplement Figure 4A). Essentially, CCI is a weighted sum of a limited number of existing diagnoses, with the weights predetermined, while PheRS is a weighted sum of all diagnoses, with the weights determined by training in the cohort of interest. CCI can be viewed as a much simpler and general (not disease-specific) version of PheRS that captures overall morbidity. Description of how CCI was constructed was missing from the original submitted manuscript, and we have now added this to the Supplement Methods, following the reviewers suggestion:

The Charlson comorbidity index (CCI²) assigns fixed weights to a set of comorbid conditions which increase an individual's relative risk of dying. The original version had 19 comorbidity categories, and later Deyo et al.³ combined the malignancies leukaemia and lymphomas into a single category reducing it to 17 categories. The weights were based on the estimators of a Cox-PH model, adjusted for other diseases, illness severity, and the hospital admission reason. The final CCI score of an individual is the sum of all of their comorbidity weights. For our analyses, we used the adaptations of the CCI to ICD-9 and ICD-10 codes using the *comorbidity* package in R. The code can be found at <https://github.com/dsgelab/ICCI>.

We have also clarified the comparison between PheRS and CCI in the Supplement Results:

As the CCI is a weighted sum of selected existing conditions (see Supplement Methods), it is to be expected that PheRS, being essentially a weighted sum of all existing conditions but where weights are learned from the data, outperforms CCI in predicting disease outcomes.

Minor comment #3

Comment: The first paragraph of the Introduction concludes with references to 15 studies that debate the utility of PRS. This is arbitrary – I could easily list another 15 as relevant – and hence exclusionary as well, just one or two key reviews would suffice.

Response: We thank the reviewer for pointing this out and agree with the notion. We have now pruned this list of references to a single recent review article:

9. Xiang, R. *et al.* Recent advances in polygenic scores: translation, equitability, methods and FAIR tools. *Genome Med.* **16**, 33 (2024).

Minor comment #4

Comment: Number of codes was correlated with PheRS values in FinnGen. Was this not the case in the other two studies, or was the same analysis not performed?

Response: We have now performed the same analysis in all three cohorts (see updated Supplement Figure 1). The magnitude of correlation varies, but we do observe a correlation in all three studies, albeit for prostate cancer the correlation is very low in UKB and EstB. We have updated the reference to this result in the main text accordingly:

We found that all PheRS were correlated, mostly positively, with the total number of phecodes an individual had recorded (Pearsons' *r* ranging from 0.82 for asthma in FinnGen to -0.66 for breast cancer in FinnGen; Supplement Figure 1C).

A - Prediction improvements with PheRS over baseline in each study

1 - age and sex

2 - age, sex, unique number of diagnoses

B - Prediction accuracy of the PheRS

C - Correlation of PheRS with the number diagnoses

Supplement Figure 1

Panel A: 1 - Increase in prediction accuracy when adding the PheRS to a baseline model with age and sex in all three studies. The c-index and 95% CIs of the baseline model (x-axis), compared to a model with added PheRS (y-axis). **2** - Increase in prediction accuracy when adding the PheRS to a baseline model with the additional predictor of unique number of phecodes/diagnoses. Diseases with significant ($p < 0.05$) increases in a study are labeled and those passing Bonferroni correction ($p < 0.05/13$) are marked with a star. **Panel B:** The c-index of the PheRS model and 95% CIs separately in each study (FinnGen: yellow, UKB: brown, EstB: green) and meta-analyzed results (red). The effect of age and sex was removed by regressing them out from the PheRS. **Panel C:** Correlation between the PheRS and the total number of unique phecodes of each individual (FinnGen: yellow, UKB: brown, EstB: green). Pearson's r and 95% CI (x-axis) for each disease (y-axis).

Minor comment #5

Comment: In Figure 4D, please provide an intuition as to why the majority of the scaled PheRS coefficients are negative?

Response: We apologize for unclear description of the coefficient scaling in the original submitted version of the manuscript. We have scaled the coefficients to 0 mean and 1 standard deviation for each model separately for easier comparison of the importances of the phecode predictors across the studies. This normalization will naturally force approximately half of the coefficients to have negative scaled values. We have clarified the scaling issue in the revised version of the manuscript by adding the following to the caption text of Figure 4D:

Note that this normalization will force approximately half of the coefficients to have a negative scaled value with respect to the mean at zero.

Minor comment #6

Comment: Since this is a Nature Genetics submission and there is actually very little genetic analysis reported, I would recommend bringing the results in Suppl Fig 7 into the main paper (perhaps swap with Figure 3 if needed)

Response: We agree with the reviewer and have now moved some of the results shown in Supplement Figure 7 of the original submitted manuscript to an updated Figure 5 of the revised manuscript. We also improved the comparison between PheRS and PGS showing that the correlation between PheRS and PGS is not a function of how predictive the PGS (or PheRS) is (Figure 5B). Additionally, we added a Figure 5A-2 showing that the HRs of the PheRS do not change when adding PGS to the model. The new Supplement Figure 8 is the mirror image of the results of Figure 5A-2

focused on the PGS.

Figure 5: Comparison of PGS and PheRS.

Panel A: 1 - Association of PGS (x-axis) and PheRS (y-axis) scores with each disease. The meta-analyzed HRs (95% CI) for the top 10% at risk compared to the average 20% based on the scores after regressing out age, sex, and the first 10 PCs. **2 -** PheRS HRs per 1-SD increase in a model without PGS (x-axis) and one where we adjust for the effect of PGS (y-axis). **Panel B:** Correlation (Persons' r , y-axis) between the PheRS and PGS scores separately in each study (FinnGen: triangle, UKB: square, EstB: circle), with the c-index of the PGS model as reference with the effect of age, sex, and the first 10 PCs regressed out from the PGS (x-axis). **Panel C -** C-index improvements when adding PheRS to models with PGS. C-index of the models with only PGS, sex, age, and PCs (x-axis) compared to those with added PheRS (y-axis). Due to sample overlap with the GWASs, PGS could only be calculated for 4 diseases in UKB (see Methods for details). Diseases with significant differences are labelled (p-values < 0.05; two-tailed p-values based on the z-scores of the HR differences and one-tailed p-values for the c-index increases) and those passing multiple hypothesis testing (p-values < 0.05/13) are marked with *.

Reviewer #2

Remarks to the Author

Comment: Detroit and Hartonen et al provide a multi-cohort study on the ability of clinical phenotype codes (phecodes) to predict disease. This is a clever use of electronic health record data and allow comparisons of generalizability between cohorts as well as comparisons to risk scores solely based on genetics, e.g. standard polygenic risk scores. Overall, the authors find that risk scores based on phecodes (pheRS) are more predictive of disease than baseline (a model with age and sex) and that these scores explain orthogonal information from genetic PRS.

I have a few questions and requests for clarifying analyses that I believe would strength the paper and derived insights.

Response: We thank the reviewer for their interest in our study and for the insightful comments.

Comment #1

Comment: The terms generalizability and transferability were used to assess whether pheRS trained in one European cohort (such as FinnGen) are predictive of disease in a different European cohort (such as the Estonian biobank). In PRS literature, transferability typically means across different ancestries, rather than healthcare systems, which are both important. However, for the purpose of this paper, I feel that generalizability and transferability are simply required for a "good" model that is not overfit. It's a different thing to say a European model transfer well to another population, because you wouldn't necessarily expect that due to evolutionary differences in linkage and allele frequency etc. So similarly, in this manuscript, you would expect transferability of a risk score built in one European healthcare system to another. Therefore, I think the use of the terms generalizability and transferability need to be toned down to simply say that the authors tested for out-of-sample prediction to assess if the original model was overfit or not (which is a standard requirement for model assessment).

Response: We thank the reviewer for raising this point and agree that "transferability" was not the most appropriate term in this context. However, we find that the term "generalizability" still appropriately captures the meaning, because the healthcare systems and underlying population characteristics between the studies vary quite substantially. To address this, we changed all occurrences of "transferability" to "generalizability" throughout the manuscript. Additionally, we clarified our definition of "generalizability" with the following sentence in the manuscript:

In this context, we define generalizability as the extent to which models trained in one setting (e.g. a specific study or population) maintain their predictive accuracy and associations when applied to another, previously unseen context (e.g. a different study or population).

We also updated the manuscript title to reflect this change:

Trans-biobank generalizability and accuracy of electronic health record-based predictors compared to polygenic scores

Comment #2

Comment: In the latter half of the manuscript, the authors compare the prediction of a pheRS with a genetic PRS and conclude that the two scores provide orthogonal information. This analysis leaves the reader (especially if they are a population geneticist) with several questions. First, how predictive were the genetic PRS to start? Standard PRS from pruning and thresholding have notoriously low accuracy, and although the authors use a newer method (<https://www.nature.com/articles/s41467-021-24485-y>), it would be very helpful to report how predictive the PRS were in pseudo-R2 values, such that the authors can further compare this to the SNP-based heritability of the phenotype from those recent GWAS. It's impossible to interpret that from the hazard ratios in Figure 5A. With pseudo-R2 values representing predictive accuracy and the heritability (theoretical max of the

predictive accuracy), it would be more straightforward to interpret the relationship with pheRS. How much variance of the PRS can be explained by the pheRS? While this is shown in Figure 5B, there is no reference point for how predictive the PRS were in the first place, in terms of variance (an R2 or pseudo-R2, which would be on a linear scale with the correlation presented in Figure 5B).

Response: We thank the reviewer for their insightful comments and agree that providing additional metrics on the predictive performance of the PRS would help contextualize the relationship between PRS and PheRS. We chose to focus on c-index, to stay consistent with the other results presented in the manuscript but also added a new Supplement Figure 9 with the Nagelkerke's pseudo-R² as a reference and a Supplement Table 6 reporting the pseudo-R² of the models.

To generally improve the interpretability of the correlations between PheRS and PGS, we have compared the correlation between PheRS and PGS as a function of the PGS c-index (new Figure 5B). The hypothesis was that if the low correlation between PheRS and PGS were merely due to both scores being poor predictors, we would expect higher correlation for diseases where the PGS performs well. However, this was not observed, indicating that the correlation between PheRS and PGS is independent of the predictive performance of the PGS. If anything, we observed a tendency for the correlation to be slightly higher with lower predictiveness of the PGS.

Figure 5: Comparison of PGS and PheRS.

Panel A: 1 - Association of PGS (x-axis) and PheRS (y-axis) scores with each disease. The meta-analyzed HRs (95% CI) for the top 10% at risk compared to those in the average 20% risk group (40%-60% percentile) based on the scores after regressing out age, sex, and the first 10 PCs. **2** - PheRS HRs per 1-SD increase in a model without PGS (x-axis) and one where we adjust for the effect of PGS (y-axis). **Panel B:** Correlation (Person's *r*, y-axis) between the PheRS and PGS scores separately in each study (FinnGen: triangle, UKB: square, EstB: circle), with the c-index of the PGS model as a reference

with the effect of age, sex, and the first 10 PCs regressed out from the PGS (x-axis). **Panel C** - C-index improvements when adding PheRS to models with PGS. C-index of the models with only PGS, sex, age, and PCs (x-axis) compared to those with added PheRS (y-axis). Due to sample overlap with the GWASs, PGS could only be calculated for 4 diseases in UKB (see Methods for details). Diseases with significant differences are labelled (p-values < 0.05; two-tailed p-values based on the z-scores of the HR differences and one-tailed p-values for the c-index increases) and those passing multiple hypothesis testing (p-values < 0.05/13) are marked with *.

Supplement Figure 9. Both PGS and PheRS add information on top of the other predictor, age, and sex. Due to sample overlap with the GWASs, PGS could only be calculated for 4 diseases in UKB (see Methods for details). The HRs of the PGS per 1-SD were significantly larger than those of the PheRS for 5/13 diseases in the meta-analysis (nominal p-value<0.05; T2D, AF, colorectal cancer, breast cancer, prostate cancer; **Supplement Figure 9A-1**). When adding PheRS to a model with PGS, age, and sex, the HRs of the PGS do not change (**Supplement 9A-2**). In all three studies, the PheRS and PGS were correlated for most diseases. However the correlation was small with an average Pearson's *r* of 0.022 (**Figure 5B**, **Supplement Figure 9B**, Supplement Table 7). The overall variance explained by the scores was low, with the highest Nagelkerke pseudo-R² for prostate cancer around 0.02 but largely consistent across the three studies. When interpreting these results, it is important to keep in mind that we predicted the disease onset in a very specific 8-year window and excluded all individuals with an earlier onset of disease. Additionally, the pseudo-R² does not have a defined confidence interval and is not directly comparable to the coefficient of determination R² of a linear regression. With the exception of T2D, the correlation tended to be lower between the two scores the more predictive the PGS were. This indicates that the low correlation between PheRS and PGS is not just due to the low predictiveness of the two scores. Further, adding PGS to a model with PheRS, age, and sex led to significant improvements for 11/13 in FinnGen, 3/4 in the UKB, and 4/13 in the EstB. In the meta-analysis only the improvements for breast cancer and AF were significant (nominal p-value<0.05; **Supplement Figure 9C**). This can be explained by the difference in c-index for the baseline models with age and sex, making a meta-analysis of the delta c-indices difficult.

Supplement Figure 9

Panel A: 1 - Association of PGS (x-axis) and PheRS (y-axis) scores with each disease. The meta-analyzed HRs (95% CI) for 1-SD increase on the scores after regression out age, sex, and the first 10 PCs. **2** - PGS HRs for 1-SD increase in a model without PheRS (x-axis) and one where we adjust for the effect of PheRS (y-axis). **Panel B:** Correlation (Person's r , y-axis) between the PheRS and PGS scores separately in each study (FinnGen: pyramids, UKB: squares, EstB: circles), with the Nagelkerke's pseudo- R^2 of the PGS (x-axis). **Panel C** - C-index improvements when adding PGS to models with PheRS. C-index of the models with only PheRS, sex, age, and PCs (x-axis) compared to those with added PGS (y-axis). **10** Due to sample overlap with the GWASs, PGS could only be calculated for 4 diseases in UKB (see Methods for details). Diseases with significant differences are labelled (p-values < 0.05; two-tailed p-values based on the z-scores of the HR differences and one-tailed p-values for the c-index increases) and those passing multiple hypothesis testing (p-values < 0.05/13) are marked with *.

We have put a reference to the availability of pseudo-R2 in the Supplement and added a short interpretation to the results section:

Overall, we found that the EHR data and genetic information capture largely orthogonal information as shown by the low correlation between the two scores (average Pearson's $r=0.02$, range 0.00-0.08, Supplement Table 7), which is not just driven by the low predictiveness of the PGS (**Figure 5B**, Supplement Figure 9B).

Comment #3

Comment: 242 phecodes (as mentioned on page 5) seems like a small number. How many did the authors begin with and was the only criterion for selection that the phecode be in at least 1% of any study? It's especially small when considering that only 20% of these were present in all of the compared cohorts.

Response: We thank the reviewer for pointing out that the logic of how we ended up with the final phecode numbers was not sufficiently well described in the original submitted version of the manuscript. We have now added the following description to the revised version of the manuscript:

While we found 527 different 3-digit phecodes recorded in all three studies for at least 5 individuals, only 234 had a prevalence of at least 1% in any of the studies and only 48 were common across all studies (**Figure 4A**).

There were minor changes in the included phecodes in each study, as these numbers were based on prevalence of phecodes of individuals aged 32-70, while the selection of features in the PheRS was based on the whole population. To harmonize the presentation, we have now based the phecode prevalence calculations to individuals aged 32-70. Thus, there were now 234 and not 242 common phecodes.

Comment #4

Comment: If sex, age, and genotyping PCs were regressed out of the pheRS and PRS, then I think Figure 2B should have "C-index: sex + age + genotyping PCs" on the x-axis. Similarly need to add geno PCs to the y-axis.

Response: We agree that this is confusing and have updated Figure 2B and Figure 3 so that only age and sex are regressed out from the PheRS in these figures not involving comparisons with PGS. Originally, we also regressed out the PCs from the PheRS here, but this does actually not make too much sense when not comparing them to the PGS, as the PheRS are diagnosis based and do not include genetic information themselves. In the comparisons of PGS vs PheRS, we also regress out the

genotyping PCs. We have updated the Figure 2 and 3 captions accordingly to state that only age and sex were regressed out here.

We updated the following sentences in the Results and Methods sections:

The effect of age and sex were regressed out from the PheRS and, when comparing PheRS and PGS, also the ten first genetic principal components (PCs) to make the scores comparable.

When only considering PheRS performance we regressed out only age and sex.

Figure 2: PheRS performance across studies.

Panel A: Association between PheRS and disease onset during the prediction period independent of age and sex. The HRs and 95% CIs in each study - FinnGen: yellow, UKB: brown, EstB: green - and meta-analyzed results (red). The HRs are shown for an increase of the PheRS by 1 standard deviation (SD) after regressing out age and sex. **Panel B:** Increase in predictive accuracy when adding the PheRS to a baseline model with age and sex. The meta-analyzed c-indices and 95% CIs of the baseline model (x-axis) compared to a model with added PheRS (y-axis). Diseases with a significant differences are labelled (one-tailed p-values < 0.05 based on the z-scores of the c-index increases) and those passing multiple hypothesis testing (p-value < 0.05/13) are marked with *.

Comment #5

Comment: In Figure 2B, there is a very marginal improvement of the pheRS compared to the baseline model. While the authors report that 7/13 of the diseases were significantly better at p < 0.05 compared to baseline, did they perform multiple hypothesis correction? It looks like MDD and Gout may be the only ones to survive.

Response: Figure 2B creates some confusion because it reports the meta-analyzed results. Due to the large differences across studies in c-index already for the baseline models (age+sex), the meta-analyzed results show large confidence intervals. This masks the improvement in performances seen within each study. Supplement Figure 1, however, shows that there are indeed many improvements when looking at each study separately and these are also significant when adjusting for multiple hypothesis correction for all studies and diseases (N=39).

We have now highlighted this aspect in the manuscript

However, in the meta-analysis, the differences in baseline (age+sex) c-index in the different studies meant that only the improvements for MDD, Gout, epilepsy, and asthma were significant (p-value < 0.05; **Figure 2B**).

We have also added multiple hypothesis testing corrections within each study (N=13) to the results and report differences both at the nominal and the Bonferroni corrected level. In most figures the disease are labeled if the nominal p-value was < 0.05 and those diseases that also passed the multiple hypothesis corrected p-values < 0.05/13 are marked with a star (*). We have modified the text in Results section as follows:

Figure 2: Diseases with a significant differences are labelled (one-tailed p-values < 0.05 based on the z-scores of the c-index increases) and those passing multiple hypothesis testing (p-value < 0.05/13) are marked with *.

Figure 3: Diseases with no significant differences are labelled (two-tailed p-values < 0.05 based on the z-scores of the HR differences) and those passing multiple hypothesis testing (p-value < 0.05/13) are marked with *.

Figure 5: Diseases with significant differences are labelled (p-values < 0.05; two-tailed p-values based on the z-scores of the HR differences and one-tailed p-values for the c-index increases) and those passing multiple hypothesis testing (p-values < 0.05/13) are marked with *.

Comment #6

Comment: The authors state that pheRS were strongly correlated with the number of phecodes an individual had listed. This is concerning and makes me wonder if this is biological (and truly predictive) or an artifact of the dataset (such as some doctors are more descriptive and prescribe more phecodes even if redundant). Is the number of listed phecodes a covariate that the authors should have in the baseline model as well? Especially while different cohorts have different phecodes available? How would Figure 2B change if the x-axis baseline model was age + sex + geno PCs + number of phecodes per individual? Moreover, was there a systematic difference between number of phecodes for one cohort vs another?

Response: We agree with the reviewer that it is interesting that we observed the number of different diagnoses an individual has is correlated with the PheRS for some diseases. One would expect the number of different diagnoses to be a proxy for overall health - or disease burden - of the individual, somewhat similarly as in the Charlson Comorbidity Index that creates a mortality predictor by counting the number of diagnoses of high burden common diseases.

As suggested by the reviewer, we have now created additional baseline models incorporating the number of distinct phecodes on top of age and sex, and compared the performance of this model to the PheRS across the cohorts in new Supplement Figure 1A-2 and Supplement Figure 2.

First, we saw that the PheRS also improved the c-index over this extended baseline (Supplement Figure 1A-2). Second, we found that the PheRS were generally more predictive of diseases than just using the number of distinct phecodes (Supplement Figure 2A). Third, we showed that, after adjusting for the number of distinct phecodes, the magnitude of the PheRS HRs reduces only slightly (Supplement Figure 2B) indicating that the number of distinct phecodes is unlikely to be the sole driver of the observed associations between PheRS and diseases.

Extending these analyses to the number of unique phecodes is interesting but beyond the scope of this manuscript. We have expanded the description of the correlation between the number of phecodes and the PheRS in the Results section of the revised manuscript as follows:

While the two predictors were strongly correlated, we found that the PheRS improved predictions also over an extended baseline model accounting for the number of unique diagnoses for 9/13, 11/13 and 4/13 diseases in FinnGen, UKB and EstB, respectively (Supplement Figure 1A-2) and that the magnitude of the PheRS HRs was only slightly reduced for 1/13 diseases in FinnGen, 0/13 in the UKB, and 5/13 in the EstB (Supplement Figure 2).

Further, we added a short description to the Supplement Methods:

The unique number of phecodes is a simple count of the number of phecodes for which an individual had at least one recorded diagnosis during the observation period.

and expanded the Discussion section as follows:

PheRS outperformed a baseline of age and sex for all 13 diseases across the studies (except for breast and prostate cancers in the EstB), and significantly improved over counting the number of previous diagnoses for 9, 11 and 4 diseases in FinnGen, UKB and EstB, respectively, indicating that different existing diagnoses contribute to disease risk differently depending on the target disease predicted.

Supplement Figure 2

Panel A - Association of the unique number of phecodes (x-axis) vs. PheRS (y-axis) with each disease. **Panel B** - Association of the PheRS with each disease in a model without (x-axis) and with the unique number of phecodes (y-axis). Diseases with a significant difference are labelled (two-sided p-values < 0.05) and those passing multiple testing (p-values < 0.05/13) are marked with a star.

Comment #7

Comment: In Figure 2B, another reason why the pheRS might perform better is simply the addition of an extra feature. Therefore, what would the figure look like if the authors reported the adjusted R2 between the baseline model and the pheRS model? Does the c-index account for multiple predictors like an adjusted R2 does?

Response: The C-index measures how well the predictor is able to place the test individuals in correct order based on the incidence times of the disease of interest. Thus adding variables that add predictive power to the model will lead to significant improvements in the C-index. As can be seen from Supplement Figure 3A, adding Charlson Comorbidity Index (CCI) on top of the age+sex leads to clearly smaller improvements than when adding PheRS (Supplement Figure 1A). This indicates that the improvement in C-index is not simply due to having one additional predictor, but how predictive is the predictor, matters. As the Reviewer might be interested in other metrics to quantify improvements in prediction performance, we have also added a new Supplement Table 6 including Nagelkerke's pseudo-R² improvements across models.

Comment #8

Comment: Figure 3A would visually benefit from adding error bars on the correlations.

Response: Thank you for the comment, the correlations shown in the Figure have error bars which are very small. We have added the description of how error bars are shown and computed to Figure captions when missing.

Comment #9

Comment: I was hoping that the authors did more to deal with the fact that there are different phecodes available (or found with at least 1% prevalence) in different cohorts. What if the authors imputed the "missing" phecodes for other cohorts (since they are highly correlated anyway) such that they have data for all 242 phecodes for each individual of each cohort. This might help the interpretation of the finding that the number of recorded phecodes was highly predictive of the pheRS and would allow some standardization. This might also help out-of-cohort prediction.

Response: We agree with the reviewer that a major hurdle affecting the transferability of the PheRS is the differential usage of the diagnostic codes in different cohorts. We think there are two main reasons for why certain codes are not found only in certain studies: 1) No primary care data, as is the case with UKB and FinnGen, but not with EstB, or 2) Different codes are used in different studies as a marker for the same underlying diagnosis.

Out of these two reasons, the former could perhaps be addressed using imputation, but it is likely very challenging given the fact that Finland, UK and Estonia have different healthcare systems and it would be complicated to impute primary care codes from in/outpatient registers. Reason 2 would require "imputation" of the meaning of the codes in the contexts of the different healthcare systems, and equating the codes with similar meanings across the cohorts. We feel that both of these questions are interesting, but hard to address and are outside of the scope of this work.

Here, we have made the observation that elastic net models seem to allow the PheRS models to utilize multiple correlating features, and this seems to alleviate the problem of differential code usage across the cohorts.

Nevertheless, the point that the reviewer raises is important, and we have added some discussion to the Discussion section of revised version of the manuscript better highlighting this issue as a potential future research direction:

Regardless, differences in the types of data available in different cohorts remains a challenge. For example in our study, EstB was the only study with primary care information. Such fundamental differences would be best answered by focusing on compiling comprehensive datasets. Even if all the studies would include the same sources of health data, medical codes are used differently in different countries to describe the same underlying condition. This would require approaches that can map the meaning of the codes used, rather than the actual codes.

Moreover, we have included a sensitivity analysis in the revised version of the manuscript, where we train additional consensus PheRS models across the studies so that only the codes common in all three cohorts are used as features. This analysis is presented in the new Supplement Figure 7, with the following description in the Supplement Results:

Supplement Figure 7. In this sensitivity analysis, we train additional consensus PheRS models across the studies with only phecodes common (prevalence of at least 1%) to all three studies. When considering internal testing, we see a significant decrease in performances for 7/13, 13/13 and 8/13 models in FinnGen, UKB and EstB, respectively, with only one minor improvement for breast cancer in the UKB. However, interestingly, when considering external testing, we find a stronger association for 8/13 diseases in FinnGen with the EstB-trained models and only for 2/13 diseases a weaker association. This is likely explained by the similarity of the EstB and FinnGen data, with the main difference being the wider availability of primary care data in EstB compared to FinnGen. For the other models, the changes are comparatively minor. Performance in the external test set decreases significantly for epilepsy and gout when testing the UKB-trained models in FinnGen, for asthma and gout when testing the EstB-trained models in FinnGen and for knee OA, epilepsy and lung cancer when testing the EstB-trained models in UKB.

Association of PheRS trained all phecodes vs. only common phecodes with each disease

A - internally-tested

B - externally-tested

Supplement Figure 7

Association of PheRS trained on all phecodes in a study (x-axis) vs. PheRS trained only on phecodes common in all three studies (y-axis). Phecodes are considered common if they are recorded for at least 1% of the population during the observation period. **A** - PheRS trained and tested in the same study. **B** - PheRS trained and tested in different studies. The first and second plot show the PheRS tested in FinnGen and trained in the UKB and EstB. The third plot shows the PheRS tested in the UKB and trained in the EstB. Diseases with a significant difference are labelled (two-tailed p-values < 0.05 based on the z-scores of the HR differences) and those passing multiple hypothesis (p-value < 0.05/13) testing are marked with *.

Comparing consensus PheRS with the original pheRS, we observed a significant decrease in the within-study PheRS performances for 7/13, 13/13 and 8/13 models in FinnGen, UKB and EstB, respectively, with only one minor improvement (breast cancer in the UKB). However, we see improvement or comparable performances when considering external validation. Here we found a stronger association of the consensus PheRS for 8/13 diseases in FinnGen with the EstB-trained models and only for 2/13 diseases a weaker association. For the remaining models, the differences were minor. This observation is likely explained by the similarity of the EstB and FinnGen data, with the main difference being the broader availability of primary care data in EstB compared to FinnGen. We have also added the following description of this analysis to the Results section:

Notably, we found that training PheRS only using codes common (prevalence of at least 1%) to all three studies increases PheRS generalizability from EstB to FinnGen (Supplement Figure 7).

Comment #10

Comment: Regarding out of cohort prediction, when sparsity-enforcing regression like lasso/enet picks 1 of several highly correlated phecodes as a predictive feature for cohort A and that specific phecode isn't recorded or isn't prevalent in cohort B, how do the authors handle this? This is obviously going to make the out of cohort prediction take a hit and immediately restricts transferability across healthcare systems in a needlessly random way. Related to the above, imputation might help here.

Response: In general, when a specific phecode was not observed in the FinnGen test set, the corresponding coefficient of the externally-trained PheRS is multiplied by zero and could potentially restrict transferability, as the reviewer argues (see also response to reviewer #1 comment #10). We have now clarified this by adding the following to the Results of the revised manuscript:

All PheRS models for a given disease share the same phecode predictors. When a phecode was not observed in a study, but has a non-zero coefficient in a model trained in an external study, the coefficient is multiplied by zero and does not affect the prediction.

While it is true that sparsity-enforcing regressions might reduce the generalizability of the models, we have performed comparisons of elastic net and ridge, and found similar results. To underline this point, we show in Reviewer's Figure 1 below a comparison of the HRs between PheRS trained with elastic net and ridge regression in the UKB and FinnGen. Specifically, when testing performance within a study, the only difference in FinnGen was that the epilepsy model performs slightly better using elastic net, while in UKB epilepsy, CHD, AF, asthma, knee and hip OA and prostate and breast cancer ridge regression models performed better. Importantly, when testing performance across cohorts (UKB→FinnGen) the PheRS associations were almost identical between the two models except for gout, where elastic net transfers slightly better. In the light of these results, we have decided to keep elastic net as the method of choice for fitting the PheRS. Experimenting with different types of models for PheRS is an interesting avenue for future work.

Editorial Figure 1. A) Hazard ratios (HRs) from PheRS fitted using ridge regression (y-axis) and elastic net (x-axis) in FinnGen and tested in FinnGen test set. Both models include also age and sex as predictors. **B)** Same as A, but training and testing in UKB. **C)** Training in UKB, testing in FinnGen.

Colors indicate different diseases, and diseases where the difference between the two models is nominally significant (p-values < 0.05; two-tailed p-values based on the z-scores of the HR differences and one-tailed p-values for the c-index increases) are labelled, with additionally the ones passing multiple hypothesis testing correction starred (p-values < 0.05/13).

Comment #11

Comment: Figure 4D could be misleading because phecodes are highly correlated with one another, and the authors used feature selection models that randomly chose between correlated features (e.g. lasso/enet). Therefore, I'm wondering, if you remove each of the top predictors, one at a time in a leave-one-out analysis, how does it affect model performance? Does it simply identify other phecodes that are correlated and predictive? Specifically, I'm interested in whether "sleep disorders" is really the most predictive feature, or if it's just a proxy for something else being captured by other phecodes?

Response: We agree with the reviewer that the use of the elastic net penalty introduces some randomness in how model weights are distributed among correlated features, especially when these features are similarly predictive of the outcome. This underscores the need for caution when interpreting individual coefficients in the PheRS models. However, as we demonstrated in the answer to the previous comment that there is minimal performance difference between elastic net and ridge regression models, supporting the robustness of our approach.

Given the complex nature of MDD and its comorbidities, it is reasonable to interpret phecodes such as "sleep disorders" and "alcohol abuse" as proxies for broader underlying factors that may not be fully captured in our data and are challenging to disentangle. Nonetheless, it is notable that these diagnoses contribute meaningfully to the prediction of an individual's future risk of an MDD diagnosis.

In response to the reviewer's suggestion, we have added a new analysis to the revised manuscript. Supplement Figure 11 now includes a leave-one-out (LOO) analysis in FinnGen, in which we systematically removed each of the top 20 features (ranked by absolute weight in the original model) one at a time and evaluated the impact on model performance. To manage computational complexity, we used ridge regression for these analyses, as elastic net models with hyperparameter tuning would have been significantly more resource-intensive to fit.

Our results reveal that, while the removal of a few predictors, such as "chronic airway obstruction" for lung cancer prediction and "hypertension" for gout, led to significant reductions in AUC, the removal of most features resulted in only minimal changes. This aligns with our understanding of the highly correlated nature of phecodes and the complex interplay of risk factors for each disease, which are only imperfectly captured in EHR data. We have added description of this analysis into Supplement Results:

Supplement Figure 11. This Figure shows the results of a leave-one-out analysis performed in FinnGen. We sequentially removed each of the top 20 predictors (ranked by absolute weight in the original elastic net model) and evaluated the performance of the resulting ridge regression models. Ridge regression was used due to the computational complexity of fitting multiple elastic net models with hyperparameter tuning and we have previously observed that, in particular in FinnGen, there is little performance difference between ridge and elastic-net models (**Supplement Figure 11A**). The

analysis shows that removing some predictors, such as "chronic airway obstruction" (in the lung cancer model) and "hypertension" (in the gout model), led to significant reductions in AUC, indicating these features carry unique predictive information. In contrast, the removal of most other top predictors resulted in only minimal changes in model performance (**Supplement Figure 11B**). This observation reflects the highly correlated nature of phecodes and the complex interplay of risk factors, which are only partially captured in EHR data. Overall, the LOO analysis highlights that no single predictor dominates the model and that the models distribute weights across correlated features effectively.

Supplement Figure 11

Panel A: Hazard ratios (HRs) from PheRS fitted using ridge regression (y-axis) and elastic net (x-axis) in FinnGen and tested in the FinnGen test set. Both models also included age and sex as predictors. **Panel B:** Leave-one-out analysis of the top 20 predictors of each PheRS model for the different diseases, using ridge regression. Area under the receiver-operating curve (AUC) of each PheRS model, with the bars indicating the 95% CI intervals of the full model with all predictors using bootstrapping.

In light of this new analysis, we have also updated the Results section describing results from Figure 4 as follows:

An additional leave-one-out analysis (LOO) performed in FinnGen (Supplement Figure 11) showed that removing most top predictors only minimally reduced the AUCs, underscoring the shared contribution of correlated features to the overall model performance.

and added a section to the methods:

The leave-one-out analysis assessing impact of removal of individual phecodes to PheRS performance was performed using ridge penalty instead of elastic net. This was done to cut running time significantly, as using ridge removes the L1 to L2 ratio hyperparameter and its optimization. Otherwise, the ridge models were fitted similarly as the elastic net models. Before running the LOO analysis, we tested that switching to ridge did not generally hurt the PheRS performance in FinnGen (Supplement Figure 11A).

Comment #12

Comment: On page 10, the authors say that pheRS and PRS associations were both significantly associated with disease in the meta-analysis. Was multiple hypothesis testing correction applied? If no, why not?

Response: We have now added multiple hypothesis testing corrections to the results, and report differences both at the nominal and the Bonferroni corrected level. In most figures the disease are labeled if the nominal p-value was < 0.05 and those diseases that also passed the multiple hypothesis corrected p-values $< 0.05/13$ are marked with a star (*). We have modified the text in Results section as follows:

Figure 2: Diseases with a significant differences are labelled (one-tailed p-values < 0.05 based on the z-scores of the c-index increases) and those passing multiple hypothesis testing (p-value $< 0.05/13$) are marked with *.

Figure 3: Diseases with no significant differences are labelled (two-tailed p-values < 0.05 based on the z-scores of the HR differences) and those passing multiple hypothesis testing (p-value $< 0.05/13$) are marked with *.

Figure 5: Diseases with significant differences are labelled (p-values < 0.05 ; two-tailed p-values based on the z-scores of the HR differences and one-tailed p-values for the c-index increases) and those passing multiple hypothesis testing (p-values $< 0.05/13$) are marked with *.

Minor comment #1:

Comment: There are some typos throughout the paper that if fixed would improve the presentation of the writing and the findings.

- 1) Page 1 last line: "the PGS for 8/13 disease", needs an s.
- 2) Page 2: "All of US", the s should be lowercase.
- 3) Page 2: values such as 1.347 and 1.568 should be presented with commas.
- 4) Figure 1: $-\log_{10}$ p-value on the y-axis of manhattan plot is upside down, and would look better right side up.
- 5) Please define "washout period" used on page 5
- 6) Page 6: "4/13 disease", needs an s.
- 7) Page 7: "moderate to strongly", should be moderately.

Response: We thank the reviewer for pointing out this typos. We have fixed all of them according to the reviewer's suggestions in the revised version of the manuscript.

Remark on Code availability

The code provides a resource for recreating the INTERVENE PheRS, including a tutorial, data / dummy data, and documentation on the inputs and different analyses (including downsampling). The instructions provided are sufficient to install and run the application.

Other changes

Due to the various updates to Figures and additional Supplement Figures, most of the Supplement text has changed significantly.

Citation retracted

A study we cited (citation number 31 in the original submitted manuscript) was retracted while we were revising the manuscript (<https://www.nature.com/articles/s41467-024-48568-8>) but was re-published on the 10th of January 2025. We updated the citation.

Prevalences updates

The prevalence of phecodes shown in Figure 4 of the original manuscript were based on individuals aged 32-70, while the selection of features in the PheRS was based on the whole population. To harmonize the presentation, we have now based the phecode prevalence calculations in Figure 4 also to individuals aged 32-70, and changed the following figures and statements with only minor changes overall:

In total, we considered 234 [previously 242] phecodes with a prevalence of at least 1% in any study.

When considering codes with a prevalence of at least 1%, only 21% [previously 20%] of phecodes (N=48 [previously 49]) [...] For example, the inclusion of primary care diagnoses in the EstB study leads to a higher number of phecodes, with 33% [previously 32%] (N=77) [...]

Figure 4A - 51 shared phecodes between FinnGen and EstB [previously 58]. 21 phecodes FinnGen specific [previously 24]. 49 phecodes shared between all three studies [previously 49]. 9 phecodes shared between FinnGen and UKB [previously 7]. 9 phecodes shared between EstB and UKB [previously 8].

Figure 4B - Minor changes to bars, general trend stays the same.

Further, the original Supplement Table 6 and 7 (now Supplement Table 8 and 9) were updated with the new prevalences.

New Tables

New Supplement Table 5 - AUPRCs

New Supplement Table 6 - Pseudo-R²

All following tables numbers shifted by 2

New Supplement Table 9 - Correlation of internal- and externally-trained PheRS.

All following tables numbers shifted by 3

New Text

We added the following results to the discussion to enhance the comparison of PheRS and PGS:

Overall, we observe very low correlation between PGS and PheRS **and no significant changes in HRs when adding the PGS to a model with the PheRS (and vice versa)**, indicating that these two data sources contain largely independent information that is predictive of disease onset.

New Figures

Updated Supplement Figure 3

In order to fit the new Figures for the new extended baseline with the number of unique phecodes, we moved all other c-index comparisons from Supplement Figure 1-3 with the CCI, Education and

PheRS into a single Supplement Figure 3. We removed Supplement Figure 2B from the original submission due to lack of space and relevance.

A - Prediction improvements with CCI and education over baseline in each study

1 - CCI

2 - Education

3 - CCI + Education

Supplement Figure 3

Increase in prediction accuracy when adding: CCI (**Panel A**), education (**Panel B**), and both CCI and education (**Panel C**), to a baseline model with age and sex in all three studies. All panels show the c-index and 95% CIs of the baseline model (x-axis), compared to a model with added predictors (y-axis). Diseases with significant increases in c-index are labelled (p-values < 0.05; one-tailed p-values based on the z-scores of the c-index increases) and those passing multiple hypothesis testing (p-values < 0.05/13) are marked with *.

Updated Supplement Figure 4

and moved the remaining Figures for the CCI and Education into a single Supplement Figure 4

A - Prediction improvements with PheRS over an extended baseline in each study

B - Association of the CCI with each disease

1 - top 10% vs. rest

2 - per 1-SD

C - Association of education with each disease

D - Correlation of PheRS and CCI

Supplement Figure 4

Panel A: Increase in prediction accuracy when adding the PheRS to an extended baseline with age, sex, CCI, and education in all three studies. Association of the CCI with each disease in a model without (x-axis) and with the PheRS (y-axis). Diseases with significant increases in c-index are labelled (p-values < 0.05; one-tailed p-values based on the z-scores of the c-index increases) and those passing multiple hypothesis testing (p-values < 0.05/13) are marked with *. **Panel B:** The HRs of the CCI for the top 10% of individuals with the highest score compared to the rest 90%. Age and sex were regressed-out from the CCI. The top 10% corresponds largely to individuals with a CCI>=2 and a few younger ones with a CCI of 1. **Panel C:** This shows the increase in relative risk (HR and 95% CI) with 1-SD increase of the CCI. **Panel D:** Association of lower education with each disease. Here we show the increase in relative risk (HR and 95% CI) for individuals with lower education (ISCED-11<5) compared to those with high achieved education level (ISCED-11>=5). **Panel E:** Correlation of the PheRS and CCI after regressing-out the effect of age and sex.